# OVERSMOOTHING AS LOSS OF SIGN: TOWARDS STRUCTURAL BALANCE IN GRAPH NEURAL NETWORKS

## ABSTRACT

Oversmoothing is a common phenomenon in a wide range of graph neural networks (GNNs), where node representation becomes homogeneous and thus model performance worsens as the number of layers increases. Various strategies have been proposed to combat oversmoothing, but they are based on different heuristics and lack a unified understanding of their inherent mechanisms. In this paper, we revisit the concept of signed graphs and show that a wide class of anti-oversmoothing techniques can be viewed as the propagation on corresponding signed graphs with both positive and negative edges. Leveraging the classic theory of signed graphs, we characterize the asymptotic behaviors of existing methods and reveal that they deviate from the ideal state of structural balance that provably prevents oversmoothing and improves node classification performance. Driven by this unified analysis and theoretical insights, we propose **S**tructural **B**alanced **P**ropagation (SBP) where we explicitly enhance the structural balance of the signed graph with the help of label and feature information. We theoretically and empirically prove that SBP can improve the structural balance to alleviate oversmoothing under certain conditions. Experiments on synthetic and real-world datasets demonstrate the effectiveness of our methods, highlighting the value of our signed graph framework.

## 1 INTRODUCTION

Graph neural networks (GNNs) are a powerful framework for processing graph-structured data across a wide range of fields, such as drug discovery, recommender systems and social networks (Gori et al., 2005; Scarselli et al., 2009; Bruna et al., 2014; Duvenaud et al., 2015; Defferrard et al., 2016; Battaglia et al., 2016; Li et al., 2016). Most GNN models follow the *message-passing* paradigm, where node features are computed by recursively aggregating information from all neighboring nodes along the unsigned edges (Kipf & Welling, 2017; Wu et al., 2019; Veličković et al., 2018; Xu et al., 2019). Despite notable advancements, oversmoothing remains a prevalent issue for GNNs characterized by the convergence of all node features to a constant when stacking a substantial number of layers (Li et al., 2018; Oono & Suzuki, 2020; Cai & Wang, 2020; Wu et al., 2023a). Several strategies have been developed to counteract oversmoothing in GNNs, such as normalization layers (Guo et al., 2023; Zhao & Akoglu, 2019; Ioffe & Szegedy, 2015), random edge dropping (Rong et al., 2019; Do et al., 2020) and residual connections (Wang et al., 2021; Gasteiger et al., 2018; Xu et al., 2018). However, there lacks a unified understanding of these different strategies for dissecting their limitations, as it seems that these method can only mitigate oversmoothing to certain extent, and the model performance still degrades significantly after a large number of propagation steps (Li et al., 2018; Wu et al., 2023b).

In this paper, we revisit *signed graphs* (Derr et al., 2018; Shi et al., 2019) where each edge has a positive or negative sign and propose that edge signs can serve as a remedy to combat the overmoothing issue. Moreover, we collect eight oversmoothing countermeasures in the existing literature. Our analysis shows that all these methods can be interpreted as injecting negative edges, with different designs, into the original unsigned graph and conducting message-passing over the resulting signed graphs, leading to a clear repulsion among neighboring nodes when oversmoothing was supposed to occur when edges are unsigned.

This observation not only reveals how these methods work to alleviate oversmoothing, but also inspires us to further make use of the signed graph to theoretically address the oversmoothing issue. We explore a fundamental signed graph property, known as *structural balance* (Cartwright & Harary, 1956). In a structurally balanced signed graph, nodes can be grouped into clusters where only positive edges exist within each cluster, and only negative edges exist between clusters, as illustrated in

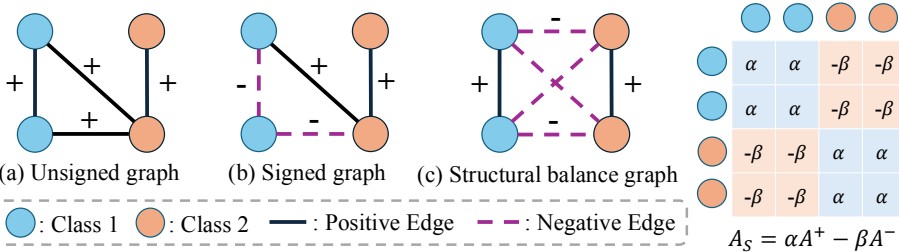

Figure 1: Left: Example of structurally balance graph. (a) is an unsigned graph, (b) is a signed graph, and (c) is a signed graph with the structural balance property. Right: Adjacency matrix $A_s$ of a structural balance graph composed of a positive adjacency matrix $A^+$ and a negative adjacency matrix $A^-$. $\alpha$, and $\beta$ are the weights representing the strengths of attraction and repulsion, respectively.

Figure 1. We demonstrate that message-passing over a structurally balanced signed graph exhibits controllable asymptotic behavior: nodes within the same cluster converge to a shared value, while different clusters repel each other to have distinct values (Theorem 4.3). This scenario theoretically prevents oversmoothing and enhances the node classification accuracy.

Motivated by our structural balance analysis, we propose the theoretically optimized signed propagation: **S**tructural **B**alanced **P**ropagation (SBP) to artificially construct a structurally balanced graph to conduct message-passing. Specifically, we leverage the ground truth in the training set (label) to assign positive edges among intra-class nodes and negative edges among inter-class nodes without any additional learnable parameters, which we denote as Label-SBP. We theoretically demonstrate that Label-SBP induces a structurally balanced signed graph under certain conditions, successfully repelling nodes from different label classes (Proposition 4.7). Built upon Label-SBP, we further design a feature-induced variant to accommodate scenarios without sufficient ground truth information, dubbed Feature-SBP. Empirically, we conduct extensive experiments on nine synthetic and real-world benchmark datasets and SBP consistently outperforms existing oversmoothing counter-measures across all deep depths and datasets. The main contributions of our work are summarized as follows:

- We provide a signed graph perspective to unify three major types of anti-oversmoothing techniques and show that all of them amount to certain implicit designs of adding negative edges to the original graph, highlighting the insight provided by our signed graph perspective.

- We introduce the concept of structural balance from the signed graph theory as an ideal state of graph propagation. By measuring the degree of structural balance, we find that due to the lack of class-awareness, existing oversmoothing couteracting techniques fail at improving the structural balance of the graph, which explains their inefficacy under more propagation steps.

- Driven by the structural balance theory, we propose two parameter-free strategies—Label-SBP and Feature-SBP by explicitly designing the addition of negative edges to improve the structural balance of the graph and thus prevent undesirable oversmoothing across different classes. Experiments on nine synthetic and real-world graphs show that our method attains superior performance at counteracting oversmoothing in both homophilic and heterophilic settings.

## 2 RELATED WORK

**Theory of Oversmoothing**   The concept of oversmoothing was initially introduced by Li et al. (2018): when the number of layers becomes large, the representations of different nodes tend to converge to a common value after excessively exchanging messages with neighboring nodes. Oono & Suzuki (2020); Wu et al. (2023a) rigorously show that the convergence of node representations to a common value happens at an exponential rate as the number of layers increases to infinity, for GCNs and attention-based GNNs, respectively. Wu et al. (2023b) theoretically proves that oversmoothing can start to happen even in shallow depth under certain random graph settings. Zhou et al. (2021) proposed an appropriate residual connection according to the lower limit of Dirichlet energy and connected to previous methods qualitatively.

**Signed Graph-Based Methods**   In the heterophilic graphs, various methods are inspired by the signed graph propagation (Tsitsulin et al., 2023; Song et al., 2023; Yan et al., 2022; Wang et al., 2022;

Table 1: The overall correspondences between positive and negative graphs in signed graph propagation of various anti-oversmoothing methods.

| Method | Characteristic | Positive $A^+$ | Negative $A^-$ |
|---|---|---|---|
| GCN | $K$-layer graph convolutions | $\hat{A}$ | 0 |
| SGC | $K$-layer linear graph convolutions | $\hat{A}$ | 0 |
| BatchNorm | Normalized with column means and variance | $\hat{A}$ | $\mathbb{1}_n \mathbb{1}_n^T / n\hat{A}$ |
| PairNorm | Normalized with the overall means and variance | $\hat{A}$ | $\mathbb{1}_n \mathbb{1}_n^T / n\hat{A}$ |
| ContraNorm | Uniformed norm derived from contrastive loss | $\hat{A}$ | $(XX^T)\hat{A}$ |
| DropEdge | Randomized graph | $\hat{A}$ | $A_m$ |
| Residual | Last layer connection | $\hat{A}$ | $I$ |
| APPNP | Personalized pagerank | $\Sigma_{i=0}^{k+1} \alpha^i \hat{A}^i$ | $\alpha \Sigma_{j=0}^{k} \alpha^j \hat{A}^j$ |
| JKNET | Jumping to the last layer | $\Sigma_{i=0}^{k} \alpha^i \hat{A}^i + \hat{A}^{k+1}$ | $\Sigma_{j=0}^{k} \alpha^i \hat{A}^k$ |
| DAGNN | Adaptively incorporating different layer | $\Sigma_{i=0}^{k} \alpha^i \hat{A}^i + \hat{A}^{k+1}$ | $\Sigma_{j=0}^{k} \alpha^i \hat{A}^k$ |
| Feature-SBP (ours) | Label-induced negative graph | $\hat{A}$ | $-XX^T$ |
| Label-SBP (ours) | Feature-induced negative graph | $\hat{A}$ | $A_l$ |

Chien et al., 2020). In particular, Yan et al. (2022); Wang et al. (2022) utilize the idea that the negative edges denote connections between nodes that are "not similar to each other" to create repulsion between them during message-passing. Chien et al. (2020) extend the coefficients of the output of different layers in the final aggregation to be learnable and find that the odd layer coefficients tends to be negative for heterophilic graphs, suggesting that learning naturally finds signed-graph message-passing. However, Liang et al. (2024b) show that under some specific random graph settings, the oversmoothing will even happen under signed graph propagation which aligns with the case in our analysis for Theorem 4.1 when the repulsion among nodes are not sufficient. Nevertheless, we extend the theory to generic graphs and prove that in the ideal state—structural balance, signed edges can indeed serve as a remedy to effectively combat oversmoothing.

**Structural Balance**    Structual balance theory has gained significant attention in recent years (Derr et al., 2018; Yan et al., 2022; Liang et al., 2024a; Wang et al., 2022). Inspired by the structural balance theory, Derr et al. (2018) characterizes the balanced path intuitively to learn both balanced and unbalanced representations on each layer. Liang et al. (2024a) predicts the signed adjacency matrix by an off-the-shelf neural network classifier to generate pseudo labels with the low-rank assumption. Shi et al. (2019) introduces the definition of the Laplacian for signed graphs and develops a comprehensive mathematical theory. In this paper, we rigorously show that structural balance is the optimal solution under the signed graph message-passing and propose practical solutions to oversmoothing based on the property without any additional learnable parameters.

In addition to the above methods which explicitly make use of the signed graph propagation, in this paper, we also revisit a wide class of previous anti-oversmoothing methods that do not explicitly claim to use signed message-passing. We find that all of them can be attributed to some kind of design of negative edges to the original graph.

## 3   A SIGNED GRAPH PERSPECTIVE ON EXISTING OVERSMOOTHING COUNTERMEASURES

In this section, we introduce a signed graph perspective to unify many popular anti-oversmoothing techniques. Specially, we link eight classic methods to the signed graph propagation and summarize their positive and negative graphs in Table 1 for clarity.

**Notations**    Consider a graph $\mathcal{G} = (V, E)$ with node set $V$ and edge set $E$. $n = |V|$ is the number of nodes. The nodes feature matrix is denoted by $X = \{x_1, x_2, \ldots, x_n\} \in \mathbb{R}^{n \times d}$, where $x_i$ is the feature for node $i$ and $d$ is the dimension of node features. The set of neighbors of a node $i$ is denoted by $N_i$. Let $A \in \mathbb{R}^{n \times n}$ denote the adjacency matrix where $A_{i,j} = 1$ if $\{i, j\} \in E$, otherwise 0. The diagonal degree matrix is denoted as $D = \text{diag}(d_1, \ldots, d_n)$, where $d_i = \sum_j A_{i,j}$. Then the raw normalized adjacency matrix is $\hat{A} = D^{-1}A$. $\mathbb{1}_n$ is the all-one vector of length $n$ and $|| \cdot ||_F$ denotes the Frobenius norm.

**Signed Graphs** The signed graph enables $A$ to incorporate both positive and negative values. In this paper, we define the signed graph adjacency matrix $\hat{A}_s = \alpha\hat{A}^+ - \beta\hat{A}^-$, where $\hat{A}^+$ is the raw normalized version of the positive adjacency matrix $A^+ \in \{0,1\}^{n \times n}$ and $\hat{A}^-$ is that of the negative adjacency matrix $A^- \in \{0,1\}^{n \times n}$, $\alpha, \beta > 0$ are the weight parameters. For convenience, following Wu et al. (2019; 2023a), we simplify the $k$-th ($k \in [1, K]$) layer graph convolutional operation in a similar way by ignoring the non-linear activation with $\sigma(x) = x$ and let $W^* = W^{(0)}W^{(1)}\cdots W^{(K-1)}$. The resulting $K$-layer signed graph propagation (Shi et al., 2019; Derr et al., 2018) is as follows:

$$X^{(k+1)} = (1 - \alpha + \beta)X^{(k)} + \alpha\hat{A}^+ X^{(k)} - \beta\hat{A}^- X^{(k)}, \tag{1}$$

$$X^{(0)} = X, \ H = X^{(K)}W^*. \tag{2}$$

Note that when $\beta = 0, \alpha = 1$, (1) would correspond to the conventional (unsigned) graph propagation. In the following sections, we interpret existing oversmoothing couteracting methods from a signed graph propagation perspective in the form of (1).

## 3.1 Normalization Techniques

Normalization is a series of methods to operate the node features after each message-passing step. A few representative examples include BatchNorm (Ioffe & Szegedy, 2015), PairNorm (Zhao & Akoglu, 2020), and ContraNorm (Guo et al., 2023), where PairNorm and ContraNorm were proposed specifically to address the oversmoothing issue in GNNs. Specifically, BatchNorm centers the node representations $X$ to have zero mean and unit variance across nodes for each feature, which can be written as $\text{BatchNorm}(x_i) = \frac{1}{\sqrt{\sigma^2 + \epsilon}}(x_i - \frac{1}{n}\Sigma_{i=1}^n x_i)$ where $\epsilon > 0$ and $\sigma^2$ is the variance of the feature across all nodes. Meanwhile, PairNorm is a normalization technique specifically developed for GNNs to combat oversmoothing, where its only difference from BatchNorm is that PairNorm scales all the entries in $X$ using the same number rather than scaling each column by its own variance. It can be written as $\text{PairNorm}(x_i) = \frac{s}{\sqrt{\Gamma^2 + \epsilon}}(x_i - \frac{1}{n}\Sigma_{i=1}^n x_i)$ where $\Gamma = \|(\hat{A} - \mathbb{1}_n\mathbb{1}_n^T/n)X\|_F/\sqrt{n}$ and $s$ is a scalar. Apart from these two methods, ContraNorm is inspired by the uniformity loss from contrastive learning, aiming to alleviate dimensional feature collapse. For simplicity, we consider the spectral version of ContraNorm that takes the following form: $\text{ContraNorm}(X) = (1+\alpha)X - \alpha/\tau(XX^T)X$ where $\alpha \in (0, 1)$ and $\tau > 0$ are hyperparameters. We discuss further details about these three normalization methods in Appendix C.

Despite the differences in motivation and implementation, all the three normalization methods can be seen as a signed graph propagation with different designs of the negative graph:

**Theorem 3.1** *BatchNorm, PairNorm and ContraNorm can be interpreted as signed graph propagation defined in (1), sharing the same positive adjacency matrix $\hat{A}^+ = \hat{A}$ while having different negative adjacency matrices transformed from $\hat{A}^+$, such as $\hat{A}^- = \frac{\mathbb{1}_n\mathbb{1}_n^T}{n}\hat{A}$ for BatchNorm and PairNorm, and $\hat{A}^- = (XX^T)\hat{A}$ for ContraNorm.*

The result shows that PairNorm shares the same fixed positive and negative graphs (up to scale) as BatchNorm. In contrast, ContraNorm extends the negative graph to an adaptive one based on similarities in node features.

## 3.2 Augmentation-based Methods

Node or edge dropping (Rong et al., 2019) is another popular type of empirical method deployed to combat oversmoothing. Denote $A_m \in \{0,1\}^{n \times n}$ as the negative adjacency matrix where $(A_m)_{i,j} = (\hat{A})_{i,j}$ if the edge $\{i,j\}$ is dropped and otherwise 0. Then the signed propagation after (randomly) dropping some edges can be formulated as: $\hat{X} = \hat{A}X - A_m X$. This simplified negative graph derived from the random generation shows that just the existence of the negative edges raises the chance of alleviating oversmoothing.

## 3.3 Residual Connections

Besides normalization layers and edge-dropping, residual connections can also be seen through the lens of signed graph propagation. Given the variety of methods in this class, we provide analysis for the following three types of residual connections.

The standard residual connection (Wang et al., 2021; Chen et al., 2020a) directly combines the previous and the current layer features together. It can be formulated as: $\hat{X} = (1 - \alpha)X + \alpha\hat{A}X$. Another residual connection type propagation, inspired by PageRank, is APPNP (Gasteiger et al., 2018) which can be viewed as a layer-wise graph convolution with a residual connection to the initial feature matrix $X^{(0)}$, written as: $\hat{X}^{(k+1)} = (1-\alpha)X^{(0)} + \alpha\hat{A}X^{(k)}$. In addition to combining with the last and initial layer features, the last type of residual connections integrates several intermediate layer features, which includes representative methods such as JKNET (Xu et al., 2018) and DAGNN (Liu et al., 2020). JKNET selectively combines aggregations from different layers through operations such as concatenation or max-pooling at the output, i.e., the representations "jump" to the last layer. Deep Adaptive Graph Neural Networks (DAGNN) (Liu et al., 2020) tries to adaptively add all the features from the previous layer to the current layer features with the additional learnable coefficients. Formally, we establish the following theorem that all these three types of residual connections can be seen as signed graph propagation. More detailed dicussion can be found in Appendix C.2:

**Theorem 3.2** *With $\hat{A}^+ = \hat{A}$ and $\hat{A}^- = I$, propagation under standard residual connections follows the signed graph propagation defined in (1). With $\hat{A}^+ = \Sigma_{i=0}^{k+1}\alpha^i\hat{A}^i$ and $\hat{A}^- = \alpha\Sigma_{j=0}^{k}\alpha^j\hat{A}^j$, propagation under APPNP follows the signed graph propagation in (1). With $\hat{A}^+ = \Sigma_{i=0}^{k-1}\alpha^i\hat{A}^i + \hat{A}^k$ and $\hat{A}^- = \Sigma_{j=0}^{k-1}\alpha^j\hat{A}^k$, propagation under JKNET and DAGNN follows the signed graph propagation in (1).*

The above discussion reveals that many existing methods can be unified and interpreted through the signed graph perspective. Normalization layers create the negative graph through linearly transforming the adjacency matrix, augmentation methods do so by randomly masking their elements as 0, and connection methods construct them by combining different orders of the adjacency matrix linearly. However, while empirically constructing the signed graph propagation may initially provide some benefits in terms of preventing oversmoothing, there is still a lack of theoretical guidance to fully understand the complex interplay between the signed graph structure and the resulting node feature dynamics.

# 4 STRUCTURAL BALANCED PROPAGATION

In this section, we first uncover the limitation of the general signed graph propagation. Then, we prove that the optimal theoretical solution of the signed graph propagation is the *structural balance graphs*. Inspired by the structural balance theory, we design an efficient and easy-to-implement method: **S**tructural **B**alanced **P**ropagation (SBP).

## 4.1 STRUCTURAL BALANCE THEORY

We first emphasize that insufficient repulsion leads to oversmoothing under the signed graph propagation, while excessive repulsion can also hinder performance.

**Theorem 4.1** *Suppose that the positive adjacency matrix $A^+$ represents a connected graph and $x_i(t)$ represents the value of node $i$ after $t$ rounds of the signed propagation in equation 1. Then along (1) for any $0 < \alpha < 1/\max_{i \in V} d_i^+$, there exists a critical value $\beta_* \geq 0$ for $\beta$ such that if $\beta < \beta_*$, then we have $\lim_{t\to\infty} x_i(t) = \sum_{j=1}^n x_j(0)/n$ for all initial values $x(0)$; if $\beta > \beta_*$, then $\lim_{t\to\infty} \|x(t)\| = \infty$ for almost all initial values w.r.t. Lebesgue measure.*

Theorem 4.1 suggests that if the weight $\beta$ assigned to negative edges is small, especially when $\beta = 0$ which represents the standard unsigned graph propagation, node features will converge to a common value unavoidably, resulting in oversmoothing. But if the weight $\beta$ is significantly large, the node features will diverge rather than converge. The effect of the weight $\beta$ represents the repulsion between the nodes, and this repulsive force serves to mitigate the homogenizing trend, preserving the heterogeneity of node features within the network.

In what follows, we present a theoretically optimal solution for signed graph propagation, known as the structural balance graph. Formally, following Shi et al. (2019); Cartwright & Harary (1956), we define structural balance graphs as follows.

**Definition 4.2 (Structural Balance)** *A signed graph $\mathcal{G}$ is **structurally balanced** if there is a partition of the node set into $V = V_1 \cup V_2$ with $V_1$ and $V_2$ being nonempty and mutually disjoint, where any edge between the two node subsets $V_1$ and $V_2$ is negative, and any edge within each $V_i$ is positive.*

The structural balance property divides the graph into two disjoint groups ($V_1$ and $V_2$) and separate intra-group and inter-group edges by their signs as illustrated in Figure 1(c). Moreover, Theorem 4.1 shows that although negative edges can mitigate the convergence of node representations, unbounded divergence would happen when the repulsion is too strong, which would also harm the model's performance due to issues such as numerical instability (Wang et al., 2022). Then we prove that under a bounded function to constrain the node features from diverging, the convergence only occurs within each node group separately, which is even beneficial for node classification tasks.

**Theorem 4.3** *Assume that node $i$ is connected to node $j$ and $x_i(t)$ represents the value of node $i$ after $t$ round of propagation in (1). $\mathcal{F}(z)_c$ is a bounded function satisfying: if $z < -c$, $\mathcal{F}(z)_c = -c$; if $z > c$, $\mathcal{F}(z)_c = c$; if $-c < z < c$, $\mathcal{F}(z)_c = z$. Let $\theta = \alpha$ if the edge $\{i,j\}$ is positive and $\theta = -\beta$ if the edge $\{i,j\}$ is negative. Consider the constrained signed propagation update:*

$$x_i(t+1) = \mathcal{F}_c((1-\theta)x_i(t) + \theta x_j(t)). \tag{3}$$

*Let $\alpha \in (0, 1/2)$. Assume that $\mathcal{G}$ is a structurally balanced complete graph under the partition $V = V_1 \cup V_2$. When $\beta$ is sufficiently large, for almost all initial values $x(0)$ w.r.t. Lebesgue measure, there exists a binary random variable $l(x(0))$ taking values in $\{-c, c\}$ such that*

$$\mathbb{P}\left(\lim_{t\to\infty} x_i(t) = l(x(0)), i \in V_1; \lim_{t\to\infty} x_i(t) = -l(x(0)), i \in V_2\right) = 1. \tag{4}$$

Theorem 4.3 shows that if the graph is structurally balanced and is constrained with $\mathcal{F}_c(z)$, node features converge to their respective group-specific values asymptotically under the signed graph propagation defined in (1). We discuss the relationship of Theorem 4.1 and Theorem 4.3 to over-smoothing in Appendix F. Furthermore, different groups will repel each other to have distinct values even asymptotically.

**Remark 4.4** *We can generalize the two distinct groups in the above result to a generic number of distinct groups by introducing a more general notion, weakly structural balance. See a detailed discussion in Appendix G.*

### 4.2 Measure of Structural Balance in Practice

Despite Theorem 4.3 establishes that the structural balanced graph can provably alleviate oversmoothing, the condition seems hard to satisfy in practice. Thus, measuring the degree of structural balance of a signed graph in real scenarios is necessary. Borrowing from a classic literature (Harary, 1959), we define the *structural imbalance degree* ($\mathcal{SID}$) of a graph by counting the number of edges that must be changed to achieve the structural balance:

**Definition 4.5 (Structural Imbalance Degree)** *For each node $v$ in a graph $\mathcal{G}$ of $n$ nodes, let $\mathcal{P}(v)$ denote the subset of nodes that has the same label as $v$ but connected to $v$ by a non-positive edge; let $\mathcal{N}(v)$ denote the subset of nodes that has a different label from $v$ but connected to $v$ by a non-negative edge. Then the structural imbalance degree of $\mathcal{G}$ is defined as $\mathcal{SID}(\mathcal{G}) = \frac{1}{2n} \sum_{v\in\mathcal{G}}(|\mathcal{P}(v)|+|\mathcal{N}(v)|)$.*

**Proposition 4.6** *For a structural balanced complete graph $\mathcal{G}$, we have $\mathcal{SID}(\mathcal{G}) = 0$.*

We observe that $\mathcal{SID}$ increases as more edge signs deviate from the ideal structural balance complete graph criterion, indicating a growing structural imbalance. Based on the $\mathcal{SID}$, we can measure the distance between a real signed graph and the ideal structural balanced signed graph.

### 4.3 Design Signed propagation for GNNs

To reduce $\mathcal{SID}$ and achieve a more structurally balanced signed graph, we design two methods label-enhanced structural balanced propagation (Label-SBP) and feature-enhanced structural balanced propagation (Feature-SBP) by utilizing the label and feature information to construct the negative graphs respectively.

Table 2: The $\mathcal{SID}$ on CSBM.

| Method | $\mathcal{P}_\downarrow$ | $\mathcal{N}_\downarrow$ | $\mathcal{SID}_\downarrow$ |
|---|---|---|---|
| DropEdge | 92.62 | 100.00 | 96.31 |
| Residual | 90.87 | 100.00 | 95.44 |
| GCN/SGC | 89.87 | 100.00 | 94.94 |
| APPNP | 0.00 | 100.00 | 50.00 |
| JKNET | 0.00 | 100.00 | 50.00 |
| DAGNN | 0.00 | 100.00 | 50.00 |
| BatchNorm | 89.87 | 4.56 | 47.22 |
| PairNorm | 89.87 | 4.56 | 47.22 |
| ContraNorm | 89.87 | 4.56 | 47.22 |
| Feature-SBP (ours) | 89.87 | 4.56 | 47.22 |
| Label-SBP (ours) | 32.46 | 36.16 | 34.31 |

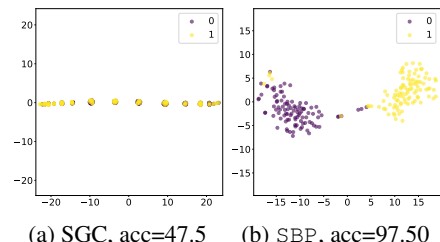

(a) SGC, acc=47.5    (b) SBP, acc=97.50

Figure 2: The t-SNE visualization of the node features on CSBM and Layer=300.

**Label-enhanced Structural Balanced Propagation**    Inspired by the Theorem 4.3, we first utilize the label signal as the negative adjacency matrix to mitigate across-label oversmoothing, termed as **Label**-enhanced **S**tructural **B**alanced **P**ropagation (Label-SBP). In particular, we use the label in the train set to repel nodes from different classes and attract nodes from the same class as follows:

$$(A_l)_{ij} = \begin{cases} 1 & \text{if } y_i \neq y_j\,, \\ -1 & \text{if } y_i = y_j\,, \\ 0 & \text{otherwise}\,, \end{cases} \tag{5}$$

where if nodes $i, j$ have different labels, we assign a positive value 1 to repel them. Conversely, if they have the same label, we assign $-1$ to attract them. Note that if labels are unknown, we assign 0 to neither repel nor attract them. We employ a row-stochastic adjacency matrix $\hat{A}$ as the positive adjacency matrix, which is a non-negative matrix with row sums equal to one, learning the global graph structure. Then we give the following result showing that Label-SBP effectively aggregates classes to separate constants.

**Proposition 4.7** *Assume that node label classes are balanced $|Y_1| = |Y_2|$ and denote the ratio of labeled nodes as $p$. Then we have that the signed graph adjacency matrix $A_s = A - A_l$ and $\mathcal{SID}(\mathcal{G}) \leq (1-p)n/2$, where $\mathcal{SID}$ decreases with a larger labelling ratio $p$. In particular, with full supervision ($p = 1$), we have $\mathcal{SID}(\mathcal{G}) = 0$, i.e., a perfectly balanced complete signed graph. Under the constrained signed propagation equation 3, the nodes from different classes will converge to distinct constants.*

Nonetheless, although Label-SBP can induce the ideal structural balance in the train set, it requires access to labels that might be scarce under a low labeling rate. To deal with this situation, we further propose a variant of our method that estimates a negative graph based on feature similarities.

**Feature-enhanced Structural Balanced Propagation**    If labels are unavailable, we can utilize the similarity matrix derived from the node features to create the negative matrix $A_f$, termed as **Feature**-enhanced **S**tructural **B**alanced **P**ropagation (Feature-SBP). $A_f$ is defined as follows:

$$A_f = -XX^T, \tag{6}$$

where $X \in \mathbb{R}^{n \times d}$ is the initial node feature matrix. While the use of the node feature as the repulsion signal may not be as precise as the Label-SBP, it can make use of all node features and thus be adjusted to the test set to improve the overall structural balance property of the graph.

**Implementation Details of SBP**    As shown in Theorem 4.1 and Theorem 4.3, without a bound function the features will diverge to infinity due to strong repulsion, so we implement the constrained function $\mathcal{F}_c$ by using LayerNorm (Ba et al., 2016). To avoid numerical instability for repeated message-passing, we ensure that the sum of the coefficients combining the node representations $X^{(k)}$ and the node representations updates by our SBP remains 1. Additionally, we apply the softmax function to the negative matrix, resulting in $\hat{A}^- = \text{softmax}(A^-)$. As a result, SBP can be written as:

$$(\text{SBP}) \quad X^{(k+1)} = (1-\lambda)X^{(k)} + \lambda(\alpha\hat{A} - \beta\,\text{softmax}(\hat{A}_l \text{ or } \hat{A}_f))X^{(k)}\,. \tag{7}$$

**Scalability of SBP**    It is noted that the direct application of SBP to large-scale graphs can destroy the sparsity of the original adjacency matrix because we try to assign either a plus or minus sign

Table 3: Node classification accuracy (%) on 8 datasets and $H(G)$ refers to the edge homophily level.The best results are marked in blue and the second best results are marked in gray on every layer. Overall `SBP` performs best in both homophilic and heterophilic datasets.

| $H(G)$
Dataset | 0.81
Cora | 0.74
Citeseer | 0.80
PubMed | 0.22
Squirrel | 0.38
Amazon-ratings | 0.21
Texas | 0.11
Wisconsin | 0.30
Cornell |
|---|---|---|---|---|---|---|---|---|
| MLP | 48.82 ± 0.98 | 47.89 ± 1.21 | 69.20 ± 0.83 | 32.58 ± 0.19 | 38.14 ± 0.03 | 73.51 ± 2.36 | 70.98 ± 1.18 | 68.11 ± 2.65 |
| SGC | 80.21 ± 0.07 | 71.88 ± 0.27 | 76.99 ± 0.38 | 43.30 ± 0.30 | 42.83 ± 0.04 | 45.95 ± 0.00 | 47.06 ± 0.00 | 48.11 ± 3.15 |
| GCNII | 78.58 ± 0.00 | 61.66 ± 0.67 | 75.41 ± 0.00 | 31.22 ± 0.00 | 40.10 ± 0.28 | 63.24 ± 1.34 | 60.78 ± 0.00 | 38.38 ± 1.08 |
| $w$GCN | 80.97 ± 0.28 | 66.21 ± 0.63 | 76.35 ± 0.38 | 43.78 ± 0.23 | 42.65 ± 0.20 | 49.73 ± 2.16 | 58.82 ± 0.00 | 43.24 ± 0.00 |
| BatchNorm | 77.90 ± 0.00 | 60.85 ± 0.09 | 77.15 ± 0.09 | 44.22 ± 0.11 | 39.68 ± 0.01 | 39.73 ± 1.24 | 52.94 ± 0.00 | 46.49 ± 1.08 |
| PairNorm | 80.30 ± 0.05 | 70.83 ± 0.06 | 77.69 ± 0.26 | 46.21 ± 0.09 | 42.30 ± 0.05 | 51.35 ± 0.00 | 58.82 ± 0.00 | 51.35 ± 0.00 |
| ContraNorm | 81.60 ± 0.00 | 72.25 ± 0.08 | 79.30 ± 0.10 | 48.63 ± 0.16 | 42.98 ± 0.04 | 48.38 ± 4.43 | 49.61 ± 1.53 | 48.63 ± 0.16 |
| DropEdge | 73.58 ± 2.76 | 65.63 ± 1.76 | 74.64 ± 1.37 | 42.30 ± 0.62 | 42.30 ± 0.09 | 59.46 ± 8.11 | 52.55 ± 4.45 | 45.95 ± 7.05 |
| Residual | 77.81 ± 0.03 | 71.61 ± 0.17 | 77.40 ± 0.06 | 43.63 ± 0.34 | 42.69 ± 0.03 | 65.95 ± 1.32 | 63.73 ± 0.98 | 61.89 ± 3.91 |
| APPNP | 77.78 ± 0.93 | 67.42 ± 1.31 | 74.52 ± 0.49 | 42.15 ± 0.17 | 42.47 ± 0.03 | 68.38 ± 4.37 | 65.10 ± 1.71 | 64.59 ± 3.30 |
| JKNET | 78.20 ± 0.20 | 66.80 ± 0.33 | 75.62 ± 0.37 | 48.16 ± 0.25 | 42.21 ± 0.05 | 60.00 ± 2.36 | 42.55 ± 2.92 | 39.73 ± 2.72 |
| DAGNN | 65.98 ± 1.49 | 60.04 ± 1.98 | 72.39 ± 0.90 | 33.39 ± 0.19 | 40.61 ± 0.03 | 61.35 ± 1.73 | 57.45 ± 1.97 | 44.87 ± 3.24 |
| Feature-SBP | 82.46 ± 0.07 | 70.63 ± 0.52 | 77.41 ± 0.21 | 49.16 ± 0.19 | 42.31 ± 0.03 | 78.38 ± 0.00 | 80.39 ± 0.00 | 72.97 ± 0.00 |
| Label-SBP | 82.90 ± 0.00 | 73.04 ± 0.10 | 80.32 ± 0.04 | 45.60 ± 0.11 | 42.41 ± 0.02 | 78.38 ± 0.00 | 80.39 ± 0.00 | 70.27 ± 0.00 |

between each pair of nodes. So we propose a modified version on the `SBP`. For Label-`SBP`, we only remove the edge when the pair of nodes are from different classes, which would even increase the sparsity of the graph while maintaining the structural balance property. For Feature-`SBP`, we reorder the matrix multiplication from $HH^T \in \mathbb{R}^{n \times n}$ to $H^T H \in \mathbb{R}^{d \times d}$ to preserve the distinctiveness of node representations across the feature dimension, rather than across the node dimension as through the original node-level repulsion. More implementation and time complexity analysis are provided in the Appendix K.

**Simulation Results** To verify our theory, we conduct simulation experiments on the Contextual Stochastic Block Model (CSBM). We set the two class means $u_1 = -1$ and $u_2 = 1$ respectively, the number of nodes $N = 100$, intra-class edge probability $p = 2 \log 100 / 100$ and inter-class edge probability $q = \log 100 / 100$. Table 2 shows the simulation results for binary-class cases with 50 nodes for each class from CSBM. We measure the $\mathcal{SID}$ of previous anti-oversmoothing methods which show that previous methods either remain a high $\mathcal{SID}$ or an imbalance $\mathcal{P}$ and $\mathcal{N}$, but our methods can effectively decrease the $\mathcal{SID}$ to achieve a more structural balance graph, or at least be on par with previous methods. Figure 2 presents the visualization of node features using `SBP`, showing that our methods effectively repel nodes from different classes, achieving a high accuracy of 97.5% even with 300 layers. We discuss more observations of $\mathcal{SID}$ in Appendix H.4.

## 5 EXPERIMENTS

In this section, we conduct a comprehensive evaluation of `SBP` on various benchmark datasets, including both homophilic and heterophilic graphs. We aim to answer three key research questions: (**RQ1**) How does `SBP` perform in node classification tasks? (**RQ2**) How effectively does `SBP` mitigate oversmoothing? (**RQ3**) How sensitive, robust, and scalable is `SBP`? More comprehensive experiments are provided in Appendix L.

**Dataset** We use eight widely-used node classification benchmark datasets (Table 8), where four of them are heterophilic (Texas, Wisconsin, Cornell, Squirrel, and Amazon-rating (Platonov et al., 2023)), and the remaining four are homophilic (Cora (McCallum et al., 2000), Citeseer (Giles et al., 1998), and Pubmed (Canese & Weis, 2013)) including one large graph dataset (Ogbn-Arxiv). In line with prior research, we employ the default training/validation/test splits provided by Pytorch Geometric (PyG).

**Baselines** We compare the performance of `SBP` against the following ten baseline models. 1) **Classic models**: MLP, SGC (Wu et al., 2019) and GCN (Kipf & Welling, 2017). 2) **GNNs with normalization**: BatchNorm (Ioffe & Szegedy, 2015), PairNorm (Zhao & Akoglu, 2019) and ContraNorm (Guo et al., 2023). 3) **Augmentation-based GNNs**: DropEdge (Rong et al., 2019). 4) **GNNs with residual connections**: Residual, APPNP (Gasteiger et al., 2018), JKNET (Xu et al., 2018) and DAGNN (Liu et al., 2020). We implement all of the methods on the SGC backbone and choose the best of scale controller $\alpha$ in range of $\{0.1, 0.5, 0.9\}$ for ContraNorm, DropEdge, and GNNs with residual connections. For both Label-`SBP` and Feature-`SBP`, we choose the best of $\lambda$ in the range of

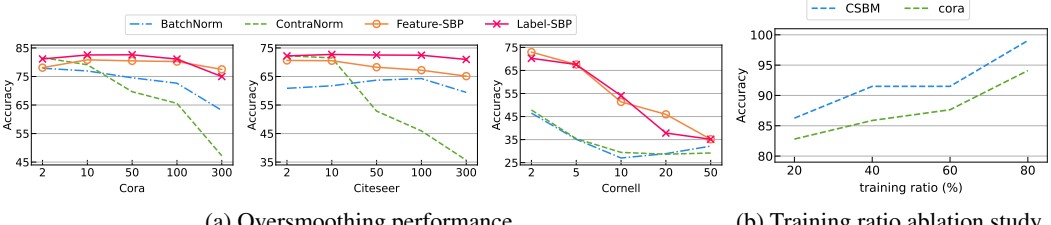

      (a) Oversmoothing performance.            (b) Training ratio ablation study.

Figure 3: Left is the performance comparison of SBP against Normalization GNNs under various model depths where the X-axis has the number of layers, and the Y-axis has node classification accuracy. Right is the ablation study on Label-SBP where the X-axis indicates the ratio of the training node numbers.

$\{0.1, 0.5, 0.9\}$. Although both positive and negative graphs have weights, we do not carefully adjust them. Instead, we fix $\alpha = 1$ and only select the best value for $\beta$.

## 5.1 RQ1: NODE CLASSIFICATION PERFORMANCE RESULTS

In Table 3, we provide the mean of the node classification accuracy along with their corresponding standard deviations across 10 random seeds following Wang et al. (2021). We keep $\beta = 1$, run all methods on 2 layers, and imply SBP on SGC. In summary, SBP achieves the best performance across eight datasets under 14 methods based on the results in Table 8 where our Label/Feature-SBP performs best on 7 out of the 8 datasets. In particular, we make the following three observations: First, SBP outperforms all normalization methods. Since our theoretical findings suggest that these normalization methods are essentially implicit signed graph propagation, the theoretical properties of structural balance (Section 4.1) contribute to the enhanced classification accuracy of SBP. Second, SBP outperforms random argumentation based GNNs. Since DropEdge randomly drops edges, it isn't easy to characterize their exact behaviors, but we highlight that it works when it happens to remove edges between different classes of nodes thanks to our structurally balanced theory, as DropEdge still follows the unified signed graph analysis in its message-passing scheme. Lastly, SBP outperforms residual connection based GNNs, including the last layer connection: residual and multilayer feature connection: APPNP, JKNET, and DAGNN. In our analysis, GNNs with residual connections can be seen as a special case of signed graph propagation, where their positive and negative adjacency matrices are the linear combination of adjacency matrices of different orders, yet they are not the theoretically best solution to alleviate oversmoothing. This validates the effectiveness of our novel insight from a signed graph perspective.

## 5.2 RQ2: ANTI-OVERSMOOTHING ANALYSIS

We further evaluate the robustness of SBP by assessing its performance at deeper model depths: $K \in \{2, 10, 50, 100, 300\}$ for homophilic datasets and $K \in \{2, 5, 10, 20, 50\}$ for heterophilic datasets. To provide a comparative analysis against other GNNs, we also evaluate two best performed normalization-based GNNs: BatchNorm and ContraNorm. We evaluate the performances of these methods on one heterophilic graph and two homophilic graphs. Figure 3a shows that the performance of Feature/Label-SBP remains relatively stable with varying numbers of layers, achieving its best performance when deeper layers ($K = 50$) are employed. In contrast, the normalization methods utilized in the experiment exhibit a substantial decrease in performance as the number of layers increases, indicating their empirical susceptibility to the oversmoothing problem. This further confirms the effectiveness of SBP at counteracting oversmoothing.

## 5.3 RQ3: ABLATION STUDY

**Training ratio sensitivity analysis.** Since Label-SBP leverages the ground truth label information to construct the negative graph, we conduct an ablation study examining the impact of different training data ratios. As shown in Figure 3b, Label-SBP's performance on the CSBM and Cora datasets improves as the training ratio increases. Even with a modest training ratio of 20%, the worst-performing models still achieve an impressive 80% accuracy, while the best models approach 100% accuracy when the training ratio is increased to 80%. This is in line with our theoretical insights that increasing training ratio leads to more structural balance resulting from our method SBP. Moreover,

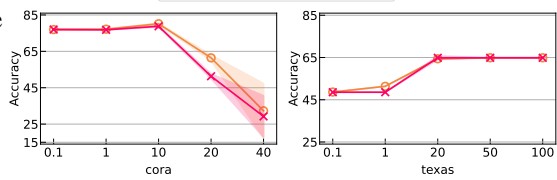

Figure 4: Figure (a)-(d) shows the effect of negative graph weight $\beta$ by SBP on CSBM. In all cases, $\lambda = 0.5$ and $\alpha = 1$. The X-axis is the $\beta$ and the Y-axis is the test accuracy. $\phi$ is the hyperparameter to control the level of homophily and $H(G)$ measure the homophily level. SBP1 indicates Label-SBP and SBP2 indicates Feature-SBP.

Table 4: Node classification accuracy (%) on the large-scale dataset *ogbn-arxiv*.

| Model | #L=2 | #L=4 | #L=8 | #L=16 |
|---|---|---|---|---|
| GCN | $67.32 \pm 0.28$ | $67.79 \pm 0.25$ | $65.54 \pm 0.31$ | $59.13 \pm 0.95$ |
| BatchNorm | $70.14 \pm 0.28$ | $70.93 \pm 0.15$ | $70.14 \pm 0.43$ | $63.24 \pm 1.40$ |
| PairNorm | $70.48 \pm 0.20$ | $71.59 \pm 0.17$ | $71.24 \pm 0.07$ | $68.92 \pm 0.43$ |
| ContraNorm | OOM | OOM | OOM | OOM |
| DropEdge | $64.07 \pm 0.32$ | $63.92 \pm 0.27$ | $60.74 \pm 0.45$ | $52.52 \pm 0.34$ |
| Residual | $66.90 \pm 0.14$ | $66.67 \pm 0.25$ | $61.76 \pm 0.62$ | $53.25 \pm 0.75$ |
| Feature-SBP | $67.89 \pm 0.10$ | $68.47 \pm 0.26$ | $65.09 \pm 0.30$ | $60.34 \pm 0.94$ |
| Label-SBP | $70.55 \pm 0.22$ | $71.54 \pm 0.18$ | $71.07 \pm 0.28$ | $69.33 \pm 0.59$ |

Figure 5: Significance of negative graph weight $\beta$ on Cora and Texas datasets where we fix the positive graph weight $\alpha = 1$.

our main experiments detailed in Table 3 demonstrate that Label-SBP outperforms other methods, even when adopting the default training set ratios in those datasets, indicating its effectiveness in real-world graph settings.

**Attraction and repulsion hyparameters sensitivity analysis.** In order to test the ability of SBP on graphs with arbitrary levels of homophily and heterophily, we conduct anablation study on the synthetic CSBM following (Chien et al., 2020). The parameter $\phi$ in the CSBM controls the relative importance of node features and graph topology in determining the homophily level. Specifically, $\phi$ ranges from -1 to 1, with lower values corresponding to strongly heterophilic graphs and higher values indicating strongly homophilic graphs. We fix $\lambda = 0.5$ and then vary $\beta$ which indicates the strength of the repulsive force between the two nodes introduced by the negative edge connecting them. As shown in Figure 4, Feature/Label-SBP performs best in homophilic graphs when all nodes are effectively attracted to one another, i.e., $\beta$ is small. As $\beta$ increases, the performance of the model degrades. In contrast, for heterophilic graphs, when the attraction power dominates, SBP achieves only 50% accuracy. In contrast, as $\beta$ increases, the negative graph becomes more dominant, and the model's performance gets significantly better. Furthermore, we observe similar phenomena in the real graph datasets shown in Figure 5.

**Large-scale dataset** Finally, we conduct an evaluation of the SBP approach on the large-scale ogbn-arxiv dataset, and the results presented in Table 4. To maintain the sparsity of the graph structure and avoid additional computational overhead, we adopt variants of the SBP approach mentioned in Section 4.3. Overall, the results demonstrate that Label-SBP achieves comparable or even superior performance compared to previous normalization methods, particularly in the deep layer ($L = 16$) setting. This verifies the empirical superiority and robustness of our proposed signed graph construction and SBP approach, which effectively leverages the available label information to alleviate the oversmoothing of the graph, even at scale.

## 6  CONCLUSION

In this work, we propose a novel unified signed graph perspective by revisiting the concept and theory of signed graphs to study the oversmoothing issue in GNNs. We find that many previous methods alleviating oversmoothing can be seen as adopting different negative graphs, and further propose two novel methods Label-SBP and Feature-SBP inspired by the structural balance theory. Our work provides new insights from signed graphs on analyzing and addressing the oversmoothing issue, which helps motivate further research in this direction.

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

# Appendix

CONTENTS

## A MORE DISCUSSION ON GNNS

### A.1 MESSAGE-PASSING GRAPH NEURAL NETWORKS (MP-GNNS)

Let $\mathcal{G} = (A, X)$ denote a graph with $n$ nodes and $m$ edges, where $A \in \{0, 1\}^{n \times n}$ is the adjacency matrix, and $X \in \mathbb{R}^{n \times d}$ is the node feature matrix with a node feature dimension of $d$. Usually, we will transform the concrete adjacency matrix $A$ to the normalized adjacency matrix $\hat{A}$ by the degree matrix. Define $D = diag(d_1, d_2, \ldots, d_n)$ where $d_i$ is the degree of the node $i$. Then the normalized adjacency matrix $\hat{A} = D^{-\frac{1}{2}} A D^{-\frac{1}{2}}$. Moreover, many theoretical works simplified the normalized adjacency matrix to be $D^{-1}A$ or $AD^{-1}$ as the raw-normalized or column-normalized stochastic matrix where the sum of every raw (column) is 1 and every entry is non-negative. In this paper, we use $\hat{A} = D^{-\frac{1}{2}} A D^{-\frac{1}{2}}$.

Different GNNs typically share a common propagation mechanism, where node features are aggregated and transformed along the network's topology to a certain depth. The $k$-th layer propagation can be formalized as

$$H_{(k)} = \textbf{PROPAGATE}(X; \mathcal{G}; k) = \left\langle \textbf{\textit{Trans}}\Big(\textbf{\textit{Agg}}\big\{\mathcal{G}; H_{(k-1)}\big\}\Big) \right\rangle_k, \tag{8}$$

with $H_{(0)} = X$ and $H_{(k)}$ is the output after the $k$-layer propagation. The notation $\langle\rangle_k$ generally varies from GNN models and denotes the generalized combination operation following $k$ convolutions. $\textbf{\textit{Agg}}\{\mathcal{G}; H_{(k-1)}\}$ refers to aggregating the $(k-1)$-layer output $\mathbf{H}^{(k-1)}$ along graph $\mathcal{G}$. Meanwhile, $\textbf{\textit{Trans}}(\cdot)$ is the corresponding layer-wise feature transformation which often includes a non-linear activation function (e.g., ReLU) and a layer-specific learnable weight matrix $W$ for transformation

### A.2 GCN

To deal with non-Euclidean graph data, GCNs are proposed for direct convolution operation over graph, and have drawn interests from various domains. GCNisfirstly introduced for a spectral perspective Kipf & Welling (2017), but soon it becomes popular as a general message-passing algorithm in the spatial domain. In the feature transformation stage, GCN adopts a non-linear activation function (e.g., ReLU) and a layer-specific learnable weight matrix $W$ for transformation. The propagation rule of GCN can formulated as follow:

$$H_{(k)} = ReLU((\hat{A}H_{(k-1)})W_{(k)}) \tag{9}$$

### A.3 SGC

SGC Wu et al. (2019) simplifies and separates the two stages of GCNs: feature propagation and (non-linear) feature transformation. It finds that utilizing only a simple logistic regression after feature propagation (removing the non-linearities), which makes it a linear GCN, can obtain comparable performance to canonical GCNs. The propagation rule of GCN can formulated as follow:

$$H_{(k)} = \hat{A}H_{(k-1)})W_{(k)} = \hat{A}^k H_{(0)})W_{(k)}...W_{(1)} \tag{10}$$

Moreover, SGC transforms $W_{(k)}...W_{(1)}$ to a general learnable parameter $W$, so the formula of SGC can be this:

$$H_{(k)} = \hat{A}^k H_{(0)})W \tag{11}$$

## B MORE BACKGROUND ABOUT SIGNED GRAPH

### B.1 SIGNED GRAPH PROPAGATION

Classical GNNs (Kipf & Welling, 2017; Wu et al., 2019; Veličković et al., 2018; Xu et al., 2019) primarily focused on message-passing on unsigned graphs or graphs composed solely of positive edges. For example, if there exists a edge $\{i, j\}$ or the sign of edge $\{i, j\}$ is positive, the node $x_i$ updates its value by:

$$\hat{x}_i = x_i + \alpha(x_j - x_i) = (1 - \alpha)x_i + \alpha x_j, \alpha \in (0, 1). \tag{12}$$

Compared to the unsigned graph, a signed graph extends the edges to either positive or negative. Notably, if the sign of edge $\{i, j\}$ is negative, the node $x_i$ update its value using the following expression:

$$\hat{x}_i = x_i - \beta(x_j - x_i) = (1 + \beta)x_i - \beta x_j, \beta \in (0, 1). \tag{13}$$

In words, the positive interaction equation 12 indicates the attraction while the negative interaction equation 13 indicates that the nodes will repel their neighbors.

More generally, when considering all of the neighbors of node $x_i$, let $N_i^+$ denote the positive neighbor set while $N_i^-$ denote the negative neighbor set, where $N_i^+ \cup N_i^- = N_i$ and $N_i^+ \cap N_i^- = \emptyset$. The representation of $x_i$ is thus updated by:

$$\hat{x}_i = (1 - \alpha + \beta)x_i + \frac{\alpha}{|N_i^+|} \sum_{j \in N_i^+} x_j - \frac{\beta}{|N_i^-|} \sum_{j \in N_i^-} x_j. \tag{14}$$

In particular, the two parameters $\alpha$ and $\beta$ mark the strength of positive and negative edges, respectively. Furthermore, the signed propagation rule equation 14 from a single node can be generalized over the whole graph $\mathcal{G}$ written in the matrix update form as:

$$\hat{X} = (1 - \alpha + \beta)X + \alpha \hat{A}^+ X - \beta \hat{A}^- X, \tag{15}$$

where $\hat{A}^+$ is the raw normalized version of the positive adjacency matrix $A^+ \in \{0, 1\}^{n \times n}$ and $\hat{A}^-$ is that of the negative adjacency matrix $A^- \in \{0, 1\}^{n \times n}$.

### B.2 DEFINITION OF NEGATIVE GRAPH

For further proofs of the theorems and propositions in the paper, we give a more simple and detailed definition in this section.

Let $D_{G^+} = \text{diag}(deg_1^+, \ldots, deg_n^+)$ and $D_{G^-} = \text{diag}(deg_1^-, \ldots, deg_n^-)$ be the degree matrices of the positive subgraph and negative subgraph, respectively. Let $A_{G^+}$ be the adjacency matrix of the graph $G^+$ with $[A_{G^+}]_{ij} = 1$ if $\{i, j\} \in E^+$ and $[A_{G^+}]_{ij} = 0$ otherwise. The adjacency matrix $A_{G^-}$ of the negative subgraph $G^-$ is defined by $[A_{G^-}]_{ij} = -1$ for $\{i, j\} \in E^-$ and $[A_{G^-}]_{ij} = 0$ for $\{i, j\} \notin E^-$.

The Laplacian plays a central role in the algebraic representation of structural properties of graphs. In the presence of negative edges, more than one definition of Laplacian is possible; see Shi et al. (2019). The Laplacian of the positive subgraph $G^+$ is $L_{G^+} := D_{G^+} - A_{G^+}$, while for the negative subgraph $G^-$ the following two variants can be used: $L_{G^-}^o := D_{G^-} - A_{G^-}$ and $L_{G^-}^r := -D_{G^-} - A_{G^-}$. Consequently, we have the following definitions.

Definition 1. Given the signed graph $G$, its opposing Laplacian is defined as

$$L_G^o := L_{G^+} + L_{G^-}^o = D_{G^+} + D_{G^-} - A_{G^+} - A_{G^-}, \tag{16}$$

and its repelling Laplacian is defined as

$$L_G^r = L_{G^+} + L_{G^-}^r = D_{G^+} - D_{G^-} - A_{G^+} - A_{G^-}. \tag{17}$$

### B.3 POSITIVE / NEGATIVE INTERACTION

Time is slotted at $t = 0, 1, \ldots$. Each node $i$ holds a state $x_i(t) \in \Omega$ at time $t$ and interacts with its neighbors at each time to revise its state. The interaction rule is specified by the sign of the links. Let $\alpha, \beta \geq 0$. We first focus on a particular link $\{i, j\} \in E$ and specify for the moment the dynamics along this link isolating all other interactions.

The DeGroot Rule:

$$x_s(t + 1) = x_s(t) + \alpha(x_{-s}(t) - x_s(t)) = (1 - \alpha)x_s(t) + \alpha x_{-s}(t), \tag{18}$$

where $-s \in \{i, j\} \setminus \{s\}$ with $\alpha \in (0, 1)$

The Opposing Rule:

$$x_s(t + 1) = x_s(t) + \beta(-x_{-s}(t) - x_s(t)) = (1 - \beta)x_s(t) - \beta x_{-s}(t); \tag{19}$$

or The Repelling Rule:

$$x_s(t + 1) = x_s(t) - \beta(x_{-s}(t) - x_s(t)) = (1 + \beta)x_s(t) - \beta x_{-s}(t). \tag{20}$$

### B.4 DETERMINISTIC NETWORKS

The Repelling Negative Dynamics:

$$
\begin{aligned}
x_i(t+1) &= x_i(t) + \alpha \sum_{j \in N_i^+} (x_j(t) - x_i(t)) - \beta \sum_{j \in N_i^-} (x_j(t) - x_i(t)) \\
&= (1 - \alpha deg_i^+ + \beta deg_i^-)x_i(t) + \alpha \sum_{j \in N_i^+} x_j(t) - \beta \sum_{j \in N_i^-} x_j(t).
\end{aligned}
\tag{21}
$$

Denote $\mathbf{x}(t) = (x_1(t) \ldots x_n(t))^T$. We can now rewrite 21 in the compact form

$$
\mathbf{x}(t+1) = M_G \mathbf{x}(t) = (I - \alpha L_{G_+} - \beta L_{G_-}^r)\mathbf{x}(t).
\tag{22}
$$

Here,
$$
M_G = I - \alpha L_{G^+} - \beta L_{G^-}^r = I - L_G^{rw},
\tag{23}
$$

with $L_G^{rw} = \alpha L_{G^+} + \beta L_{G^-}^r$ being the repelling weighted Laplacian of $G$. From Equation 22, $M_G \mathbf{1} = \mathbf{1}$ always holds. We present the following result, which by itself is merely a straightforward look into the spectrum of the repelling Laplacian $L_G^{rw}$.

Note that our Equation equation 1 is consistent with Equation equation 21, only need to replace the $\alpha$ and $\beta$ with $\frac{\alpha}{deg_i^+}$ and $\frac{\beta}{deg_i^-}$ respectively.

## C ANALYSIS OF PREVIOUS METHODS VIA SIGNED GRAPH

### C.1 DISCUSSION OF NORMALIZATION

**BatchNorm** BatchNorm centers the node representations $X$ to zero mean and unit variance and can be written as $\text{BatchNorm}(x_i) = \frac{1}{\sqrt{\sigma^2 + \epsilon}}(x_i - \frac{1}{n}\Sigma_{i=1}^n x_i)$, where $\epsilon > 0$ and $\sigma^2$ is the variance of node features. We rewrite BatchNorm in the signed graph propagation form as follows:

$$
\hat{X} = \hat{A}X\Gamma_d^{-1} - \frac{\mathbb{1}_n \mathbb{1}_n^T}{n}\hat{A}X\Gamma_d^{-1} = \hat{A}\tilde{X} - \frac{\mathbb{1}_n \mathbb{1}_n^T}{n}\hat{A}\tilde{X},
\tag{24}
$$

where $\Gamma_d = \text{diag}(\sigma_1, \ldots, \sigma_d)$ is a diagonal matrix that represents column-wise variance with $\sigma_i^2 = \frac{1}{n}\sum_{j=1}^n ((\hat{A}X)_{ji} - \mathbb{1}_n^\top \hat{A}X/n)^2$, and $\tilde{X} = X\Gamma_d^{-1}$ is a normalized version of $X$. We can correspond to the positive graph $A^+$ to $\hat{A}$ and the negative graph $A^-$ to $\frac{\mathbb{1}_n \mathbb{1}_n^T}{n}\hat{A}$ in Eq. equation 24.

**PairNorm** We then introduce another method called PairNorm where the only difference between it and BatchNorm is that PairNorm scales all the entries in $X$ using the same number rather than scaling each column by its own variance. The formulation of PairNorm can be rewritten as follows:

$$
\hat{X} = \frac{1}{\Gamma}\hat{A}X - \frac{1}{\Gamma}\frac{\mathbb{1}_n \mathbb{1}_n^T}{n}\hat{A}X = \frac{1}{\Gamma}(\hat{A}X - \frac{\mathbb{1}_n \mathbb{1}_n^T}{n}\hat{A}X),
\tag{25}
$$

where $\Gamma = \|(\hat{A} - \mathbb{1}_n \mathbb{1}_n^T/n)X\|_F/\sqrt{n}$. We observe that PairNorm shares the same positive and negative graphs (up to scale) as BatchNorm. Another normalization technique, ContraNorm, turns out to extend the negative graph to an adaptive one based on node feature similarities.

**ContraNorm** ContraNorm is inspired by the uniformity loss from contrastive learning, aiming to alleviate dimensional feature collapse. For simplicity, we consider the spectral version of ContraNorm that takes the following form:

$$
\hat{X} = (1 + \alpha)\hat{A}X - \alpha/\tau(XX^T)\hat{A}X,
\tag{26}
$$

where $\alpha \in (0, 1)$ and $\tau > 0$ are hyperparameters. We can see that $\hat{A}$ is again the positive graph and $(XX^T)\hat{A}$ is the negative graph in the corresponding signed graph propagation.

**Proposition C.1** *Consider the update:*

$$\hat{X} = AX - \frac{\mathbb{1}_n \mathbb{1}_n^T}{n} AX, \tag{27}$$

*where $A \in \{0,1\}^{n \times n}$ is the adjacency matrix. Define the overall signed graph adjacency matrix $A_s$ as $A - \frac{\mathbb{1}_n \mathbb{1}_n^T}{n} A$. Then we have that the signed graph is (weakly) structurally balanced only if the original graph can be divided into several isolated complete subgraphs.*

**Proof.** Assume that there is no isolated node and no node has edges with all the other nodes. $(A_s)_{i,j} = (A)_{i,j} - \frac{deg_j}{n}$. If $(A)_{i,j} = 1$, then we have $(A_s)_{i,j} > 0$. If $(A)_{i,j} = 0$, then we have $(A_s)_{i,j} < 0$.

If the nodes can be divided into several isolated complete subgraphs, then the nodes set $V = V_1 \cup V_2 \dots V_m$, where $|V_i| > 1$, $m$ is the number of the isolated complete subgraphs. So only the nodes within the same set have edges, thus relative entries of $A_s > 0$, while nodes from different sets do not, thus relative entries of $A_s < 0$.

On the other hand, if $A_s$ is (weakly) structurally balanced, then the nodes set can be expressed as $V = V_1 \cup V_2 \dots V_k$, where $|V_i| > 1$, $k$ is the number of the separated parties in the signed graph. The entry of $A_s$ in the same parties is positive, while between different parties is negative. According to $(A_s)_{i,j} = (A)_{i,j} - \frac{deg_j}{n}$, we know that nodes in the same parties are connected in the original graph while not connected in the original graph between different parties. So the graph can be divided into several isolated complete subgraphs.

Overall, the signed graph is (weakly) structurally balanced only if the original graph can be divided into several isolated complete subgraphs, the proof is over.

The Proposition shows that in order for the structural balance property to hold for the signed graph of normalization, the graph needs to satisfy an unrealistic condition where the edges strictly cluster the nodes.

**Discussion of ContraNorm** Consider the update:

$$\hat{X} = AX - \frac{XX^T}{n} AX, \tag{28}$$

Define the overall signed graph adjacency matrix $A_s = A - \frac{XX^T}{n} A$ where $(A_s)_{i,j} = (A)_{i,j} - \frac{1}{n} \Sigma_{k=1}^n x_i x_k^T (A)_{k,j}$ .

Assume that the nodes feature is normalized every update, that is $||x_i||_2 = 1$ for every $i$.

If $(A)_{i,j} = 1$, then we have that

$$
\begin{aligned}
(A_s)_{i,j} &= (A)_{i,j} - \frac{1}{n} \Sigma_{k=1}^n x_i x_k^T (A)_{k,j} \\
&= 1 - \frac{1}{n} \Sigma_{k=1}^n x_i x_k^T (A)_{k,j} \\
&> 1 - \frac{1}{n} \Sigma_{k=1}^n (A)_{k,j} \\
&= 1 - \frac{d_j}{n} > 0.
\end{aligned}
\tag{29}
$$

That means if $(A)_{i,j} = 1$, then $(A_s)_{i,j} > 0$. However, if $(A)_{i,j} = 0$, then we have that

$$
\begin{aligned}
(A_s)_{i,j} &= (A)_{i,j} - \frac{1}{n} \Sigma_{k=1}^n x_i x_k^T (A)_{k,j} \\
&= -\frac{1}{n} \Sigma_{k=1}^n x_i x_k^T (A)_{k,j} \\
&= -\frac{1}{n} \Sigma_{k \in N_j} x_i x_k^T.
\end{aligned}
\tag{30}
$$

Intuitively, if $x_i$ has similar features to $x_j$'s neighbors, then we have that $(A_s)_{i,j} < 0$, which means trying to repel nodes with similar representations. If $x_i$ has different features to $x_j$'s neighbors,

then we have that $(A_s)_{i,j} > 0$, which means trying to aggregate nodes with original different representations.

If graph $G$ is a completed graph, then all entries of $(A_s) > 0$, however, when all of the nodes coverage to each other, $\Sigma_{k=1}^n x_i x_k^T (A)_{k,j} = \Sigma_{k=1}^n x_i x_k^T$ will also become bigger.

## C.2 Discussion of Residual Connection

The standard residual connection (Wang et al., 2021; Chen et al., 2020a) directly combines the previous and the current layer features together. It can be formulated as:

$$\hat{X} = (1 - \alpha)X + \alpha\hat{A}X = X + \alpha\hat{A}X - \alpha I X. \tag{31}$$

For residual connections, the positive adjacency matrix is $\hat{A}$ and the negative adjacency matrix $I$ in the corresponding signed graph propagation.

**APPNP** We reformulate the method APPNP (Gasteiger et al., 2018) as the signed propagation form of the initial node feature. Another propagation process is APPNP (Gasteiger et al., 2018) which can be viewed as a layer-wise graph convolution with a residual connection to the initial transformed feature matrix $X^{(0)}$, expressed as:

$$\hat{X}^{(k+1)} = (1 - \alpha)X^{(0)} + \alpha\hat{A}X^{(k)}. \tag{32}$$

**Theorem C.2** *With $\hat{A}^+ = \Sigma_{i=0}^{k+1}\alpha^i\hat{A}^i$ and $\hat{A}^- = \alpha\Sigma_{j=0}^k\alpha^j\hat{A}^j$, the propagation process of APPNP following the signed graph propagation.*

**Proof.** Easily prove with mathematical induction.

In addition to combining with the last and initial layer features, the last type integrates several intermediate layer features. The established representations are JKNET (Xu et al., 2018) and DAGNN (Liu et al., 2020).

**JKET** JKNET is a deep graph neural network which exploits information from neighborhoods of differing locality. JKNET selectively combines aggregations from different layers with Concatenation/Max-pooling/Attention at the output, i.e., the representations "jump" to the last layer. Using attention mechanism for combination at the last layer, the $k + 1$-layer propagation result of JKNET can be written as:

$$\begin{aligned} X^{(k+1)} &= \alpha_0 X^{(0)} + \alpha_1 X^{(1)} + \cdots \alpha_k X^{(k)} \\ &= \Sigma_{i=0}^k \alpha_i \hat{A}^i X^{(0)}, \end{aligned} \tag{33}$$

where $\alpha_0, \alpha_1, \cdots, \alpha_k$ are the learnable fusion weights with $\Sigma_{i=0}^k \alpha_i = 1$.

**DAGNN** Deep Adaptive Graph Neural Networks (DAGNN) (Liu et al., 2020) tries to adaptively add all the features from the previous layer to the current layer features with the additional learnable coefficients. After decoupling representation transformation and propagation, the propagation mechanism of DAGNN is similar to that of JKNET.

$$X^{(k+1)} = \Sigma_{i=0}^k \alpha_i \hat{A}^i H^{(0)}, \ H^{(0)} = f_\theta(X^{(0)}) \tag{34}$$

$H^{(0)} = f_\theta(X^{(0)})$ ) is the non-linear feature transformation using an MLP network, which is conducted before the propagation process and $\alpha_0, \alpha_1, \cdots, \alpha_k$ are the learnable fusion weights with $\Sigma_{i=0}^k \alpha_i = 1$.

**Theorem C.3** *With $\hat{A}^+ = \Sigma_{i=0}^{k-1}\alpha^i\hat{A}^i + \hat{A}^k$ and $\hat{A}^- = \Sigma_{j=0}^{k-1}\alpha^j\hat{A}^k$, the propagation process of JKNET and DAGNN following the signed graph propagation.*

**Proof.** Easily prove with mathematical induction.

As for more residual inspired methods (Chen et al., 2020b; Eliasof et al., 2022; Luan et al., 2022; Eliasof et al., 2021), we select GCNII and wGCN to give a detailed discussion as follows.

- As for GCNII (Chen et al., 2020b), it is an improved version of APPNP with the learnable coefficients $\alpha_i$ and changes the learnable weight W to $(1 - \beta_i)I + \beta_i W$ each layer, so it shares the same positive and negative graph as APPNP.

- As for the wGCN (Eliasof et al., 2022), it incorporates trainable channel-wise weighting factors $\omega$ to learn and mix multiple smoothing and sharpening propagation operators at each layer, same as the init residual combines but change parameters $\alpha$ to be learnable with a more detailed selection strategy.

### C.3 DISCUSSION OF DROPMESSAGE

For DropMessage (Fang et al., 2022), it is a unified way of DropNode, DropEdge and Dropout but with a more flexible mask strategy. We have discussed the DropNode and DropEdge in our paper. DropMessage can be viewed as randomly dropping some dimension of the aggregated node features instead of the whole node or the whole edge. We give the unified positive and negative graph of DropMessage in the term of the signed graph. The propagation of DropMessage can be expressed as $H^{(k)} = AH^{(k-1)} - M_m$, where if dropping $AH_{ij}^{(k-1)}$, then $M_{ij} = AH_{ij}^{(k-1)}$ else $M_{ij} = 0$.

## D PROOF OF THEOREM 4.1

Now consider the combined theorem.

**Theorem D.1** *Suppose that the positive edges are connected. Then along Equation 21 for any $0 < \alpha < 1/\max_{i \in V} \deg_i^+$, there exists a critical value $\beta_* \geq 0$ for $\beta$ such that*

*(i) if $\beta < \beta_*$, then we have $\lim_{t \to \infty} x_i(t) = \sum_{j=1}^n x_j(0)/n$ for all initial values $x(0)$;*

*(ii) if $\beta > \beta_*$, then $\lim_{t \to \infty} \|x(t)\| = \infty$ for almost all initial values w.r.t. Lebesgue measure.*

**Proof.** we change the signed graph update to the equivalent version of $x_i(t)$ read as:

$$x_i(t+1) = x_i(t) + \alpha \sum_{j \in N_i^+} (x_j(t) - x_i(t)) - \beta \sum_{j \in N_i^-} (x_j(t) - x_i(t)).$$

This can be expressed as:

$$x(t+1) = (1 - \alpha \deg^+ + \beta \deg^-)x_i(t) + \alpha \sum_{j \in N^+} x_j(t) - \beta \sum_{j \in N^-} x_j(t). \tag{35}$$

Algorithm 35 can be written as:

$$x(t+1) = M_G x(t) = (I - \alpha L_G^+ - \beta L_G^-)x(t). \tag{36}$$

Here, $M_G = I - \alpha L_G^+ - \beta L_G^-$, with $L_G^+ = \alpha L_C^+ + \beta L_C^-$ being the repelling weighted Laplacian of $G$, defined in Sec.B.2. From Eq.equation 36, $M_G \mathbf{1} = \mathbf{1}$ always holds. We present the following result, which by itself is merely a straightforward look into the spectrum of the repelling Laplacian $L_G^-$.

So theorem D.1 can be changed to the following version:

Suppose $G^+$ is connected. Then along Eq.equation 36 for any $0 < \alpha < 1/\max_{i \in V} \deg_i^+$, there exists a critical value $\beta > 0$ for $\beta$ such that:

(i) if $\beta < \beta_*$, then average consensus is reached in the sense that $\lim_{t \to \infty} x_i(t) = \frac{1}{n} \sum_{j=1}^n x_j(0)$ for all initial values $x(0)$;

(ii) if $\beta > \beta_*$, then $\lim_{t \to \infty} \|x(t)\| = \infty$ for almost all initial values w.r.t. Lebesgue measure.

**Proof.** Define $J = 11^T/n$. Fix $\alpha \in (0, 1/\max_{i \in V} \deg_i^+)$ and consider $f(\beta) = \lambda_{\max}(I - \alpha L_G^+ - \beta L_G^- - J)$, and $g(\beta) = \lambda_{\min}(I - \alpha L_G^+ - \beta L_G^- - J)$. The Courant–Fischer Theorem implies that both $f(\cdot)$ and $g(\cdot)$ are continuous and nondecreasing functions over $[0, \infty)$. The matrix $J$ always

commutes with $I - \alpha L_G^+ - \beta L_G^-$, and 1 is the only nonzero eigenvalue of $J$. Moreover, the eigenvalue 1 of $J$ shares a common eigenvector $\mathbf{1}$ with the eigenvalue 1 of $I - \alpha L_G^+ - \beta L_G^-$.

Since $G^+$ is connected, the second smallest eigenvalue of $L_{G^+}$ is positive. Since $0 < \alpha < \frac{1}{\max_{i \in V} \deg_i^+}$, there holds $\lambda_{\min}(I - \alpha L_{G^+}) \geq -1$, again due to the Gershgorin Circle Theorem. Therefore, $f(0) < 1$, $g(0) \geq -1$. Noticing $f(\infty) = \infty > 1$, there exists $\beta_* > 0$ satisfying $f(\beta_*) = 1$. We can then verify the following facts:

- There hold $f(\beta) < 1$ and $g(\beta) > -1$ if $\beta < \beta_*$. In this case, along Eq.equation 36 $\lim_{t \to \infty}(I - J)x(t) = 0$, which in turn implies that $x(t)$ converges to the eigenspace corresponding to the eigenvalue 1 of $M_G$. This leads to the average consensus statement in (i).

- There holds $f(\beta) \geq 1$ if $\beta > \beta_*$. In this case, along Eq.equation 36 $x(t)$ diverges as long as the initial value $x(0)$ has a nonzero projection onto the eigenspace corresponding to $\lambda_{\max}(M_G)$ of $M_G$. This leads to the almost everywhere divergence statement in (ii).

The proof is now complete.

## E   PROOF OF THEOREM 4.3

**Theorem E.1** *let $A > 0$ be a constant and define $\mathcal{F}(z)_c$ by $\mathcal{F}(z)_c = -c, z < -c$, $\mathcal{F}(z)_c = z, z \in [-c, c]$, and $\mathcal{F}(z)_c = c, z > c$. Define the function $\theta : E \to \mathbb{R}$ so that $\theta(\{i, j\}) = \alpha$ if $\{i, j\} \in E^+$ and $\theta(\{i, j\}) = -\beta$ if $\{i, j\} \in E^-$. Assume that node $i$ interacts with node $j$ at time $t$ and consider the following node interaction under the signed propagation rules:*

$$x_s(t+1) = \mathcal{F}(z)_c((1 - \theta)x_s(t) + \theta x_{-s}(t)), \ s \in \{i, j\}. \tag{37}$$

*let $\alpha \in (0, 1/2)$. Assume that $G$ is a structurally balanced complete graph under the partition $V = V_1 \cup V_2$. When $\beta$ is sufficiently large, for almost all initial values $x(0)$ w.r.t. Lebesgue measure, there exists a binary random variable $l(x(0))$ taking values in $\{-c, c\}$ such that*

$$\mathbb{P}\left(\lim_{t \to \infty} x_i(t) = l(x(0)), i \in V_1; \lim_{t \to \infty} x_i(t) = -l(x(0)), i \in V_2\right) = 1. \tag{38}$$

**Proof.**   The proof is based on the following lemmas.

**Lemma E.2** *Fix $\alpha \in (0, 1)$ with $\alpha \neq \frac{1}{2}$. For the dynamics 37 with the random pair selection process, there exists $\beta^*(\alpha) > 0$ such that*

$$\mathbb{P}\left(\limsup_{t \to \infty} \max_{i,j \in V} |x_i(t) - x_j(t)| = 2A\right) = 1$$

*for almost all initial beliefs if $\beta > \beta^*$.*

**Lemma E.3** *Fix $\alpha \in (1/2, 1)$ and $\beta \geq 2/(2\alpha - 1)$. Consider the dynamics 37 with the random pair selection process. Let $G$ be the complete graph with $\kappa(G^+) \geq 2$. Suppose for time $t$ there are $i_1, j_1 \in V$ with $x_{i_1}(t) = -c$ and $x_{j_1}(t) = c$. Then for any $\epsilon \in [0, (2\alpha - 1)c/2\alpha]$ and any $i_* \in V$, the following statements hold:*

- (i) *There exist an integer $Z(\epsilon)$ and a sequence of node pair realizations, $G_{t+s}(\omega)$, for $s = 0, 1, \ldots, Z - 1$, under which $x_{i_*}(t + Z)(\omega) \leq -A + \epsilon$.*

- (ii) *There exist an integer $Z(\epsilon)$ and a sequence of node pair realizations, $G_{t+s}(\omega)$, for $s = 0, 1, \ldots, Z - 1$, under which $x_{i_*}(t + Z)(\omega) \geq A - \epsilon$.*

**Proof.**   From our standing assumption, the negative graph $G^-$ contains at least one edge. Let $k_*, m_* \in V$ share a negative link. We assume the two nodes $i_1, j_1 \in V$ labeled in the lemma are different from $k_*, m_*$, for ease of presentation. We can then analyze all possible sign patterns among the four nodes $i_1, j_1, k_*, m_*$. We present here just the analysis for the case with

$$\{i_1, k_*\} \in E^+, \{i_1, m_*\} \in E^+, \{j_1, k_*\} \in E^+, \{j_1, m_*\} \in E^+.$$

The other cases are indeed simpler and can be studied via similar techniques.

Without loss of generality we let $x_{m_*}(t) \geq x_{k_*}(t)$. First of all we select $G_t = \{i_1, k_*\}$ and $G_{t+1} = \{j_1, m_*\}$. It is then straightforward to verify that

$$x_{m_*}(t+2) \geq x_{k_*}(t+2) + 2\alpha c.$$

By selecting $G_{t+2} = \{m_*, k_*\}$ we know from $\beta \geq \frac{2}{(2\alpha-1)} > \frac{1}{\alpha}$ that

$$x_{k_*}(t+3) = -c, \quad x_{m_*}(t+3) = c.$$

There will be two cases:

(a) Let $i_* \in \{m_*, k_*\}$. Noting that $\kappa(G^+) \geq 2$, there will be a path connecting to $k_*$ from $i_*$ without passing through $m_*$ in $G^+$. It is then obvious that we can select a finite number $Z_1$ of links which alternate between $\{m_*, k_*\}$ and the edges over that path so that $x_{i_*}(t+3+Z_1) \geq -c + \epsilon$. Here $Z_1$ depends only on $\alpha$ and $n$.

(b) Let $i_* \in \{m_*, k_*\}$. We only need to show that we can select pair realizations so that $x_{m_*}$ can get close to $-c$, and $x_{k_*}$ gets close to $c$ after $t+3$. Since $G^+$ is connected, either $m_*$ or $k_*$ has at least one positive neighbor. For the moment assume $m'$ is a positive neighbor of $m_*$ and $k'$ is a positive neighbor of $k_*$ with $m' \neq k'$. Then from part (a) we can select $Z_2$ node pairs so that

$$x_{m_*}(t+3+Z_2) \leq -c + \epsilon, \quad x_{k_*}(t+3+Z_2) \geq c - \epsilon.$$

Thus, selecting the negative edge $\{m_*, k_*\}$ for $t+5+Z_2$ implies $x_{m_*}(t+6+Z_2) = c$ for $\beta \geq \frac{2}{(2\alpha-1)}$. The case with $m' = k'$ can be dealt with by a similar treatment, leading to the same conclusion.

This concludes the proof of the lemma.

In view of Lemmas E.2 and E.3, the desired theorem is a consequence of the second Borel–Cantelli Lemma.

## F  THE RELATIONSHIP OF OVERSMOOTHING AND THEOREM 4.1 AND THEOREM 4.3

**Discussion with training methods**   While Peng et al. (2024) questions the existence of oversmoothing in trained GNNs, their observations are primarily based on specific experimental settings that may not fully capture the oversmoothing challenge present in the literature. Specifically, the empirical observations in Peng et al. (2024) are based on 10-layer GCNs, which, while useful for their argument, may not represent the behavior of deeper networks or other GNN architectures. Moreover, Figure 2 in Peng et al. (2024) is based on a normalized metric, which might not be the most appropriate. To see this point, suppose one wants to classify two points. In one case, we have 0.01 vs -0.01 and in the other case, we have 100 vs -100. While the normalized distance considered in Peng et al. (2024) would be the same for these two cases, the latter case has a much larger margin, and it would be thus much easier to learn a classifier. On the other hand, Cong et al. (2021) suggests that the trainability of deep GNNs is more of a problem than over-smoothing. However, over-smoothing naturally presents challenges for training deep GNNs, as when oversmoothing happens, gradients vanish across different nodes. Besides, Cong et al. (2021)compares 3 models GCN, ResGCN and GCNII, proving that GCNII is the best backbone. We have adapted our SBP to GCNII in Table 13 and the results showed that our SBP outperforms GCNII on average, especially in the middle layers.

**Measure on oversmoothing**   There exist a variety of different approaches to quantify oversmoothing in deep GNNs, here we choose the measure based on the Dirichlet energy on graphs (Wu et al., 2023a; Rusch et al., 2023).

$$\epsilon(X(t)) = \frac{1}{v}\Sigma_{i \in V}\Sigma_{j \in N_i}||x_i(t) - x_j(t)||_2^2, \tag{39}$$

where $v$ is the number of the nodes, $x_i(t)$ is the node feature of node $i$ at time $t$. $N_i$ represents the neighbor set of node $i$, In the signed graph, it including nodes connected to $i$ by both positive and negative edges. Oversmoothing means that when the layers are infinity, all of the node features will converge, that is to say $\lim_{t \to \infty} \epsilon(X(t)) \to 0$.

In Theorem 4.1, there are 2 cases:

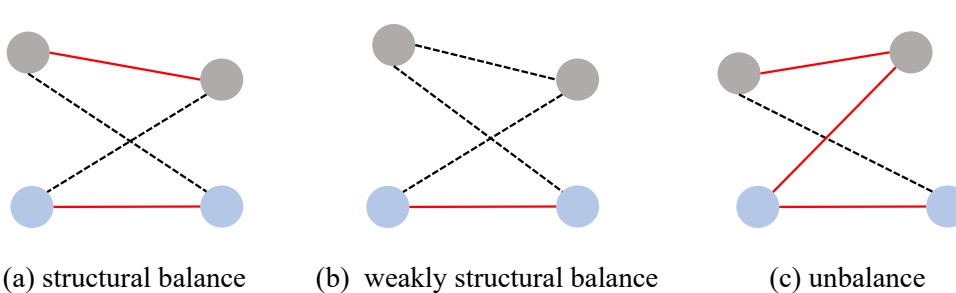

Figure 6: Examples of structural balanced (left), weakly structural balanced (middle), and unbalanced signed graphs (right). Here red lines represent positive edges; black dashed lines represent negative edges; gray and blue circles represent nodes from different labels

- $if \beta < \beta_*$, then we have $\lim_{t\to\infty} x_i(t) = \sum_{j=1}^{n} x_j(0)/n$ for all initial values $x(0)$
- $if \beta > \beta_*$, then $\lim_{t\to\infty} \|x(t)\| = \infty$ for almost all initial values w.r.t. Lebesgue measure.

In the first case, all the node features will coverage to the mean of them and therefore $\lim_{t\to\infty} \epsilon(X(t)) \to 0$, then oversmoothing happens. In the second case, the node features will diverge to infinity and thus $\lim_{t\to\infty} \epsilon(X(t)) \to 0$ or $\infty$ which is also not what we want.

Theorem 4.1 demonstrated that both insufficient repulsion and excessive repulsion caused by the negative graph can hinder performance in signed graph propagation. From this, we conclude that relying solely on the negative signs is insufficient to alleviate oversmoothing. Therefore, we propose the provable solution: a structurally balanced graph to efficiently alleviate oversmoothing in Theorem 4.3. Specifically, we have the following conclusion from the structurally balanced graph in Theorem 4.3:

$$\mathbb{P}\left(\lim_{t\to\infty} x_i(t) = l(x(0)), i \in V_1; \lim_{t\to\infty} x_i(t) = -l(x(0)), i \in V_2\right) = 1. \tag{40}$$

Then we have:

$$\lim_{t\to\infty} \epsilon(X(t)) = \lim_{t\to\infty} \frac{1}{v} \Sigma_{i\in V} \Sigma_{j\in N_i} \|x_i(t) - x_j(t)\|_2^2 \tag{41}$$

$$= \lim_{t\to\infty} \frac{1}{v} \Sigma_{i\in V_1} \Sigma_{j\in N_i} \|x_i(t) - x_j(t)\|_2^2 + \frac{1}{v} \Sigma_{i\in V_2} \Sigma_{j\in N_i} \|x_i(t) - x_j(t)\|_2^2 \tag{42}$$

$$= \lim_{t\to\infty} \frac{1}{v} \Sigma_{i\in V_1} \Sigma_{j\in N_i, y_i \neq y_j} \|x_i(t) - x_j(t)\|_2^2 + \frac{1}{v} \Sigma_{i\in V_2} \Sigma_{j\in N_i, y_i \neq y_j} \|x_i(t) - x_j(t)\|_2^2 \tag{43}$$

$$= \lim_{t\to\infty} \frac{1}{v} \Sigma_{i\in V_1} \frac{v}{2} \times 2c + \frac{1}{v} \Sigma_{i\in V_2} \frac{v}{2} \times 2c \tag{44}$$

$$= \lim_{t\to\infty} \frac{1}{v} (\frac{v}{2} \times \frac{v}{2} \times 2c + \frac{v}{2} \times \frac{v}{2} \times 2c) \tag{45}$$

$$= vc \geq 0 \tag{46}$$

So Theorem 4.3 proves that under certain conditions, structural balance can alleviate oversmoothing even when the layers are infinity.

## G    EXTENSION OF STRUCTURAL BALANCE

To clarify the concept of structural balance, weakly structural balance and unbalance signed graph, we give the examples as shown in Figure 6. The notion of structural balance can be weakened in the following definition G.1.

**Definition G.1** *A signed graph $G$ is **weakly structurally balanced** if there is a partition of $V$ into $V = V_1 \cup V_2 \cup \ldots \cup V_m$, $m \geq 2$ with $V_1, \ldots, V_m$ being nonempty and mutually disjoint, where any edge between different $V_i$'s is negative, and any edge within each $V_i$ is positive.*

Then we show that when $\mathcal{G}$ is a complete graph, weak structural balance also leads to clustering of node states.

**Theorem G.2 (Shi et al. (2019), Theorem 10)** *Assume that node $i$ interacts with node $j$ and $x_i(t)$ represents the value of node $i$ at time $t$. Let $\theta = \alpha$ if the edge $\{i, j\}$ is positive and $\theta = \beta$ if the edge $\{i, j\}$ is negative. Consider the constrained signed propagation update:*

$$x_i(t+1) = \mathcal{F}_c((1-\theta)x_i(t) + \theta x_J(t)). \tag{47}$$

*Let $\alpha \in (0, 1/2)$. Assume that $\mathcal{G}$ is a weakly structurally balanced complete graph under the partition $V = V_1 \cup V_2 \cdots \cup V_m$. When $\beta$ is sufficiently large, for almost all initial values $x(0)$ w.r.t. Lebesgue measure, there exists $m$ random variable $l_1(x(0))$, $l_2(x(0))$, $\ldots$, $l_m(x(0))$, each of which taking values in $\{-c, c\}$ such that*

$$\mathbb{P}\left(\lim_{t \to \infty} x_i(t) = l_j(x(0)), i \in V_j, j = 1, \ldots, m\right) = 1. \tag{48}$$

## H    Discussion about $\mathcal{SID}$

We give the details of CSBM and a more clear formula of $\mathcal{SID}$, $\mathcal{P}$ and $\mathcal{N}$ as suggested in Tabel 2 in this section.

### H.1    Definition of CSBM

To quantify the structural balance of the mentioned methods, we simplified the graph to 2-CSBM$(N, p, q, \mu_1, \mu_2, \sigma^2)$ following Wu et al. (2023b). It consists of two classes $\mathcal{C}_1$ and $\mathcal{C}_2$ of nodes of equal size, in total with $N$ nodes. For any two nodes in the graph, if they are from the same class, they are connected by an edge independently with probability $p$, or if they are from different classes, the probability is $q$. For each node $v \in \mathcal{C}_i, i \in \{1, -1\}$, the initial feature $X_v$ is sampled independently from a Gaussian distribution $\mathcal{N}(\mu_i, \sigma^2)$, where $\mu_i = \mathcal{C}_i, \sigma = I$. In this paper, we assign $N = 100$ and the feature dimension is 8.

### H.2    Measure of $\mathcal{SID}$

$$\mathcal{P} = \frac{1}{|V|} \sum_{v \in V} \text{Number of nodes who have the same label as } v \text{ and the non-positive edge.} \tag{49}$$

$$\mathcal{N} = \frac{1}{|V|} \sum_{v \in V} \text{Number of nodes who have the different label from } v \text{ and the non-negative edge.}$$

$$\tag{50}$$

$$\mathcal{SB} = \frac{1}{2}(\mathcal{P} + \mathcal{N}) \tag{51}$$

### H.3    Proof of Proposition 4.6

**Proposition H.1** *For a structural balanced complete graph $\mathcal{G}$, we have $\mathcal{SID}(\mathcal{G}) = 0$.*

**Proof**    To better understand, we give an example of the structural balance graph as shown in Figure 7. we can see that for a node $v$, $\mathcal{P}(v) = 0$ and $\mathcal{N}(v) = 0$ due to the structural balance complete graph assumption. So $\mathcal{SID}(\mathcal{G}) = 0$.

### H.4    More observations of $\mathcal{SID}$

Apart from Table 2 on CSBM, we further present the Structural Imbalance Degree ($\mathcal{SID}$) for Cora across different methods in Table 5. As the performance of these methods is similar in shallow layers (2), we focus on layer 16 to showcase the results.

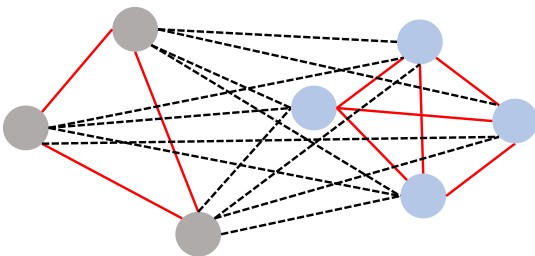

structural balance complete graph

Figure 7: Example of structural complete graph. Here red lines represent positive edges; black dashed lines represent negative edges; gray and blue circles represent nodes from different labels

Table 5: $\mathcal{SID}$ on Cora datasets. We implement all of the methods on SGC under 100 epochs and the accuracy is the result.

| | label-SBP | feature-SBP | BatchNorm | ContraNorm | Residual | DropEdge |
|---|---|---|---|---|---|---|
| $\mathcal{P}$ | 482.1123 | 482.1123 | 482.1123 | 482.1123 | 482.5137 | 484.2075 |
| $\mathcal{N}$ | 0.7408 | 0.7408 | 0.7408 | 0.7408 | 2221.7305 | 2221.7305 |
| $\mathcal{SID}$ | 241.4265 | 241.4265 | 241.4265 | 241.4265 | 1352.1221 | 1352.9620 |
| Accuracy | $77.43 \pm 1.49$ | $77.22 \pm 0.55$ | $70.79 \pm 0.00$ | $63.35 \pm 0.00$ | $40.91 \pm 0.00$ | $22.24 \pm 3.04$ |

We have two key observations: 1) Methods with higher $\mathcal{SID}$ generally lead to worse accuracy, while those with lower $\mathcal{SID}$ tend to produce better accuracy. 2) $\mathcal{SID}$ is a coarse-grained metric; different methods can yield the same $\mathcal{SID}$ values while their performance varies. These observations can also align with the experiments in cSBM in Table 2. The observation may stem from the fact that structural balance is an inherent property of graph structure, which is challenging to measure precisely using a numerical metric like $\mathcal{SID}$. Proposition 4.6 in the paper proves that when $\mathcal{SID} = 0$, the graph is structurally balanced. However, for graphs that are not structurally balanced, the properties remain unclear. For future work, we aim to develop a more nuanced metric to quantify the structural balance property of graphs.

# I   PROOF OF PROPOSITION 4.7

**Proposition I.1** *Assume that node label classes are balanced $|Y_1| = |Y_2|$ and denote the ratio of labelled nodes as $p$. Then we have that the signed graph adjacency matrix $A_s = A - A_l$ and $\mathcal{SID}(\mathcal{G}) \leq (1-p)\frac{n}{2}$, where $\mathcal{SID}$ decreases with a larger labelling ratio $p$. In particular, when $p = 1$ (full supervision), we have $\mathcal{SID}(\mathcal{G}) = 0$, i.e., a perfectly balanced graph. Under the constrained signed propagation equation 3, the nodes from different classes will converge to distinct constants.*

**Proof.**   Without loss of generality, assume that the node feature has been normalized which means that $\|x_i\|_2 = 1$ for every $i$. If $x_i$ and $x_j$ has the same label, then we have that, $(A_s)_{i,j} = (A)_{i,j} + 1 > 1$. If $x_i$ and $x_j$ has different labels, then we have that $(A_s)_{i,j} = (A)_{i,j} - 1 \leq 0$.

We first prove that $\mathcal{SID}(\mathcal{G}, p) \leq (1-p)\frac{n}{2}$ where $n$ is the nodes number and $p$ is the label ratio. We have that

$$\mathcal{P}(v) + \mathcal{N}(v) \leq (1-p)n, \tag{52}$$

because for a single node $v$ only the remaining $(1-p)n$ nodes' labels are unknown and therefore their edges may need to change so that

$$\mathcal{SID}(\mathcal{G}) = \frac{1}{2n} \sum_{v \in \mathcal{G}} (|\mathcal{P}(v)| + |\mathcal{N}(v)|)$$

$$\leq \frac{1}{2n} \sum_{v \in \mathcal{G}} (1-p)n \tag{53}$$

$$= (1-p)\frac{n}{2}.$$

We know that when $\mathcal{SID}(\mathcal{G}) = 0$, then we have that the nodes $V$ set can be divided into $V_1 \cup V_1 \cdots \cup V_L$ where $L$ is the number of the node classes. There are only positive edges with the node subset and only negative edges between the node subset.

Since $C = 2$, the node set can be divided into $V_1$ and $V_2$, the signed graph is structurally balanced. According to Theorem 4.3, we have that the nodes in $V_1$ will converge to the $c$ where $||c||_2 = 1$ and the nodes in $V_2$ will converge to $-c$. Thus under Label-SBP propagation, the oversmoothing will only happen within the same label and repel different labels to the boundary.

## J   MORE DISCUSSION ON STRUCTURAL BALANCED PROPAGATION

The overall update of **S**tructural **B**alanced **P**ropagation is as following:
$$X^{(k)} = \text{Layernorm}\{(1-\lambda)X^{(k-1)} + \lambda(\alpha A^+ X^{(k-1)} - \beta A^- X^{(k-1)})\}, \tag{54}$$

Our methods adopt the normalized adjacency matrix as the positive graph $A^+ = \hat{A}$, while use different negative graphs. Although both the positive and negative graphs have hyperparameters, we do not carefully adjust the hyperparameters. Instead, we fix $\alpha = 1$ and only select the best value for $\beta$. You can also change $\alpha$ and $\beta$ together to achieve the best performance.

**Label-Induced Negative Graph**   The negative graph for Label-SBP is:
$$A^-_{ij} = \begin{cases} 1 & \text{if i,j has the different labels,} \\ -1 & \text{if i,j has the same labels,} \\ 0 & \text{if i,j has the unknown labels.} \end{cases} \tag{55}$$

For practice, we apply softmax to it:
$$\tilde{A}^- = \text{softmax}(A^-). \tag{56}$$

Applying softmax makes the negative graph the row-stochastic which is a non-negative matrix with row sum equal to one. We also tried the normalization method, which is not as good as the softmax. This may be because the softmax method is more aligned with the row-stochastic adjacency, where every element is non-negative.

**Similarity-Induced Negative Graph**   The negative graph for Feature-SBP is:
$$A^- = -X^{(0)} X^{(0)T} \tag{57}$$

We also attempted using the last layer node features for the negative graph, but they are not as effective as the initial layer node features. This may be due to oversmoothing as the layers go deeper. For practice, we apply softmax as the Label-SBP to it:
$$\tilde{A}^- = \text{softmax}(-XX^T) \tag{58}$$

## K   TIME COMPLEXITY ANALYSIS AND THE MODIFIED SBP

**Label-SBP**   As shown in equation 7, we maintain the positive adjacency matrix $A^+ = \hat{A}$ and construct the negative adjacency matrix $A_l$ by assigning 1 when nodes $i, j$ have different labels, -1 when they share the same label, and 0 when either label is unknown. We then apply softmax to $A_l$ to normalize the negative adjacency matrix. The overall signed adjacency matrix is $A_{sign} = \alpha A^+ - \beta softmax(A_l)$, where $\alpha$ and $\beta$ are hyperparameters. Given $n_t$ training nodes and $d$ edges in the graph, our Label-SBP increases the edge count from $O(d)$ to $O(n_t^2)$, thereby increasing the computational complexity to $O(n_t^2 d)$.

**Feature-SBP**   When labels are unavailable, we propose Feature-SBP, which uses the similarity matrix of node features to create the negative adjacency matrix. As depicted in equation 7, we design the negative adjacency matrix as $A_f = -X_0 X_0^T$. We then apply softmax to $A_f$ to normalize it. The overall matrix follows the same format as Label-SBP: $A_{sign} = \alpha A^+ - \beta softmax(A_f)$, where $\alpha$ and $\beta$ are hyperparameters. The additional computational complexity primarily stems from the negative graph propagation, which involves $X_0 X_0^T \in \mathbb{R}^{n \times n}$, increasing the overall complexity to $O(n^2 d)$.

We show the computation time of different methods in the Table 6. On average, we improve performance on 8 out of 9 datasets (as shown in Table 3) with less than 0.05s overhead—even faster than three other baselines. We believe this time overhead is acceptable given the benefits it provides.

Table 6: Estimated training time of `SBP` on Cora dataset. All experiments are run under 2 layers. s is the abbreviation for second. Precompute time is the aggregation time across layers, train time is the update time of the SGC weight $W$, total time is the sum of them.

| | Label-`SBP` | Feature-`SBP` | BatchNorm | ContraNorm | Residual | JKNET | DAGNN | SGC |
|---|---|---|---|---|---|---|---|---|
| Precompute time | 0.1809s | 0.1520s | 0.1860s | 0.1888s | 0.0604s | 0.0577s | 0.1438s | 0.1307s |
| Train time | 0.1071s | 0.1060s | 0.1076s | 0.1038s | 0.1368s | 0.1446s | 0.1348s | 0.1034s |
| Total time | 0.2879s | 0.2580s | 0.2935s | 0.2926s | 0.1972s | 0.2023s | 0.2786s | 0.2341s |
| Rank | 6 | 4 | 8 | 7 | 1 | 2 | 5 | 3 |

**Scalability of `SBP` on large-scale graph** For large-scale graphs, we introduce a modified version by only removing edges when pairs of nodes belong to different classes. This approach allows our modified Label-`SBP` to eliminate the computational overhead of the negative graph, further enhancing the sparsity of large-scale graphs. For Feature-`SBP`, as the number of nodes $n$ increases, the complexity of this matrix operation grows quadratically, i.e., $o(n^2d)$. To address this, we reorder the matrix multiplication from $-X_0 X_0^T \in \mathbb{R}^{n \times n}$ to $-X_0^T X_0 \in \mathbb{R}^{d \times d}$. This preserves the distinctiveness of node representations across the feature dimension, rather than across the node dimension as in the original node-level repulsion. The modified version of Feature-`SBP` can be expressed as:

$$(\text{Feature-}\texttt{SBP}\text{-v2}) \quad X^k = (1 - \lambda)X^{(k-1)} + \lambda(\alpha \hat{A} X^{(K)} - \beta X^{(K)} \text{softmax}(-X_0^T X_0)) \quad (59)$$

This transposed alternative has a linear complexity in the number of samples, i.e., $O(nd^2)$. In cases where $n \gg d$, this modified version significantly reduces the computational burden.

We compare the compute time `SBP` with other baselines on ogbn-arxiv dataset over 100 epochs for a fair comparison. Among all the training times of the baselines, our Label-`SBP`-v2 achieves the 3rd fastest time while Feature-`SBP`-v2 ranks 5th. Therefore, we recommend using Label-`SBP`-v2 for large-scale graphs since they typically have a sufficient number of node labels. We believe that although there is a slight time increase, it is acceptable given the benefits.

Table 7: Estimated training time of `SBP` on ogbn-arixv dataset. All experiments are run under 2 layers and 100 epochs. s is the abbreviation for second.

| | Label-`SBP` | Feature-`SBP` | BatchNorm | ContraNorm | DropEdge | SGC |
|---|---|---|---|---|---|---|
| Train time (s) | 5.5850 | 6.1333 | 5.3872 | 5.8375 | 9.5727 | 5.3097 |
| Rank | 3 | 5 | 2 | 4 | 6 | 1 |

## L   DETAILS OF EXPERIMENTS

The code for the experiments will be available when our paper is acceptable. We will replace this anonymous link with a non-anonymous GitHub link after the acceptance. We implement all experiments in Python 3.9 with PyTorch Geometric on one NVIDIA Tesla V100 GPU.

### L.1   DETAILS OF THE DATASET

Table 8: Summary of datasets. $H(G)$ refers to the edge homophily level: the higher, the more homophilic the dataset is.

| Dataset | $H(G)$ | Classes | Nodes | Edges |
|---|---|---|---|---|
| **Cora** | 0.81 | 7 | 2,708 | 5,429 |
| **Citeseer** | 0.74 | 6 | 3,327 | 4,732 |
| **PubMed** | 0.80 | 3 | 19,717 | 44,338 |
| **Texas** | 0.21 | 5 | 183 | 295 |
| **Cornell** | 0.30 | 5 | 183 | 280 |
| **Amazon-ratings** | 0.38 | 5 | 24,492 | 93,050 |
| **Wisconsin** | 0.11 | 5 | 251 | 466 |
| **Squirrel** | 0.22 | 4 | 198,493 | 2,089 |
| **Ogbn-Arxiv** | 0.65 | 40 | 16,9343 | 1,166,243 |

We consider two types of datasets: Homophilic and Heterophilic. They are differentiated by the *homophily level* of a graph.

$$\mathcal{H} = \frac{1}{|V|} \sum_{v \in V} \frac{\text{Number of } v\text{'s neighbors who have the same label as } v}{\text{Number of } v\text{'s neighbors}}.$$

The low homophily level means that the dataset is more heterophilic when most of the neighbors are not in the same class, and the high homophily level indicates that the dataset is close to homophilic when similar nodes tend to be connected. In the experiments, we use four homophilic datasets, including Cora, CiteSeer, PubMed, and Ogbn-Arxiv, and four heterophilic datasets, including Texas, Wisconsin, Cornell, Squirrel, and Amazon-rating (Platonov et al., 2023)). The datasets we used covers various homophily levels.

### L.2    EXPERIMENTS SETUP

For the SGC backbone, we follow the Wang et al. (2021) setting where we run 10 runs for the fixed seed 42 and calculate the mean and the standard deviation. Furthermore, we fix the learning rate and weight decay in the same dataset and run 100 epochs for every dataset. For the GCN backbone, we follow the Guo et al. (2023) settings where we run 5 runs from the seed $\{0, 1, 2, 3, 4\}$ and calculate the mean and the standard deviation. We fix the hidden dimension to 32 and dropout rate to 0.6. Furthermore, we fix the learning rate to be 0.005 and weight decay to be $5e - 4$ and run 200 epochs for every dataset. We use the default splits in torch_geometric. We use Tesla-V100-SXM2-32GB in all experiments.

### L.3    RESULTS ANALYSIS

### L.3.1    CSBM RESULTS

The comparative results of Label-SBP and Feature-SBP against SGC are presented in Table 9. As the number of layers increases, SGC's node features suffer from oversmoothing, causing the two classes to converge and accuracy to drop by nearly 30 points from its peak at 2 layers, down to 45%. Conversely, after 300 layers, SBP maintains strong performance, with node features of different classes repelling each other. This effect limits oversmoothing to within-class interactions, and improves performance from 85 to 91 in Label-SBP and from 48 to 82 in Feature-SBP, further substantiating our approach to mitigating oversmoothing.

We visualize the node features learned by Label-SBP in Figure 9. We can see that from layer 0 to layer 200, the node features from different labels repel each other and aggregate the node features from the same labels. And we also visualize the adjacency matrix of Label-SBP and Feature-SBP in Figure 10 and Figure 11 respectively, further verifying the effectiveness of our theorem and insights.

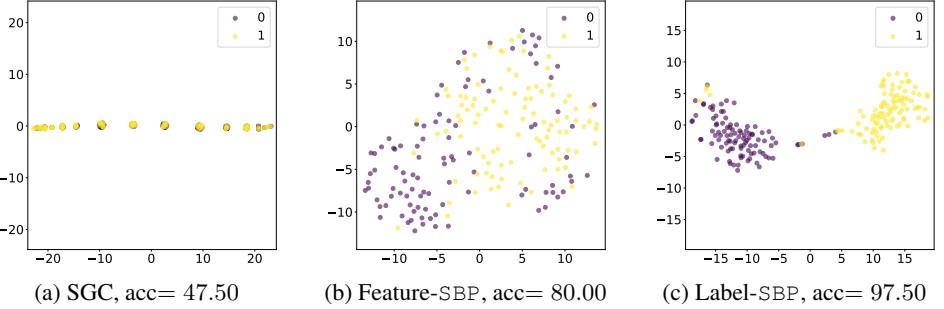

(a) SGC, acc= 47.50        (b) Feature-SBP, acc= 80.00        (c) Label-SBP, acc= 97.50

Figure 8: The t-SNE visualization of the node features and the classification accuracy from 2-CSBM and Layer= 300. Left is the result of the vallina SGC, and the middle and right are the results of SBP.

### L.3.2    GCN RESULTS

The results for GCN are detailed in Table 10, respectively. Overall, SBP consistently outperforms all previous methods, especially in deeper layers. Beyond 16 layers in GCN, SBP maintains superior

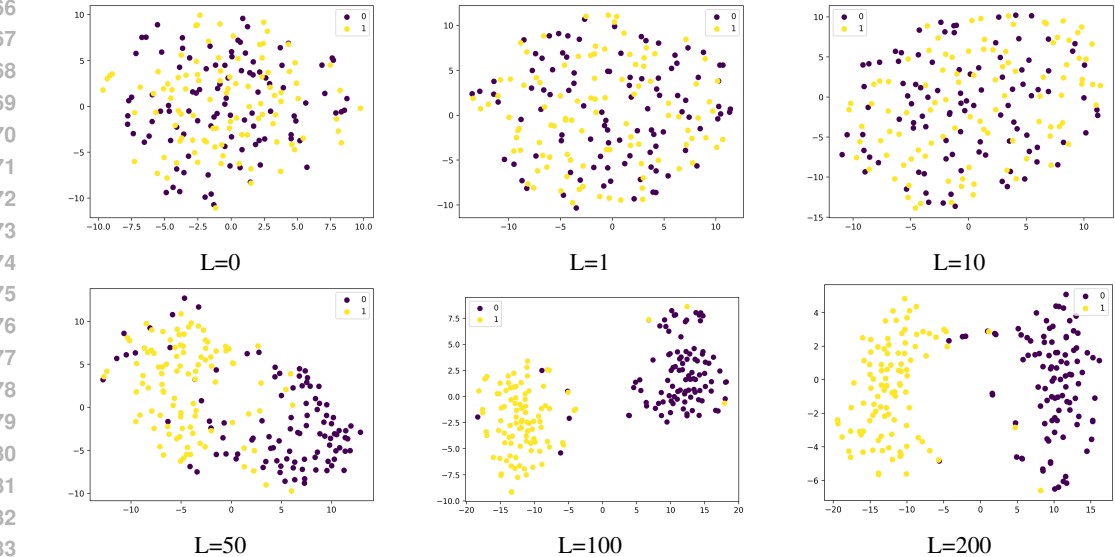

Figure 9: CSBM node features visualization. We update the features by Label-SBP. L is the propagation layer number. 0,1 represent different classes.

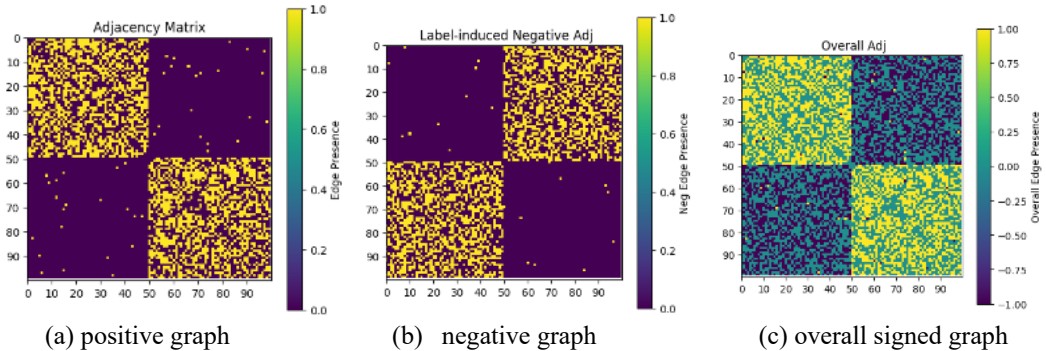

(a) positive graph      (b)   negative graph      (c) overall signed graph

Figure 10: The visualization of the adjacency matrix of Label-SBP. Here left is the positive graph; middle is the negative graph; right is the overall signed graph.

performance, affirming the effectiveness of our approach. Notably, SBP exceeds the best results of prior methods by at least 10% and up to 30% points in GCN's deepest layers, marking significant improvements. Moreover, unlike previous methods that perform best in shallow layers, SBP excels in moderately deep layers, as observed in GCN across all datasets. This further confirms the effectiveness of SBP.

### L.3.3 REPULSION ABLATION ON HETEROPHILIC DATASETS

Our method SBP can outperform other baselines under $\beta = 1$ across different layers, so we do not tune our hyper-parameters carefully. However, since $\beta$ is the weight of the negative adjacency matrix (equation 7) representing the repulsion between different nodes, as seen in Figure 4 and 5, the best performance of SBP appears when $\beta$ is larger in the heterophilic graphs, so the result in Figure 3a(a) is not the best performance of our SBP. To further show the effectiveness of our SBP, we conduct experiments on Cornell with different $\beta$ in Table 11, the best $\beta$ is 20 where the performance increases 25 points in deep layer 50.

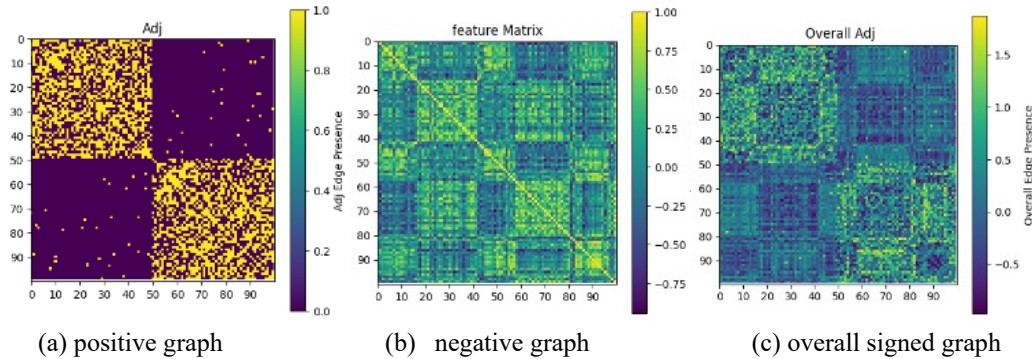

(a) positive graph        (b) negative graph        (c) overall signed graph

Figure 11: The visualization of the adjacency matrix of Feature-`SBP`. Here left is the positive graph; middle is the negative graph; right is the overall signed graph.

Table 9: CSBM test accuracy (%) comparison results. The best results are marked in blue on each layer. The second best results are marked in gray on each layer. We run 10 runs for the seed from $0 - 9$ and demonstrate the mean $\pm$ std in the table.

| Model | #L=2 | #L=5 | #L=10 | #L=20 | #L=50 | #L=100 | #L=200 |
|---|---|---|---|---|---|---|---|
| SGC | $73.25 \pm 6.90$ | $44.50 \pm 9.34$ | $45.75 \pm 9.36$ | $45.75 \pm 9.36$ | $45.75 \pm 9.36$ | $45.75 \pm 9.36$ | $45.75 \pm 9.36$ |
| Feature-SBP | $48.75 \pm 5.62$ | $53.75 \pm 6.45$ | $63.75 \pm 6.25$ | $77.00 \pm 5.45$ | $82.00 \pm 4.58$ | $82.50 \pm 5.12$ | $82.00 \pm 5.45$ |
| Label-SBP | $85.75 \pm 4.04$ | $93.50 \pm 4.06$ | $93.50 \pm 3.57$ | $93.50 \pm 3.57$ | $92.25 \pm 3.44$ | $93.25 \pm 3.72$ | $91.25 \pm 6.05$ |

### L.3.4   PERFORMANCE OF `SBP` ON MORE BENCHMARKS

We further compare our `SBP` with SGC on six additional datasets (Platonov et al., 2023) in Table 12. Our `SBP` outperforms SGC on five out of these six datasets. We believe that these six datasets, combined with the nine datasets presented in Table 3 of our paper, provide sufficient evidence to demonstrate the effectiveness of our approach.

### L.3.5   COMBINE `SBP` TO OTHER METHODS

In this paper, we focus on introducing a novel theoretic signed graph perspective for oversmoothing analysis, so we do not take many tricks into account or carefully fine-tune our hyperparameters. Thus, our results in the paper are not as comparable to previous baselines (Chen et al., 2020b; Luan et al., 2022; Eliasof et al., 2021). However, existing oversmoothing researches are indeed hard to compare, because they are often composed of multiple techniques — such as residual, BatchNorm, data augmentation — and the parameters are often heavily (over-)tuned on small-scale datasets. And it becomes clear that to attain SOTA performance, one needs to essentially compose multiple such techniques without fully understanding their individual roles. For example, GCNII uses both initial residual connection and identity map, futher combined with dropout.

Our goal is to provide a new unified understanding of these techniques, so we justified it by showing that SBP as a single simple technique can attain good performance. And we believe that it would work complementarily with other techniques in the field, because oversmoothing is still challenging to solve with a very larger depth.

To further verify the effectiveness, we combine our SBP to one of the SOTA settings GCNII (Chen et al., 2020b) and the results are as seen in Table 13. The results indicate that after combining our method, GCNII demonstrates greater robustness as the layers go deeper, particularly in the middle layers (layer=8), highlighting the efficacy of our signed graph insight.

### L.3.6   PERFORMANCE OF `SBP` ON LARGE-SCALE GRAPHS

We conducted experiments with a larger graph ogbn-products than ogbn-arxiv under 100 epochs and 2 layers in Table 14. The results indicate that our `SBP` outperforms the initial GCN baselines. Given the results presented for ogbn-arxiv in Table 5 of our paper, we believe these findings adequately demonstrate the performance of our `SBP` on large-scale graphs.

Table 10: GCN test accuracy (%) comparison results. The best results are marked in blue and the second best results are marked in gray on every layer. We run 5 runs for the seed from $0 - 4$ and demonstrate the mean $\pm$ std in the table.

| Model | #L=2 | #L=4 | #L=8 | #L=16 | #L=32 | #L=64 |
|---|---|---|---|---|---|---|
| | | *Cora* (McCallum et al., 2000) | | | | |
| GCN (Kipf & Welling, 2017) | 80.68 ±0.09 | 79.69 ±0.00 | 74.32 ±0.00 | 30.95 ±0.00 | 30.95 ±0.00 | 24.85 ±7.46 |
| GAT (Veličković et al., 2018) | 81.48 ± 0.48 | 80.69 ± 0.93 | 58.59 ± 1.95 | 25.17 ± 5.67 | 31.93 ± 0.21 | 28.38 ± 0.00 |
| wGCN (Eliasof et al., 2022) | 80.97 ± 0.28 | 80.51 ± 0.00 | 80.46 ± 1.77 | 70.53 ± 22.09 | 80.02 ± 0.12 | 27.90 ± 6.09 |
| BatchNorm (Ioffe & Szegedy, 2015) | 78.09 ±0.00 | 77.87 ±0.02 | 73.62 ±0.57 | 70.79 ±0.00 | 53.90 ±2.19 | 35.32 ±3.41 |
| PairNorm (Zhao & Akoglu, 2019) | 79.01 ±0.00 | 78.26 ±0.50 | 73.21 ±0.00 | 62.96 ±0.00 | 48.13 ±0.91 | 44.01 ±3.46 |
| ContraNorm (Guo et al., 2023) | 81.55 ±0.21 | 79.61 ±0.75 | 77.71 ±0.00 | 63.35 ±0.00 | 44.56 ±4.83 | 38.97 ±0.00 |
| DropEdge (Rong et al., 2019) | 78.38 ±0.00 | 74.47 ±0.00 | 26.91 ±0.83 | 22.24 ±3.04 | 27.18 ±0.00 | 25.98 ±6.00 |
| Residual | 80.68 ±0.09 | 78.77 ±0.00 | 79.26 ±0.21 | 40.91 ±0.00 | 30.95 ±0.00 | 27.90 ±6.09 |
| Feature-SBP | 80.44 ±0.83 | 79.26 ±1.18 | 78.56 ±0.59 | 77.22 ±0.55 | 73.65 ±0.48 | 61.62 ±5.24 |
| Label-SBP | 80.31 ±0.70 | 79.16 ±1.30 | 79.50 ±0.00 | 77.43 ±1.49 | 74.52 ±0.36 | 65.02 ±2.97 |
| | | *CiteSeer* (Giles et al., 1998) | | | | |
| GCN (Kipf & Welling, 2017) | 67.45 ±0.54 | 65.62 ±0.25 | 37.22 ±2.46 | 22.03 ±4.76 | 19.65 ±0.00 | 19.65 ±0.00 |
| GAT Veličković et al. (2018) | 69.91 ± 0.86 | 67.47 ± 0.22 | 44.71 ± 3.07 | 23.48 ± 1.36 | 24.40 ± 0.40 | 25.95 ± 2.17 |
| wGCN (Eliasof et al., 2022) | 66.21 ± 0.63 | 66.49 ± 0.69 | 66.79 ± 0.00 | 57.54 ± 18.94 | 19.65 ± 0.00 | 19.65 ± 0.00 |
| BatchNorm (Ioffe & Szegedy, 2015) | 63.44 ±0.94 | 62.34 ±0.25 | 61.36 ±0.00 | 50.58 ±1.24 | 41.41 ±0.00 | 35.00 ±1.09 |
| PairNorm (Zhao & Akoglu, 2019) | 63.58 ±0.63 | 64.32 ±0.95 | 61.95 ±1.24 | 50.06 ±0.00 | 37.21 ±1.87 | 36.09 ±0.07 |
| ContraNorm (Guo et al., 2023) | 66.83 ±0.49 | 64.78 ±0.92 | 60.70 ±0.60 | 44.79 ±1.65 | 37.36 ±0.25 | 30.85 ±0.81 |
| DropEdge (Rong et al., 2019) | 63.86 ±0.03 | 62.24 ±0.90 | 24.73 ±5.72 | 20.65 ±0.00 | 20.04 ±0.19 | 19.95 ±0.09 |
| Residual | 67.45 ±0.54 | 66.21 ±0.16 | 67.34 ±0.00 | 33.21 ±0.00 | 19.65 ±0.00 | 19.65 ±0.00 |
| Feature-SBP | 67.38 ±0.66 | 66.94 ±0.00 | 66.29 ±0.02 | 65.35 ±1.99 | 61.43 ±0.00 | 42.09 ±1.65 |
| Label-SBP | 67.23 ±0.64 | 66.72 ±0.00 | 66.29 ±0.89 | 65.50 ±2.13 | 59.93 ±0.85 | 44.41 ±1.57 |
| | | *PubMed* (Canese & Weis, 2013) | | | | |
| GCN (Kipf & Welling, 2017) | 76.44 ±0.34 | 76.52 ±0.32 | 69.58 ±5.89 | 39.92 ±0.00 | 39.92 ±0.00 | 39.92 ±0.00 |
| +BatchNorm (Ioffe & Szegedy, 2015) | 75.52 ±0.12 | 77.15 ±0.00 | 77.10 ±0.00 | 76.92 ±0.00 | 75.43 ±0.00 | 69.33 ±1.01 |
| +PairNorm (Zhao & Akoglu, 2019) | 75.66 ±0.11 | 76.71 ±0.00 | 77.99 ±0.00 | 77.22 ±0.39 | 75.52 ±2.02 | 71.22 ±3.68 |
| +ContraNorm (Guo et al., 2023) | 76.05 ±0.33 | 78.42 ±0.00 | OOM | OOM | OOM | OOM |
| +DropEdge (Rong et al., 2019) | 73.41 ±0.03 | 73.96 ±0.79 | 52.51 ±10.91 | 40.27 ±0.00 | 39.90 ±0.59 | 40.08 ±0.39 |
| +Residual | 76.44 ±0.34 | 77.28 ±0.00 | 77.38 ±0.00 | 63.14 ±3.05 | 39.92 ±0.00 | 39.92 ±0.00 |
| Feature-SBP | 75.72 ±0.06 | 76.84 ±0.00 | 78.39 ±0.00 | 79.71 ±0.00 | 77.59 ±0.23 | 78.06 ±0.13 |
| Label-SBP | 76.33 ±0.25 | 76.91 ±0.00 | 77.60 ±0.49 | 76.31 ±0.00 | 77.17 ±0.67 | 78.01 ±0.16 |

Table 11: Ablation study of negative weight $\beta$ on Cornell dataset.

| Layer | 2 | 5 | 10 | 20 | 50 |
|---|---|---|---|---|---|
| $\beta = 0.1$ | $72.97 \pm 0.00$ | $67.57 \pm 0.00$ | $51.53 \pm 0.00$ | $35.14 \pm 0.00$ | $29.73 \pm 0.00$ |
| $\beta = 1$ (default) | $72.97 \pm 0.00$ | $67.57 \pm 0.00$ | $51.53 \pm 0.00$ | $45.95 \pm 0.00$ | $35.14 \pm 0.00$ |
| $\beta = 10$ | $70.27 \pm 0.00$ | $67.57 \pm 0.00$ | $58.11 \pm 1.35$ | $51.53 \pm 0.00$ | $51.53 \pm 0.00$ |
| $\beta = 20$ (best) | $70.27 \pm 0.00$ | $70.27 \pm 0.00$ | $67.57 \pm 0.00$ | $59.46 \pm 0.00$ | $59.46 \pm 0.00$ |
| $\beta = 50$ | $64.60 \pm 0.00$ | $40.54 \pm 0.00$ | $40.54 \pm 0.00$ | $40.54 \pm 0.00$ | $40.54 \pm 0.00$ |

### L.3.7 FURTHER OPTIMIZATION BASED ON SBP

Based on the experiment results, we want to propose 2 strategies for further optimization.

1) hyper-parameter tuning on the negative weight $\beta$. As seen in Figures 4 and 5, we found that $\beta$ influences the performance a lot, our default $\beta = 1$ for Table 3 and 4 is certainly not optimal for the above 4 homophilic datasets. We suggest tuning higher $\beta$ for the heterophilic graphs since they need more repulsion and smaller for the homophilic datasets. As the layer deepens, maybe greater weight should be placed on the negative adjacency graphs to alleviate oversmoothing.

2) adapt our SBP to more effective GNNs. Our method is simple, architecture-free, without additional learnable parameters, and thus can be flexibly applied in various architectures. As seen in Appendix L.3.5, we adapt our SBP to the GCNII models, and the results increase more than

Table 12: Performance Comparison on more datasets

| | actor | penny94 | roman-empire | Tolokers | Questions | Minesweeper |
|---|---|---|---|---|---|---|
| SGC | $29.18 \pm 0.10$ | $72.56 \pm 0.05$ | $40.83 \pm 0.03$ | $78.18 \pm 0.02$ | $97.09 \pm 0.00$ | $80.43 \pm 0.00$ |
| Feature-SBP | $34.93 \pm 0.02$ | $75.68 \pm 0.01$ | $66.48 \pm 0.02$ | $78.24 \pm 0.04$ | $97.14 \pm 0.02$ | $80.00 \pm 0.00$ |
| Label-SBP | $34.94 \pm 0.00$ | $75.74 \pm 0.01$ | $66.32 \pm 0.01$ | $78.46 \pm 0.08$ | $97.15 \pm 0.02$ | $80.00 \pm 0.00$ |

Table 13: Performance Comparison between SBP and GCNII under the GCNII settings on Cora and Citesser datasets

|  |  | 2 | 4 | 8 | 16 | 32 | 64 |
|---|---|---|---|---|---|---|---|
| Cora | GCNII | $78.58 \pm 0.00$ | $77.76 \pm 0.24$ | $73.47 \pm 3.82$ | $78.12 \pm 1.32$ | $82.54 \pm 0.00$ | $81.34 \pm 0.53$ |
|  | Label-SBP | $78.74 \pm 1.54$ | $78.87 \pm 0.00$ | $79.14 \pm 0.35$ | $79.17 \pm 0.41$ | $80.86 \pm 0.32$ | $81.38 \pm 0.30$ |
|  | Feature-SBP | $77.95 \pm 0.91$ | $78.82 \pm 0.00$ | $78.11 \pm 1.62$ | $78.82 \pm 0.29$ | $81.82 \pm 0.47$ | $81.65 \pm 0.40$ |
| Citesser | GCNII | $61.66 \pm 0.67$ | $63.23 \pm 2.31$ | $64.58 \pm 2.66$ | $66.21 \pm 0.64$ | $69.38 \pm 0.83$ | $69.73 \pm 0.26$ |
|  | Label-SBP | $65.31 \pm 0.63$ | $63.93 \pm 3.66$ | $68.33 \pm 0.99$ | $66.46 \pm 0.00$ | $70.00 \pm 0.81$ | $69.47 \pm 0.25$ |
|  | Feature-SBP | $65.63 \pm 0.87$ | $64.43 \pm 3.55$ | $68.44 \pm 1.19$ | $66.94 \pm 0.00$ | $69.98 \pm 0.93$ | $69.66 \pm 0.28$ |

Table 14: Performance of different models on ogbn-products dataset. Time means the runtime, the format is (hour: minutes: seconds).

| Method | Accuracy | Time |
|---|---|---|
| GCN | 73.96 | 00:06:33 |
| BatchNorm | 74.93 | 00:06:18 |
| Feature-SBP | 74.90 | 00:06:43 |
| Label-SBP | 76.62 | 00:06:39 |

adaptation in vanilla GNN as shown in Table 3 and 4. Besides, compared to the GCNII, our SBP is more robust and stable to the layers as seen in Table 13.

