# OpenReview forum: "Oversmoothing as Loss of Sign: Towards Structural Balance in Graph Neural Networks"
_ICLR.cc/2025/Conference — Submitted to ICLR 2025_

### Official Review · Reviewer_gmfF · 2024-10-17

**Soundness:** 2
**Presentation:** 3
**Contribution:** 1
**Rating:** 3
**Confidence:** 4

**Summary:**

The paper addresses the issue of oversmoothing in Graph Neural Networks.
The authors propose a unified approach to anti-oversmoothing techniques by utilizing the concept of signed graphs, where both positive and negative edges are considered.
Through structural balance theory, they show that improving the balance of signed graphs can prevent oversmoothing and enhance node classification.
They introduce Structural Balanced Propagation (SBP), which uses both label and feature information to improve graph structure. Theoretical and empirical results on various datasets demonstrate SBP’s effectiveness.

**Strengths:**

1. The idea of viewing over-smoothing from the perspective of the sign-graph is interesting. The results show that many over-smoothing methods can indeed be generalized under the same form, which is promising.

2. I like the cSBM experiments. They demonstrate the effectiveness of the proposed method well.

**Weaknesses:**

**1. Incompleteness of Experiments:**
The experimental evaluation lacks completeness, particularly in terms of benchmark comparisons. The baselines used are outdated, as more recent state-of-the-art methods like GCNII and $\omega$GCN [1] are not considered. In fact, from Table 2 in [1], it is evident that these methods achieve significantly higher results on datasets like Cora, CiteSeer, and PubMed. A more comprehensive table, similar to Table 2 from [1], should be provided for better comparison. Additionally, the results on heterophilic datasets are not convincing, as newer methods, such as those presented in [2], show superior performance.

**2. Over-smoothing Methods Coverage:**
The selection of anti-oversmoothing methods in Section 3 is limited. While DropEdge is included, more recent methods like DropMessage [3] are ignored. Moreover, the paper overlooks the importance of residual connections, which are extensively explored in methods like GCNII and PDE-GCN, as discussed in [1]. Therefore, the paper does not cover a sufficient range of contemporary techniques aimed at addressing oversmoothing.

**3. Recent Challenges to Over-smoothing:**
The very premise of oversmoothing has been questioned in recent works [4, 5], which argue that oversmoothing may not exist structurally. If the authors are positioning SBP as a solution to oversmoothing, they should engage with these discussions. Additionally, Table 7 shows that when GCN is combined with SBP, performance degradation is still severe, whereas methods like GCNII and $\omega$GCN do not exhibit such issues. This raises doubts about whether SBP truly mitigates oversmoothing effectively.

**4. Scalability Concerns:**
The scalability of SBP is not fully explored. There is no complexity analysis or running time comparison provided. Furthermore, referring to the ogbn-arxiv dataset as "large" is misleading; it is not sufficient to demonstrate the scalability of the method. A more rigorous analysis of both space and time scalability is necessary.

**References:**
- [1] Eliasof, Moshe, Lars Ruthotto, and Eran Treister. "Improving graph neural networks with learnable propagation operators." International Conference on Machine Learning. PMLR, 2023.
- [2] Luan, Sitao, et al. "Revisiting heterophily for graph neural networks." Advances in Neural Information Processing Systems 35 (2022): 1362-1375.
- [3] Fang, Taoran, et al. "Dropmessage: Unifying random dropping for graph neural networks." Proceedings of the AAAI Conference on Artificial Intelligence. Vol. 37. No. 4. 2023.
- [4] Cong, Weilin, Morteza Ramezani, and Mehrdad Mahdavi. "On provable benefits of depth in training graph convolutional networks." Advances in Neural Information Processing Systems 34 (2021): 9936-9949.
- [5] Peng, Jie, Runlin Lei, and Zhewei Wei. "Beyond Over-smoothing: Uncovering the Trainability Challenges in Deep Graph Neural Networks." arXiv preprint arXiv:2408.03669 (2024).

**Questions:**

I'm not sure about the detailed implementation of the norm techniques used in the paper. Are the BatchNorm, contra-norm learnable? If so, will the learnable parameter affect the analysis in Theorem 3.1?

---

> ### Author Response · Authors · 2024-11-20
> **Answer to Weakness 1**
>
> We thank Reviewer gmfF for appreciating our theoretical insights, simulation verification, and practical methods. Your recommended papers are very important to our research, and we will cite all of them to fulfill our investigation. We have tried our best to address your concerns below.
> ***
> **w1.1** The baselines used are outdated, as more recent state-of-the-art methods like GCNII and wGCN are not considered. wGCN achieves significantly higher results on datasets like Cora, CiteSeer, and PubMed. A more comprehensive table should be provided for better comparison.
>
> **a1.1** We have thoughtfully chosen 9 classic baselines encompassing a diverse range of approaches, including normalization-based, randomization, and residual-based methods. With your valuable guidance in mind, we have also included GCNII and wGCN in our implementation.
>
> - GCNII
>
> We use GCNII as the backbone and adapt our Label-induced negative adjacency matrix and Feature-induced negative adjacency matrix derived from Labe/Feature-SBP to it. The results are as follows:
>
> | **#layers** | **2** | **4** | **8** | **16** | **32** | **64** |
> | --- | --- | --- | --- | --- | --- | --- |
> | cora |  |  |  |  |  |  |
> | GCNII | 78.58 ± 0.00 | 77.76 ± 0.24 | 73.47 ± 3.82 | 78.12 ± 1.32 | **82.54 ± 0.00** | 81.34 ± 0.53 |
> | label-sbp | **78.74 ± 1.54** | **78.87 ± 0.00** | **79.14 ± 0.35** | **79.17±0.41** | 80.86 ± 0.32 | 81.38 ± 0.30 |
> | Feature-sbp | 77.95 ± 0.91 | 78.82 ± 0.00 | 78.11± 1.62 | 78.82±0.29 | 81.82 ± 0.47 | **81.65 ± 0.40** |
> | citesser |  |  |  |  |  |  |
> | GCNII | 61.66 ± 0.67 | 63.23 ± 2.31 | **64.58  ± 2.66** | 66.21 ± 0.64 | 69.38 ± 0.83 | **69.73 ± 0.26** |
> | label-sbp | 65.31± 0.63 | 63.93± 3.66 | 68.33 ± 0.99 | 66.46 ± 0. 00 | **70.00 ± 0.81** | 69.47±0.25 |
> | feature-sbp | **65.63±0.87** | **64.43± 3.55** | 68.44 ± 1.19 | **66.94± 0.00** | 69.98 ± 0.93 | 69.66±0.28 |
>
> The results show that our method not only achieves higher scores than GCNII on average but also is more robust to the layers than the GCNII, especially in the middle layers (layer=8), showing the efficacy of our theory-guided method.
>
> Remark that our paper focuses on introducing a unified signed graph framework to analyze and alleviate the oversmoothing, so we do not carefully select the hyperparameters and use tricks during training. Thus our results in the paper are not so comparable to Table 2 from wGCN.
> - wGCN
>
> | **#layers** | **2** | **4** | **8** | **16** | **32** | **64** |
> | --- | --- | --- | --- | --- | --- | --- |
> | cora |  |  |  |  |  |  |
> | wGCN | **80.97±0.28** | **80.51±0.00** | **80.46 ± 1.77** | 70.53 ± 22.09 | **80.02 ± 0.12** | 27.90 ± 6.09 |
> | feature-sbp | 80.44 ±0.83 | 79.26 ±1.18 | 78.56 ± 0.59 | 77.22 ±0.55 | 73.65 ±0.48 | 61.62 ±5.24 |
> | label-sbp | 80.31 ±0.70 | 79.16 ±1.30 | 79.50 ± 0.00 | **77.43 ±1.49** | 74.52 ±0.36 | **65.02 ±2.97** |
> | citeseer |  |  |  |  |  |  |
> | wGCN | 66.21±0.63 | 66.49±0.69 | **66.79±0.00** | 57.54±18.94 | 19.65±0.00 | 19.65±0.00 |
> | feature-sbp | **67.38 ±0.66** | **66.94 ±0.00** | 66.29 ±0.02 | 65.35 ±1.99 | **61.43 ±0.00** | 42.09 ±1.65 |
> | label-sbp | 67.23 ±0.64 | 66.72 ±0.00 | 66.29 ±0.89 | **65.50 ±2.13** | 59.93 ±0.85 | **44.41 ±1.57** |
>
> Since wGCN does not have the officially published code, we compare our SBP with wGCN under our setting for a fair comparison.
>
> The results show that although they are competitive in the shadow layers (layer=2,4,8),  they are not as stable as our SBP method in the deep layers, i.e., they cause a high variance in layer=16. Furthermore,  their performance decreases fast and performs worse than ours in the deep layers, such as layer=16, 32, 64.
>
> ***
>
> **w1.2** The results on heterophilic datasets are not convincing, as newer methods, such as those presented in [2], show superior performance.
>
> **a1.2** [2] introduced ACM-GCN, a method tailored for heterophilic graphs, leveraging adaptively exploit aggregation, diversification, and identity channels node-wisely to extract richer localized information.
> In contrast, our SBP is more simple and general, not specifically tailored for heterographs but applicable to both homogeneous and heterogeneous under various GNN settings.
> Additionally, ACM-GCN employs several training techniques, such as calculating features of different channels with varying attention mechanisms which we do not employ, making direct comparisons challenging.
>
> We have added these experiments in Appendix L.
> We believe that the current experiments plus the comparison of GCNII and wGCN are sufficient to complement the experiments. Please feel free to let me know if you have more concerns.

---

> ### Author Response · Authors · 2024-11-20
> **Answer to Weakness 2**
>
> **w2.1** The selection of anti-oversmoothing methods in Section 3 is limited. Therefore, the paper does not cover a sufficient range of contemporary techniques aimed at addressing oversmoothing.
>
> **a2.1** We acknowledge the abundance of anti-oversmoothing techniques and we have discussed some of them in the related work.
> Here, we primarily selected 9 classic methods and analyzed them under our signed graph framework with the equivalent positive and negative graphs in Section 3 of the paper.
> These methods include standard GNNs, 3 normalization-based techniques (BatchNorm, PairNorm, and the specialized anti-oversmoothing method ContraNorm), 2 randomized approaches (dropedge and dropnode), and 4 classic residual-based methods (residual, APPNP, JKNET, and DAGNN).
> We believe that these methods are enough to show our unified signed graph perspective for oversmoothing.
> ***
> **w2.2** While DropEdge is included, more recent methods like DropMessage [3] are ignored.
>
> **a2.2**  DropMessage is a unified way of dropnode, dropedge and dropout with a more adaptable mask strategy. We have discussed the dropode and dropedge in our paper. DropMessage can be viewed as randomly dropping some dimension of the aggregated node features instead of the whole node.  In our signed graph framework, we formalize the propagation of DropMessage, which can be represented as $H^{(k)}= AH^{(k-1)}-M_m,$ where if dropping $AH^{(k-1)}_{ij}$, then $M_{ij}=AH^{(k-1)}_{ij}$ else $M_{i,j}=0$.
> ***
> **w2.3** the paper overlooks the importance of residual connections, which are extensively explored in methods like GCNII and PDE-GCN, as discussed in [1].
>
> **a2.3**  In this paper, we have selected four classic residual-based methods for analysis within our signed graph propagation framework, chosen for their layered combination strategies. Additionally, it is worth noting that other residual-based approaches can be viewed as combinations or refinements of these foundational types
>
> - residual: combines the last-layer and the current layers features: $H^{(k)}=(1-\alpha)H^{(k-1)}+\alpha AH^{(k-1)}$
> - APPNP: extend the last-layer connection to the initial layer connection to further allevaite the oversmoothing: $H^{(k)}=(1-\alpha)H^{(0)}+\alpha AH^{(k-1)}$
> - JKNET: selectively combines aggregations from different layers through operations such as concatenation or max-pooling at the output, i.e., the representations "jump" to the last layer. $H^{(k)}=\Sigma_{i=j_0}^{j_m}\alpha_i H^{(i)} , \Sigma_{i=0}^{m}\alpha_i=1, \{j_0,...j_m\}$ is the index of the selective layers.
> - DAGNN: tries to adaptively add all the features from the previous layer to the current layer features with the additional learnable coefficients. $H^{(k)}=\Sigma_{i=0}^{k-1}\alpha_i H^{(i)}, \Sigma_{i=0}^{k-1}\alpha_i=1$.
>
> Furthermore, we give a detailed discussion of GCNII, wGCN, and PDE-GCN.
>
> - GCNII: is an improved version of APPNP with the learnable coefficients $\alpha_i$ and changes the learnable weight W to $(1-\beta_i)I+\beta_i W$ each layer, so it shares the same positive and negative graph as APPNP.
>
> - wGCN: incorporates trainable channel-wise weighting factors $\omega$ to learn and mix multiple smoothing and sharpening propagation operators at each layer, same as the residual combining last-layer features but change parameters $\alpha$ to be learnable with a more detailed selection strategy.
>
> - PDE-GCN: In this paper, we focus on exploring diverse methods that can be formulated within the context of a signed graph. PDE-GCN, however, is a novel work that involves Partial Differential Equations (PDEs) and the trainable convolution from CNNs. Due to the unavailability of open-source code, we do not have enough time to implement and analyze it right now.
>
> ***
> We have added these discussions in Appendix C.2 and C.3.

---

> ### Author Response · Authors · 2024-11-20
> **Answer to Weakness 3**
>
> **w3.1** The premise of oversmoothing has been questioned in recent works [4, 5], which argue that oversmoothing may not exist structurally. If the authors are positioning SBP as a solution to oversmoothing, they should engage with these discussions.
>
> **a3.2** Thank you for your comment and for pointing us to these references. We appreciate the opportunity to clarify and engage with these discussions.
>
> - While [4] questions the existence of oversmoothing in trained GNNs, their observations are primarily based on specific experimental settings that may not fully capture the oversmoothing challenge present in the literature. Specifically, the empirical observations in [4] are based on 10-layer GCNs, which, while useful for their argument, may not represent the behavior of deeper networks or other GNN architectures. Moreover, Figure 2 is based on a normalized metric, which might not be the most appropriate. To see this point, suppose one wants to classify two points. In one case, we have 0.01 vs -0.01 and in the other case, we have 100 vs -100. While the normalized distance considered in [4] would be the same for these two cases, the latter case has a much larger margin, and it would be thus much easier to learn a classifier.
> - On the other hand, [5] suggests that trainability of deep GNNs is more of a problem than
> over-smoothing. However, over-smoothing naturally presents challenges for training deep GNNs, as when oversmoothing happens, gradients vanish across different nodes. Besides, [5] compares 3 models GCN, ResGCN and GCNII, proving that GCNII is the best backbone. We have adapted our SBP to GCNII in a1.1 and the results showed that our SBP outperforms GCNII on average, especially in the middle layers.
>
> ***
> **w3.2** Additionally, Table 7 shows that when GCN is combined with SBP, performance degradation is still severe, whereas methods like GCNII and ωGCN do not exhibit such issues. This raises doubts about whether SBP truly mitigates oversmoothing effectively.
>
> **a3.2** This question bears a resemblance to w1.1. Our paper is centered on presenting a unified signed graph framework aimed at analyzing and mitigating oversmoothing effects. As a result, we did not meticulously fine-tune hyperparameters or employ training optimizations. Consequently, our results in the paper may not be directly comparable to those in Table 2 of [1]. However, we have conducted a fair comparison between our SBP method and GCNII and wGCN (as discussed in a1.1), with SBP consistently outperforming these two baseline methods overall.

---

> ### Author Response · Authors · 2024-11-20
> **Answer to Weakness 4**
>
> **w4.1** The scalability of SBP is not fully explored. There is no complexity analysis or running time comparison provided.
>
> **a4.1**
> - Complexity analysis in the small graph
>     - Given $n_t$ training nodes and $d$ edges in the graph, our Label-SBP increases the edge from $O(d)$ to $O(n_t^2)$, thereby increasing the computational complexity to $O(n_t^2d)$.
>     - For Feature-SBP, the additional computational complexity primarily stems from the negative graph propagation, which involves $X_{0}X_{0}^T \in \mathbb{R}^{n\times n}$, increasing the overall complexity to $O(n^2d)$.
>
> We show the computation time of different methods in the following table. On average, we improve performance on 8 out of 9 datasets  (as shown in Table 3 of the paper) with less than 0.05s overhead—even faster than three other baselines. We believe this time overhead is acceptable given the benefits it provides.
>
> |  | **label-sbp** | **feature-sbp** | **BatchNorm** | **ContraNorm** | **Residual** | **JKNET** | **DAGNN** | **SGC** |
> | --- | --- | --- | --- | --- | --- | --- | --- | --- |
> | pre-compute time | 0.1809s | 0.1520s | 0.1860s | 0.1888s | 0.0604s | 0.0577s | 0.1438s | 0.1307s |
> | trian time | 0.1071s | 0.1060s | 0.1076s | 0.1038s | 0.1368s | 0.1446s | 0.1348s | 0.1034s |
> | totol time | 0.2879s | 0.2580s | 0.2935s | 0.2926s | 0.1972s | 0.2023s | 0.2786s | 0.2341s |
>
> ***
>
> **w4.2** Furthermore, referring to the ogbn-arxiv dataset as "large" is misleading; it is not sufficient to demonstrate the scalability of the method. A more rigorous analysis of both space and time scalability is necessary.
>
> **a4.2**
>
> For the large-scale graph, we introduce the modified version of Label-SBP and Feature-SBP.
>
> - For Label-SBP, we only remove edges when pairs of nodes belong to different classes. This approach allows our modified Label-SBP to even further enhance the sparsity of large-scale graphs (maintain $O(d)$).
> - For Feature-SBP, as the number of nodes $n$ increases, the complexity of this matrix operation grows quadratically, i.e., $o(n^2d)$. To address this, we reorder the matrix multiplication from $-X_{0}X_{0}^T \in \mathbb{R}^{n\times n}$ to $-X_{0}^TX_{0} \in \mathbb{R}^{d\times d}$. This preserves the distinctiveness of node representations across the feature dimension, rather than across the node dimension as in the original node-level repulsion. The modified version of Feature-SBP can be expressed as $X^{k}=(1-\lambda)X^{(k-1)}+\lambda(\hat{A}X^{(K-1)} - X^{(K-1)} \text{softmax}(-X_{0}^TX_{0}))$. This transposed alternative has a linear complexity in the number of samples, i.e., $O(nd^2)$. In cases where $n \gg d$, this modified version significantly reduces the computational burden.
>
> We compare our scaled SBP with other baselines on ogbn-arxiv. Among all the training times of the baselines, our Label-SBP achieves the 3rd fastest time, while Feature-SBP ranks 5th. Therefore, we recommend utilizing Label-SBP for large-scale graphs, as they usually have enough node labels and do not require significant time overhead. We believe that, although there is a slight time increase, it is acceptable given the benefits.
>
> | **ogbn-arixv** | **label-sbp** | **feature-sbp** | **BatchNorm** | **ContraNorm** | **Drop** | **SGC** |
> | --- | --- | --- | --- | --- | --- | --- |
> | train time | 5.5850 | 6.1333 | 5.3872 | 5.8375 | 9.5727 | 5.3097 |
> | rank | 3 | 5 | 2 | 4 | 6 | 1 |
> - Complexity analysis on Larger-scale graph
>
> We conducted experiments with a larger graph ogbn-products than ogbn-arxiv under 100 epochs and 2 layers. The results indicate that our SBP outperforms the initial GCN baselines. Combining the results presented for ogbn-arxiv in Table 5 of our paper, we believe these experiments on 2 large-scale datasets sufficiently demonstrate the efficacy of our SBP.
>
> |  | ogbn-products | time |
> | --- | --- | --- |
> | GCN | 73.96 | 00:06:33 |
> | BatchNorm | 74.93 | 00:06:18 |
> | Feature-SBP | 74.90 | 00:06:43 |
> | Label-SBP | 76.62 | 00:06:39 |
> ***
> We have added these experiments and discussions in Appendix K and L.

---

> ### Author Response · Authors · 2024-11-20
> **Answer to Question1**
>
> **q1**  I am not sure about the detailed of the norm techniques used in the paper. Are the BatchNorm, contra-norm learnable?  If so, will the learnable parameter affect the analysis in Theorem 3.1?
>
> **a1** BatchNorm and ContraNorm are non-learnable components. I kindly suggest referring to Section 3.1 and Appendix C.1 in our paper for a comprehensive understanding of BatchNorm and ContraNorm, where detailed descriptions are provided.
> - BatchNorm centers the node representations $X$ to zero mean and unit variance and can be written as BatchNorm($x_i$) $=\frac{1}{\sqrt{\sigma^2 + \epsilon}}(x_i - \frac{1}{n}\Sigma_{i=1}^n x_i) $, where $ \epsilon > 0$ and $\sigma^2$ is the variance of node features.
> - ContraNorm is inspired by the uniformity loss from contrastive learning, aiming to alleviate dimensional feature collapse. It can be written as $\hat{X} = (1 + \alpha) \hat{A} X- \alpha / \tau ( X X^{T} \hat{A} ) X$, where $\alpha$ and $\tau$ are the hyperparameters.
>
> If you have any other questions about the details of the normalization methods, feel free to ask us.
>
> ***
>
> Thank you again for your insightful comments, which significantly strengthen our work! We are happy to address your further concerns in the discussion stage.

---

> > ### Comment · Reviewer_gmfF · 2024-11-20
> >
> > Thank you very much for your response. I truly appreciate the detailed feedback provided in such a short timeframe. It has been helpful in clarifying the parts related to scalability and norms.
> >
> > I understand the difficulty in reproducing certain code results promptly. However, my primary concerns remain unresolved:
> >
> > 1. **Performance advantage**: The reported results for GCNII seem significantly underestimated. On Citeseer, it should achieve an accuracy of 72+, on Cora 85+, and on OGBN-Arxiv both GCNII and GCN should reach 71+ accuracy. Given that methods like GCNII and wGCN report results on the standard PyG splits, the unusually low performance here is questionable.
> >
> > 2. **Over-smoothing**: More critically, the core issue lies with over-smoothing itself. As observed, methods like GCNII and $\omega$GCN exhibit **almost no over-smoothing** on real-world graphs. They can maintain performance even with 64 layers. It is unclear how SBP improves performance on real datasets by **better addressing over-smoothing** when this does not appear to be a significant bottleneck for these methods. As discussed in [4, 5], **over-smoothing is not a core limitation for GNNs on real-world graphs**. Therefore, analyzing where SBP genuinely offers performance advantages may be a more promising direction.
> >
> > I believe the work would be better if the authors could:
> > 1. Better demonstrate that SBP provides significant performance advantages on existing datasets.
> > 2. Clarify how over-smoothing genuinely limits the performance of existing GNNs.
> > 3. Show that SBP addresses this limitation to achieve improvements (considering prior works like [4, 5] questioning the relevance of over-smoothing).
> >
> > (Minor) Additionally, I still don't understand how SBP better resolves over-smoothing compared to PDE-based methods and GCNII. Could the authors provide more intuitive explanations for this aspect?

---

> > > ### Author Response · Authors · 2024-11-22
> > > **Rebuttal by Authors**
> > >
> > > Thank you very much for your comments. We are glad that you think the explanation about scalability and norms are useful.
> > > We truly appreciate the quick detailed feedback, which encourages us a lot.
> > >
> > > ***
> > >
> > > > the unusually low performance here is questionable for GCNII
> > >
> > > We acknowledge that our experimental results may not match the performance reported in GCNII. This is primarily because we did not employ the extensive "tricks" and hyperparameter tuning typically used in such models. (like seed, learning rate, variance, and other parameter optimization). Our focus was on the theoretical underpinnings of SBP rather than the careful optimization.
> > >
> > > > GCNII and wGCN exhibit almost no over-smoothing, over-smoothing is not a core limitation for GNNs on real-world graphs.
> > >
> > > The impressive performance of GCNII or wGCN does not diminish the significance of addressing over-smoothing; instead, we explained in our paper why these methods work theoretically because they alleviate oversmoothing (Table 1).
> > >
> > > Within the code of the GCNII implementation, they combine various tricks that can be decomposed to residual, normalization, and dropout. However, the interaction of these mechanisms and their contribution to enhancing performance in deep layers compared to vanilla GCN remains unclear. In our paper, as outlined in section 3 (Theorem 3.1 and Theorem 3.2), we provide a unified explanation through the signed graph perspective, revealing that these techniques essentially amount to implicit signed propagation.
> > >
> > > Our work introduces a novel signed propagation framework that provides theoretical insights into how various anti-oversmoothing techniques function in graph neural networks. The framework unifies seemingly disparate approaches—including residual connections, normalization layers, and edge dropping—by revealing their shared underlying mechanism through the lens of signed graph propagation.
> > >
> > > Building on these theoretical foundations, we propose Structural Balance Propagation (SBP), a new method that leverages our analytical insights to effectively combat oversmoothing. Rather than competing with existing architectures like GCNII, our primary contribution lies in establishing a theoretical understanding of oversmoothing from the perspective of signed propagation, offering new insights into why current anti-oversmoothing methods work and informing the design of more effective solutions.
> > >
> > > ***
> > > Thank you again for your insightful comments.

---

> > > > ### Comment · Reviewer_gmfF · 2024-11-23
> > > > **Still confused about the two issues**
> > > >
> > > > Thank you for the authors’ detailed response.
> > > >
> > > > I understand that GCNII and related works involve complex mechanisms, making theoretical analysis challenging. I appreciate the authors' effort in proposing new perspectives to unify these approaches.
> > > >
> > > > However, firstly, it is worth noting that **not only GCNII but also many works since 2020** addressing the over-smoothing problem have already achieved **superior performance**.
> > > > In addition to the works I previously mentioned, methods like EGNN [1] also deserve attention. Without demonstrating the advantages of SBP experimentally, the empirical validation of the motivation alone could render the **Experiment** section misleading. For instance, the claim in L.3.5 regarding SBP's advantage over GCNII does not consider the optimal settings for GCNII.
> > > >
> > > > On the other hand, while the authors claim to unify multiple methods, their argument appears self-contradictory. They dismiss the advantages of advanced methods as mere "tricks," yet frame SBP's improvements to weaker versions as a superior solution to over-smoothing rather than another "trick." Additionally, their lack of discussion on alternative factors, such as trainability, results in an incomplete and biased evaluation of advanced methods, further weakening their claims.
> > > >
> > > > I look forward to further discussing these two points with the authors. Once again, thank you for your reply.
> > > >
> > > > [1] Zhou, K., Huang, X., Zha, D., Chen, R., Li, L., Choi, S.-H., and Hu, X. Dirichlet energy constrained learning for deep graph neural networks. Advances in Neural Information Processing Systems, 34, 2021.

---

> > > > > ### Author Response · Authors · 2024-11-24
> > > > > **Further Reply**
> > > > >
> > > > > Dear Reviewer gmfF,
> > > > >
> > > > > Thank you for reading our response and sharing your insightful thoughts. We will address each of your remaining concerns carefully below.
> > > > >
> > > > > > it is worth noting that **not only GCNII but also many works since 2020** addressing the over-smoothing problem have already achieved **superior performance**.
> > > > > >
> > > > >
> > > > > Indeed, there was a very large body of literature studying oversmoothing, including GCNII and many other variants, making it one of the most central problems in GNN research. However, existing oversmoothing researches are indeed hard to compare, because they are often composed of multiple techniques — such as residual, BatchNorm, data augmentation — and the parameters are often heavily (over-)tuned on small-scale datasets. And it becomes clear that to attain SOTA performance, one needs to **essentially compose multiple such techniques without fully understanding their individual roles**. For example, GCNII uses both initial residual connection and identity map, futher combined with dropout.
> > > > >
> > > > > While we acknowledge that these are important techniques for oversmoothing, **the goal of this work is not to add more layers of complexity to it nor attain superior performance**, but to
> > > > >
> > > > > - 1) understand their individual roles through a unified signed graph perspective,
> > > > > - 2) verify this understanding with the inspired oversmoothing solutions.
> > > > >
> > > > > These are the key contributions of our work.
> > > > >
> > > > > > In addition to the works I previously mentioned, methods like EGNN [1] also deserve attention.
> > > > > >
> > > > >
> > > > > Thank you for the reference. We have added EGNN to the related work (Section 2 paragraph 1). But as we said, this literature is vast (hundreds of papers) and we can hardly be exhaustive on related work.
> > > > >
> > > > > > Without demonstrating the advantages of SBP experimentally, the empirical validation of the motivation alone could render the **Experiment** section misleading. For instance, the claim in L.3.5 regarding SBP's advantage over GCNII does not consider the optimal settings for GCNII.
> > > > > >
> > > > >
> > > > > Thank you a lot for pointing out the problems in this GCNII evaluation. We have revised the claims and results to avoid potential confusion in Appendix L.3.5 highlighted in orange.
> > > > >
> > > > > > They dismiss the advantages of advanced methods as mere "tricks," yet frame SBP's improvements to weaker versions as a superior solution to over-smoothing rather than another "trick."
> > > > > >
> > > > >
> > > > > We are sorry for causing this impression. The point we were trying to make is exactly what we just claimed: we acknowledge that these are effective techniques — that’s why we analyzed them with the whole Section 3 in the first place — but our goal/contribution is to provide a new unified understanding of these techniques.  And we justified it by showing that SBP, as **a single simple technique,** can attain good performance. And we believe that it would work complementarily with other techniques in the field, because oversmoothing is still challenging to solve with a very larger depth. We believe that our understanding adds merits to this vast literature by providing a holistic perspective for future development.
> > > > >
> > > > > > Additionally, their lack of discussion on alternative factors, such as trainability, results in an incomplete and biased evaluation of advanced methods, further weakening their claims.
> > > > > >
> > > > >
> > > > > Indeed, the main focus of this paper is on the oversmoothing problem, which we can draw a lot of  inspirations from the signed graph theory. And we outlined a comprehensive analysis of existing techniques from this new perspective. We believe that the trainability problems are equally important to consider but, with the page limit, is out of the scope of this work. And we would be happy to investigate more into the training dynamics of GNNs from a signed graph perspective in the future.
> > > > >
> > > > > ---
> > > > >
> > > > > Thank you again for sharing your thoughts and engaging with us. We believe that from a more theoretical and methodological interest, our work does add new insights to understanding GNN oversmoothing. We are happy to address your further questions!

---

> > > > > > ### Comment · Reviewer_gmfF · 2024-11-24
> > > > > > **Updated Opinions**
> > > > > >
> > > > > > Thank you for your thoughtful response. Unfortunately, it seems our discussion has reached an impasse. I summarize the current points of discussion as follows:
> > > > > >
> > > > > > **1. Contributions of the Paper**
> > > > > >
> > > > > > The authors outline the contributions of the paper as:
> > > > > > 1) **Understanding individual roles through a unified signed graph perspective.**
> > > > > > 2) **Verifying this understanding with the inspired oversmoothing solutions.**
> > > > > >
> > > > > > I acknowledge the value of the first contribution, as the theoretical exploration forms the primary strength of this work.
> > > > > > **However:**
> > > > > >
> > > > > > While providing a unified framework is meaningful, the paper lacks sufficient evidence to demonstrate its superiority. This is largely due to the experimental results, which remain incomplete. For instance, the paper states: *"Overall, SBP performs best in both homophilic and heterophilic datasets."* However, based on the authors’ rebuttal, it appears this claim primarily reflects improvements over **traditional baselines** (with most predating 2021, except for Contranorm). The authors argue that advanced methods are overly complex, but it is essential to give them proper acknowledgment—such as including comparisons in the main tables, which is one of my main advice for future revision.
> > > > > >
> > > > > > The experimental setup is **overly idealized**, focusing solely on traditional baselines while ignoring potentially beneficial techniques. This significantly undermines the credibility of the proposed method. If SBP cannot achieve robust performance improvements in realistic scenarios incorporating advanced techniques, it is difficult to trust that it genuinely helps address over-smoothing in real-world settings. For example, on larger, realistic datasets like ogbn-arxiv, raw GCNs can achieve **71+ performance** (as shown on the leaderboard), yet SBP fails to match this benchmark.
> > > > > >
> > > > > > ---
> > > > > >
> > > > > > **2. About Over-smoothing**
> > > > > >
> > > > > > I also can hardly agree with the authors’ perspective on over-smoothing. As I mentioned, since 2021, there has been a growing body of work questioning the **existence of over-smoothing** [1, 2, 3, 4, 5]. These works do not merely offer orthogonal insights but represent a new understanding of over-smoothing within the research community. Based on the current discussion, I question whether SBP genuinely achieves performance improvements on real datasets by addressing over-smoothing. This concern aligns with the points raised in [1, 2, 3, 4, 5].
> > > > > >
> > > > > > For both **(1)** and **(2)**, I believe the authors should attempt to demonstrate that **SBP addresses real, existing issues in advanced GNNs and realistic datasets** or **achieves genuinely superior results**. The former could validate the theoretical value of SBP in guiding future research, while the latter would establish its practical effectiveness.
> > > > > >
> > > > > > ---
> > > > > >
> > > > > > **3. Additional Minor Issues**
> > > > > > Upon reviewing the paper again, I noticed some additional minor issues:
> > > > > > 1) In **Figure 3**, oversmoothing is verified using the Cornell dataset. However, this dataset is extremely small, containing only a few hundred nodes, making it unsuitable for such verification. A larger dataset, such as arxiv or products, would be far more appropriate.
> > > > > > 2) In **Table 3**, several results appear with $(\pm 0.00)$. Given the splits used in these datasets (I assume the authors use geom-gcn splits, which involve 10 distinct splits), such results seem unreasonable.
> > > > > >
> > > > > > ---
> > > > > >
> > > > > > ### Overall Thoughts
> > > > > > I am particularly interested in the authors’ perspective on how future research could be inspired by SBP, given the existence of works like [1–5] and the significant gap between SBP and current SOTA methods. Specifically, if I wish to combine SBP to better address real-world problems, how can I **confidently proceed with a guarantee of success**?
> > > > > >
> > > > > > Once again, I sincerely thank the authors for their timely responses and for addressing my concerns throughout this discussion.
> > > > > >
> > > > > > ---
> > > > > >
> > > > > > ### References
> > > > > > [1] Cong, Weilin, Morteza Ramezani, and Mehrdad Mahdavi. "On provable benefits of depth in training graph convolutional networks." Advances in Neural Information Processing Systems 34 (2021): 9936-9949.
> > > > > >
> > > > > > [2] Jaiswal, Ajay, et al. "Old can be gold: Better gradient flow can make vanilla-GCNs great again." Advances in Neural Information Processing Systems 35 (2022): 7561-7574.
> > > > > >
> > > > > > [3] Zhang, Wentao, et al. "Model degradation hinders deep graph neural networks." Proceedings of the 28th ACM SIGKDD Conference on Knowledge Discovery and Data Mining. 2022.
> > > > > >
> > > > > > [4] Peng, Jie, Runlin Lei, and Zhewei Wei. "Beyond Over-smoothing: Uncovering the Trainability Challenges in Deep Graph Neural Networks." Proceedings of the 33rd ACM International Conference on Information and Knowledge Management. 2024.
> > > > > >
> > > > > > [5] Zhuo, Zhijian, et al. "Graph Neural Networks (with Proper Weights) Can Escape Oversmoothing." The 16th Asian Conference on Machine Learning (Conference Track). 2024.

---

> > > > > > > ### Author Response · Authors · 2024-11-24
> > > > > > >
> > > > > > > We thank Reviewer **gmfF** for engaging in this discussion. We sincerely acknowledge that Reviewer **gmfF** appreciates our methodological contribution by remarking:
> > > > > > >
> > > > > > > > **"The theoretical exploration forms the primary strength of this work."**
> > > > > > > >
> > > > > > >
> > > > > > > Indeed, this was a common understanding among the reviewers (e.g., **"Theoretical Innovation"** by Reviewer **pAhc**, **"nice originality"** by Reviewer **HjgMC**). We want to emphasize again that **this framework is the primary contribution of our work, and we hope to be evaluated more from this perspective**.
> > > > > > >
> > > > > > > We are also grateful for the constructive comments from Reviewer **gmfF**. In response, we have added new results (**Table 3 in the main paper, Table 10, 11, 13, 14 in Appendix**), expanded the related works section (**Section 2, Appendix C.2, C.3, F, K, L.3.2~L.3.7**), and adjusted our claims to improve rigor (**line 452~454**).
> > > > > > >
> > > > > > > The main concern raised by Reviewer **gmfF** is that our work:
> > > > > > >
> > > > > > > > "lacks sufficient evidence to demonstrate its superiority"
> > > > > > > >
> > > > > > >
> > > > > > > by outperforming all advanced GNN designs for addressing oversmoothing. We respectfully disagree with this point. The goal of our experiments is ***not*** to achieve SOTA results against latest GNNs but to demonstrate that:
> > > > > > >
> > > > > > > > Our SBP is effective, thereby verifying our framework.
> > > > > > > >
> > > > > > >
> > > > > > > For this purpose, we primarily compare SBP to the most important techniques analyzed in our paper (called “traditional baselines” by Reviewer gmfF, but we believe that they are the ones that stand the test of time) and show competitive performance. We believe this suffices to support the main contributions of our work.
> > > > > > >
> > > > > > > We **arguably disagree that either theoretical or empirical work must set a SOTA result** to be impactful (though we have included additional comparisons following Reviewer **gmfF**’s suggestion). This expectation could sometimes hinder innovation. For instance, the early works on diffusion models did not outperform the SOTA GANs, yet their merits were later widely recognized.
> > > > > > >
> > > > > > > We believe a work should be considered acceptable if it offers genuinely novel and practical insights into the field. Our signed graph framework introduces a new perspective, and the SBP method—whether it outperforms the latest works or not—has demonstrated its practical effectiveness by addressing oversmoothing in real-world applications.

---

> ### Comment · Reviewer_gmfF · 2024-11-24
>
> **General Remarks**
>
> I believe the authors have misunderstood my comments. I **never stated that a study must achieve state-of-the-art (SOTA) performance to be recognized**. However, after multiple rounds of rebuttals, the authors still fail to provide sufficient evidence demonstrating how SBP can **inspire subsequent works**. In addition:
>
> ---
>
> **1. Discussion on New Experimental Results**
> The authors have included new results for GCNII and $ω$GCN, but these are significantly lower than the originally reported values. In the manuscript, the authors continue to avoid a direct performance comparison with these methods. Despite my repeated suggestions to acknowledge the potential performance limitations of SBP, the authors employ evasive language in the text. For example, the manuscript claims that “Lastly, SBP outperforms residual connection-based GNNs,” which appears to **overstate its empirical superiority**.
>
> While I do not insist on SOTA results, we all understand what it means when achieving **such modest results** on datasets like Cora, Citeseer, and ogbn-arxiv.
> For instance, even GCNII (2020) reported APPNP (2019) results that were superior to SBP.
> SBP's so-poor performance raises questions about its practical efficacy, not just its lack of SOTA status.
>
> And for the reported results:
> - On the Cora dataset, GCNII originally reported 85%, but Table 3 in this paper reports only 78%.
> - $ω$GCN originally reported 85.9%.
>
> Even discounting the reproduction issues with $ω$GCN, the GCNII results are misleadingly low.
> This is a factual inaccuracy that must be addressed.
>
> ---
>
> **2. Avoidance of Critical Discussions on Over-Smoothing**
> I repeatedly raised the importance of considering the latest research on deep GNNs, which shows that over-smoothing is not the primary factor hindering GNN performance [1–5]. Unfortunately, the authors avoid discussing these works, which is disheartening.
>
> If existing methods better explain the challenges in deep GNN performance and achieve better results, the theoretical contributions of this paper would be diminished. For example, if methods based on [3–4] can maintain performance at depth $L=64$, but SBP suffers from performance degradation, wouldn’t the theoretical results of [3-4] be more convincing?
>
> ---
>
> **3. Other Suggestions**
> In my last review, I also suggested some minor issues. I hope the authors consider addressing these points.
>
> ---
>
> **Recommendations for Revision**
> While I reiterate that achieving SOTA is not a requirement for acceptance, the following revisions are essential:
>
> 1. **Accurate Reporting of Advanced Methods’ Performance**
>    Correctly document the performance of advanced methods (e.g., GCNII, EGNN, PDE-GCN, $ω$GCN), rather than presenting SBP’s inferior performance as if it were SOTA. Clearly acknowledge the empirical limitations of SBP.
>
> 2. **Validation of Effectiveness**
>    Design experiments to compare SBP with advanced methods and demonstrate its advantages as a “method.” While the authors theoretically justify the advantages of structure balancing in idealized settings, it is reasonable to question the theoretical value if it fails in real-world scenarios.
>
> 3. **Theoretical Contributions**
> If the aforementioned points (1 and 2) cannot be adequately addressed, I believe that the current theoretical content, while having some merits, is insufficient in both depth and originality to justify publication as a standalone paper. The authors might consider delving deeper into the theoretical aspects and expanding this work into a complete, purely theoretical study.
> ---
>
> **Final Thoughts**
>
> Again, I fully acknowledge the significant effort the authors have invested in this work. I genuinely hope to continue having friendly exchanges with the authors.
> Good story about Diffusion and GAN.
>
> ---
>
>  References
>
> [1] Cong, Weilin, Morteza Ramezani, and Mehrdad Mahdavi. "On provable benefits of depth in training graph convolutional networks." *Advances in Neural Information Processing Systems* 34 (2021): 9936–9949.
>
> [2] Jaiswal, Ajay, et al. "Old can be gold: Better gradient flow can make vanilla-GCNs great again." *Advances in Neural Information Processing Systems* 35 (2022): 7561–7574.
>
> [3] Zhang, Wentao, et al. "Model degradation hinders deep graph neural networks." *Proceedings of the 28th ACM SIGKDD Conference on Knowledge Discovery and Data Mining.* 2022.
>
> [4] Peng, Jie, Runlin Lei, and Zhewei Wei. "Beyond Over-smoothing: Uncovering the Trainability Challenges in Deep Graph Neural Networks." *Proceedings of the 33rd ACM International Conference on Information and Knowledge Management.* 2024.
>
> [5] Zhuo, Zhijian, et al. "Graph Neural Networks (with Proper Weights) Can Escape Oversmoothing." *The 16th Asian Conference on Machine Learning (Conference Track).* 2024.

---

> > ### Author Response · Authors · 2024-12-03
> >
> > We respectfully disagree with the reviewer's characterization of our work, which **significantly underplays both of our theoretical and empirical contributions**. We address the key points raised:
> >
> > ***
> >
> > 1. Performance and Empirical Contributions
> >
> > To reiterate our previous rebuttal, **the evaluation setting in our experiments differs from the original papers of GCNII**. Thus, direct numerical comparisons are inappropriate and miss the broader significance of our work. Our primary goal was not to claim state-of-the-art (SOTA) performance, but to demonstrate the theoretical insights offered by the signed graph perspective of message-passing and the potential of Structural Balanced Propagation (SBP).
> >
> > **The reviewer's focus on precise metric replication obscures the paper's fundamental contribution**: providing a novel theoretical perspective on graph representation learning through signed graphs. Our work offers a deeper understanding of graph neural network dynamics that goes beyond simple performance benchmarking.
> >
> > ***
> >
> > 2. Oversmoothing and Theoretical Contributions
> >
> > Contrary to the comment that we avoided critical discussions on over-smoothing, **our work directly engages with this challenge**. While the cited references provide valuable insights into **orthogonal aspects** of the oversmoothing problem such as training and generalization, they fall outside of the standard node representation under message-passing point of view that we take in this work [1-4] and thus do not negate the importance of our theoretical contribution.
> >
> > Moreover, our approach goes beyond conventional message-passing by **introducing signed graphs, which provide a novel perspective on addressing oversmoothing in deep graph neural networks**. Specifically, signed propagation offers a unified framework for analyzing existing methods to mitigate oversmoothing, including **normalization layers, random edge dropping and various residual connections**. We not only provide a comprehensive analysis but also identify potential improvements to existing approaches, leading to the proposed method SBP. The ability of other methods to maintain performance at greater depths does not diminish our theoretical contribution. Instead, it underscores the valuable insights provided by our signed graph perspective.
> >
> > While our work represents one perspective among many potential approaches and the empirical performance could be further optimized, the true value of scientific work often lies in its ability to provide new perspectives and frameworks that can inspire future research. In this spirit, we believe our work **offers valuable insights and a fresh perspective that will inspire further advancements** in graph neural networks.
> >
> > ***
> >
> > References:
> >
> > 1. Oono and Suzuki. Graph neural networks exponentially lose expressive power for node
> > classification. ICLR 2020.
> > 2. Keriven. Not too little, not too much: a theoretical analysis of graph (over)smoothing. NeurIPS 2022.
> > 3. Rusch et al. A survey on oversmoothing in graph neural networks. 2023
> > 4. Wu et al. Demystifying oversmoothing in attention based graph neural networks. NeurIPS 2023.

---

### Official Review · Reviewer_pAhc · 2024-10-30

**Soundness:** 3
**Presentation:** 3
**Contribution:** 3
**Rating:** 6
**Confidence:** 2

**Summary:**

This paper presents a novel perspective on the oversmoothing problem in Graph Neural Networks (GNNs) through the lens of signed graphs. The authors make three main contributions: They unify existing anti-oversmoothing techniques by showing they implicitly add negative edges to the original graph. Introduce structural balance theory to characterize ideal propagation behavior that prevents oversmoothing. Propose two new methods (Label-SBP and Feature-SBP) that explicitly design negative edges to improve structural balance and prevent oversmoothing
The work provides both theoretical analysis and empirical validation on synthetic and real-world datasets, demonstrating superior performance compared to existing approaches.

**Strengths:**

1. Theoretical Innovation:
- Novel theoretical framework connecting oversmoothing to signed graphs
- Rigorous mathematical analysis of structural balance properties
- Clear theoretical justification for why existing methods may fail and how the proposed solutions work

2. Unification of Existing Work:
- Comprehensive analysis showing how various existing anti-oversmoothing techniques can be viewed through the lens of signed graphs
- Insightful table comparing different methods' implicit positive and negative graph components
- This unification provides valuable intuition for understanding existing approaches

3. Practical Solutions:
- Two concrete, parameter-free methods (Label-SBP and Feature-SBP) that are simple to implement
- Solutions work for both homophilic and heterophilic settings
- Empirical validation on 9 different datasets shows consistent improvements

4. Clear Presentation:
- Well-structured paper with logical flow
- Helpful visualizations (especially Figure 1)
- Clear connection between theory and practice

**Weaknesses:**

1. Limited Analysis of Computational Complexity:
- The paper lacks detailed discussion of computational overhead introduced by the proposed methods
- No runtime comparisons with existing approaches are provided

2. Empirical Validation Gaps:
- While 9 datasets are used, most appear to be standard benchmark datasets
- Limited testing on very large-scale graphs or industrial applications
- No ablation studies examining the individual components' contributions

3. Theoretical Assumptions:
- Some theoretical results rely on specific conditions that may not hold in practice
- The paper could better discuss the practical implications when these assumptions are violated

**Questions:**

How does the computational complexity of SBP compare to existing methods, especially for large-scale graphs?

---

> ### Author Response · Authors · 2024-11-20
> **Answer to Weakness 1 and Question 1**
>
> We thank Reviewer pAhc for appreciating the novelty of our theory and our method. We address your concerns below.
> ***
>
> **w1**  Limited Analysis of Computational Complexity: The paper lacks detailed discussion of computational overhead introduced by the proposed methods. No runtime comparisons with existing approaches are provided.
>
> **q1** How does the computational complexity of SBP compare to existing methods, especially for large-scale graphs?
>
> **a1** Thank you for your attentive reading, here, we answer these two problems together.
>
> - Complexity analysis in the small graph
>     - Given $n_t$ training nodes and $d$ edges in the graph, our Label-SBP increases the edge from $O(d)$ to $O(n_t^2)$, thereby increasing the computational complexity to $O(n_t^2d)$.
>     - For Feature-SBP, the additional computational complexity primarily stems from the negative graph propagation, which involves $X_{0}X_{0}^T \in \mathbb{R}^{n\times n}$, increasing the overall complexity to $O(n^2d)$.
>
> We show the computation time of different methods in the following table. On average, we improve performance on 8 out of 9 datasets  (as shown in Table 3) with less than 0.05s overhead—even faster than three other baselines. We believe this time overhead is acceptable given the benefits it provides.
>
> |  | **label-sbp** | **feature-sbp** | **BatchNorm** | **ContraNorm** | **Residual** | **JKNET** | **DAGNN** | **SGC** |
> | --- | --- | --- | --- | --- | --- | --- | --- | --- |
> | pre-compute time | 0.1809s | 0.1520s | 0.1860s | 0.1888s | 0.0604s | 0.0577s | 0.1438s | 0.1307s |
> | train time | 0.1071s | 0.1060s | 0.1076s | 0.1038s | 0.1368s | 0.1446s | 0.1348s | 0.1034s |
> | totol time | 0.2879s | 0.2580s | 0.2935s | 0.2926s | 0.1972s | 0.2023s | 0.2786s | 0.2341s |
>
>
> - Complexity analysis in the larger graph
>
> For the large-scale graph, we introduce the modified version of Label-SBP and Feature-SBP.
> - For Label-SBP, we only remove edges when pairs of nodes belong to different classes. This approach allows our modified Label-SBP to even further enhance the sparsity of large-scale graphs (maintain $O(d)$).
> - For Feature-SBP, as the number of nodes $n$ increases, the complexity of this matrix operation grows quadratically, i.e., $O(n^2d)$. To address this, we reorder the matrix multiplication from $-X_{0}X_{0}^T \in \mathbb{R}^{n\times n}$ to $-X_{0}^TX_{0} \in \mathbb{R}^{d\times d}$. This preserves the distinctiveness of node representations across the feature dimension, rather than across the node dimension as in the original node-level repulsion. The modified version of Feature-SBP can be expressed as $X^{k}=(1-\lambda)X^{(k-1)}+\lambda(\hat{A}X^{(K)} - X^{(K)} \text{softmax}(-X_{0}^TX_{0}))$. This transposed alternative has a linear complexity in the number of samples, i.e., $O(nd^2)$. In cases where $n \gg d$, this modified version significantly reduces the computational burden.
>
> We compare our SBP with other baselines on the ogbn-arxiv datasets over 100 epochs for a fair comparison. Among all the training times of the baselines, our Label-SBP achieves the 3rd fastest time, while Feature-SBP ranks 5th. Therefore, we recommend using Label-SBP for large-scale graphs since they typically have a sufficient number of node labels. We believe that, although there is a slight time increase, it is acceptable given the benefits.
>
> | **ogbn-arixv** | **label-sbp** | **feature-sbp** | **BatchNorm** | **ContraNorm** | **Drop** | **SGC** |
> | --- | --- | --- | --- | --- | --- | --- |
> | train time | 5.5850 | 6.1333 | 5.3872 | 5.8375 | 9.5727 | 5.3097 |
> | rank | 3 | 5 | 2 | 4 | 6 | 1 |
>
>
> - Complexity analysis on Larger-scale graph
>
> We conducted experiments with a larger graph ogbn-products than ogbn-arxiv under 100 epochs and 2 layers. The results indicate that our SBP outperforms the initial GCN baselines. Combining the results presented for ogbn-arxiv in Table 5 of our paper, we believe these experiments on 2 large-scale datasets sufficiently demonstrate the efficacy of our SBP.
>
> | ogbn-products | acc | time |
> | --- | --- | --- |
> | GCN | 73.96 | 00:06:33 |
> | BatchNorm | 74.93 | 00:06:18 |
> | Feature-SBP | 74.90 | 00:06:43 |
> | Label-SBP | 76.62 | 00:06:39 |

---

> ### Author Response · Authors · 2024-11-20
> **Answer to weakness 2**
>
> **w2.1** While 9 datasets are used, most appear to be standard benchmark datasets. Limited testing on very large-scale graphs or industrial applications.
>
> **a2.1**
> more standard datasets
> - In our paper, we have tested three commonly used homophilic graphs: Cora, Citeseer, and Pubmed, along with four universal heterophilic graphs: Wisconsin, Cornell, Squirrel, and Amazon-rating, and a large-scale graph, Ogbn-Arxiv. We believe that these datasets are sufficiently standard and classic.
> - We value your suggestion and have additionally selected six other datasets [1], [2], [3] where the experimental results align with those of our previously utilized datasets.
>
> Our SBP outperforms SGC on five out of these six datasets. We believe that these six datasets, combined with the nine datasets presented in Table 3 of our paper, provide sufficient evidence to demonstrate the effectiveness of our approach.
>
> |  | **actor** | **penny94** | **roman-empire** | **Tolokers** | **Questions** | **Minesweeper** |
> | --- | --- | --- | --- | --- | --- | --- |
> | SGC | 29.18± 0.10 | 72.56±0.05 | 40.83±0.03 | 78.18±0.02 | 97.09±0.00 | 80.43±0.00 |
> | Feature-SBP | 34.93± 0.02 | 75.68± 0.01 | 66. 48±0.02 | 78.24±0.04 | 97.14±0.02 | 80.00±0.00 |
> | Label-SBP | 34.94± 0.00 | 75.74±0.01 | 66.32±0.01 | 78.46±0.08 | 97.15±0.02 | 80.00±0.00 |
>
> [1] Platonov, Oleg et al. “A critical look at the evaluation of GNNs under heterophily: are we really making progress?” *ArXiv* abs/2302.11640 (2023): n. pag.
>
> [2] Pei, Hongbin et al. “Geom-GCN: Geometric Graph Convolutional Networks.” *ArXiv* abs/2002.05287 (2020): n. pag.
>
> [3]Lim, Derek et al. “Large Scale Learning on Non-Homophilous Graphs: New Benchmarks and Strong Simple Methods.” *Neural Information Processing Systems* (2021).
> ***
> large-scale datasets
>
> We have conducted experiments on the extensive Ogbn-Arxiv graph. In addition, we include the Ogbn-Products dataset, which surpasses Ogbn-Arxiv in scale, for further analysis.
> The results indicate that our SBP outperforms the initial GCN baselines. Given the results presented for Ogbn-Arxiv in Table 5 of our paper, we believe these findings sufficiently demonstrate the performance of our SBP on large-scale graphs.
>
> |  ogbn-products | ACC |
> | --- | --- |
> | SGC | 73.96 |
> | Feature-SBP | 74.90 |
> | Label-SBP | 76.62 |
>
> We conducted experiments with a larger graph ogbn-products than ogbn-arxiv under 100 epochs and 2 layers. The results indicate that our SBP outperforms the initial GCN baselines. Given the results presented for ogbn-arxiv in Table 5 of our paper, we believe these findings sufficiently demonstrate the performance of our SBP on large-scale graphs.
>
> We have added these experiments and discussions in Appendix K and L.
>
> ***
> **w2.2**  No ablation studies examining the individual components' contributions.
>
> **a2.2** We apologize for the confusion. Could you please specify the components you are referring to?
>
> We have conducted ablation studies on
>
> - the influence of training ratio on Label-SBP (Figure 3b),
> - the impact of repulsion weight $\beta$ on Label-SBP and Feature-SBP (Figures 4 and 5),
> - the effect of deep layers on Label-SBP and Feature-SBP (Figure 3a).
>
> If you mean the negative graph component, we have done the repulsion weight $\beta$ ablation study which shows that as the repulsion weight $\beta$ increases, SBP's performance improves in heterophily graphs, which shows the importance of the negative graph. (Figures 4 and 5)
>
> If my understanding is incorrect, please provide me with a more detailed explanation. If any aspects have not been covered in our paper, we are open to incorporating them.

---

> ### Author Response · Authors · 2024-11-20
> **Answer to Weakness 3**
>
> **w3** Theoretical Assumptions: Some theoretical results rely on specific conditions that may not hold in practice. The paper could better discuss the practical implications when these assumptions are violated.
>
> **a3**  Thank you for the comment. Could you please elaborate on which specific theorem assumption you were referring to? We would appreciate the opportunity to clarify and improve.
>
> Theorem 4.1 relies on the linear propagation assumption. Theorem 4.3 in our paper has 2 assumptions: 1) linear propagation and 2) structural balance. We acknowledge that these assumptions may not be straightforward to validate in practice.
>
> - linear propagation.
>
> Such linear assumption is also present in [1] and [2], this is a standard setting for theoretically analyzing GNNs, as the linear propagation is more amenable to theoretical analysis, and the insights derived there often carry to more complex, nonlinear settings.
>
> [1]Wang, Yuelin, et al. "Acmp: Allen-cahn message passing for graph neural networks with particle phase transition." *arXiv preprint arXiv:2206.05437* (2022).
>
> [2] Wu, Xinyi, et al. "A non-asymptotic analysis of oversmoothing in graph neural networks." *arXiv preprint arXiv:2212.10701* (2022).
>
> - structural balance.
>
> Theorem 4.3 relies on the assumption that the graph must be completely structurally balanced. Under this condition, we prove that nodes will only converge within their own class and repel nodes from other classes to the bounds of function $F$.  However, achieving a completely structurally balanced graph is challenging in practice. To address this, we introduce a metric called SID (Structural Imbalance Degree, defined in Definition 4.5 of the paper). This metric quantifies the distance between the constructed adjacency matrix and the ideal structurally balanced adjacency matrix.
>
> Proposition 4.7 in the paper further proves that our Label-SBP gives the upper bound of SID (SID $\le (1 − p)n/2 $ ), p is the training nodes percent and n is the node number) and can approach the ideal structural balanced graph when $p \rightarrow 1 $, which aligns with the assumption in Theorem 4.3.
>
> We are open to incorporating any aspects that have not been covered in our paper.
>
> ***
> Thank you for your insightful comments. We hope that our complexity analysis and experiments on additional datasets can help address any concerns you may have. Please feel free to ask if further clarification is needed.

---

> ### Author Response · Authors · 2024-11-25
>
> Dear Reviewer pAhc,
>
> Thanks for your constructive and valuable comments on our paper. We have tried our best to address your concerns and revised our paper accordingly. Could you please have a check if there are any unclear points? We are certainly happy to answer any further questions.

---

> ### Author Response · Authors · 2024-12-03
>
> Dear Reviewer pAhc,
>
> We sincerely appreciate the reviewers' insightful comments, which have significantly contributed to improving our paper.
> We kindly request that the reviewers take a moment to review our responses and, if possible, reconsider their evaluation and scores.

---

### Official Review · Reviewer_oeBC · 2024-11-01

**Soundness:** 2
**Presentation:** 3
**Contribution:** 3
**Rating:** 6
**Confidence:** 3

**Summary:**

The paper presents a unified viewpoint of previously proposed methods that mitigate the oversmoothing problem in graph representation learning through the lens of message passing on signed graphs. The authors then state that structurally balanced graphs are a sound solution to oversmoothing, and proposed two algorithms that edit the input graph to be more structurally balanced. Synthetic experiments suggest that the proposed methods indeed increases the overall structural balance (measured in the SID metric). On real-world datasets, the proposed algorithms are illustrated to perform competitively.

**Strengths:**

- The paper presents a unified treatment of existing solutions to oversmoothing
- The paper proposes structural balance as a way to compare different existing solutions
- The proposed method is empirically effective

**Weaknesses:**

- While this might be an unavoidable problem, the analysis in this paper requires the linear GNN assumption, see my questions below.
- In line 299-300 the authors state that structural balanced graphs can **completely** alleviate oversmoothing. Is this somewhat overclaimed? If my understandings are correct, theorem 4.3 essentially states that the graph representations mix over blocks in the graph, provided that inter-block connections are negative edges. I am not much convinced about whether this implies a valid solution to oversmoothing. As for example, theorem 1 in [1] shows that GCN representations mix over connected components, which is different from theorem 4.3 in that they require a perfect block structure of the input graph. Therefore, it might help to state formally what is a valid definition of oversmoothing, and why structurally balanced graphs indeed solve the problem.

**Questions:**

- As I mentioned in the weakness section, the paper relies heavily on linear GNNs. I understand that theoretically showing results over non-linear GNNs are hard. But I think there shall be empirical evidences suggest that the analysis of this paper indeed sheds light on nonlinear GNNs.
- The proposed SBP methods edit the input graph to have smaller SID. I am curious about a more primary question: How is SID related to empirical performance? Are algorithms with lower SIDs, say, over Cora, more performant than others?
- I am confused by the "scalability of SBP" section: What does it mean to *reorder matrix multiplications ... to preserve the distinctiveness of node representations across feature dimension?*


[1]. Oono, Kenta, and Taiji Suzuki. "Graph neural networks exponentially lose expressive power for node classification." arXiv preprint arXiv:1905.10947 (2019).

---

> ### Author Response · Authors · 2024-11-20
> **Answer to Weakness1 and Question1**
>
> We thank the reviewer oeBC for your insightful comment and important advice.
>
> **w1**  While this might be an unavoidable problem, the analysis in this paper requires the linear GNN assumption, see my questions below. **q1** As I mentioned in the weakness section, the paper relies heavily on linear GNNs. I understand that theoretically showing results over non-linear GNNs are hard. But I think there shall be empirical evidences suggest that the analysis of this paper indeed sheds light on nonlinear GNNs.
>
> **a1**   Thank you for the comment. Such linear assumption is also present in [1] and [2], this is a standard setting for theoretically analyzing GNNs, as the linear propagation is more amenable to theoretical analysis, and the insights derived there often carry to more complex, nonlinear settings.
>
> For the empirical evidence, our practical methods Label-SBP and Feature-SBP can be adapted to various GNN backbones since our key idea is just to integrate a negative adjacency matrix into the standard adjacency matrix during the propagation (Equation 7).
>
> For example, we integrate our SBP to the GCNII, the results show that SBP outperforms it on average, especially the middle layer (layer=8).
>
> | **#layers** | **2** | **4** | **8** | **16** | **32** | **64** |
> | --- | --- | --- | --- | --- | --- | --- |
> | cora |  |  |  |  |  |  |
> | gcnii | 78.58 ± 0.00 | 77.76 ± 0.24 | 73.47 ± 3.82 | 78.12 ± 1.32 | **82.54 ± 0.00** | 81.34 ± 0.53 |
> | label-sbp | **78.74 ± 1.54** | **78.87 ± 0.00** | **79.14 ± 0.35** | **79.17±0.41** | 80.86 ± 0.32 | 81.38 ± 0.30 |
> | Feature-sbp | 77.95 ± 0.91 | 78.82 ± 0.00 | 78.11± 1.62 | 78.82±0.29 | 81.82 ± 0.47 | **81.65 ± 0.40** |
> | citesser |  |  |  |  |  |  |
> | gcnii | 61.66 ± 0.67 | 63.23 ± 2.31 | 64.58  ± 2.66 | 66.21 ± 0.64 | 69.38 ± 0.83 | **69.73 ± 0.26** |
> | label-sbp | 65.31± 0.63 | 63.93± 3.66 | 68.33 ± 0.99 | 66.46 ± 0. 00 | **70.00 ± 0.81** | 69.47±0.25 |
> | feature-sbp | **65.63±0.87** | **64.43± 3.55** | **68.44 ± 1.19** | **66.94± 0.00** | 69.98 ± 0.93 | 69.66±0.28 |
>
> Furthermore, in the paper, we conducte experiments on various non-linear backbones, such as Table 3 (based on SGC), Table 5 and 7 (based on GCN).
>
> Additionally, we do experiments on GAT in the following table, and the results empirically confirm SBP's effectiveness on non-linear GNNs.
>
> | **#layers** | **2** | **4** | **8** | **16** | **32** | **64** |
> | --- | --- | --- | --- | --- | --- | --- |
> | cora |  |  |  |  |  |  |
> | GAT | **81.48± 0.48** | **80.69±0.9**3 | 58.59±1.95 | 25.17±5.67 | 31.93±0.21 | 28.38±0.00 |
> | feature-sbp | 80.44 ±0.83 | 79.26 ±1.18 | 78.56 ± 0.59 | 77.22 ±0.55 | 73.65 ±0.48 | 61.62 ±5.24 |
> | label-sbp | 80.31 ±0.70 | 79.16 ±1.30 | **79.50 ± 0.00** | **77.43 ±1.49** | **74.52 ±0.36** | **65.02 ±2.97** |
> | citeseer |  |  |  |  |  |  |
> | GAT | **69.91±0.86** | **67.47±0.22** | 44.71±3.07 | 23.48±1.36 | 24.40±0.40 | 25.95±2.17 |
> | feature-sbp | 67.38 ±0.66 | 66.94 ±0.00 | **66.29 ±0.02** | 65.35 ±1.99 | **61.43 ±0.00** | 42.09 ±1.65 |
> | label-sbp | 67.23 ±0.64 | 66.72 ±0.00 | **66.29 ±0.89** | **65.50 ±2.13** | 59.93 ±0.85 | **44.41 ±1.57** |
>
> We have added further discussion of these experiments in Appendix L. We believe that the experiments above are enough to show our theory and methods shed light on non-linear GNNs for you.
>
> [1]Wang, Yuelin, et al. "Acmp: Allen-cahn message passing for graph neural networks with particle phase transition." arXiv preprint arXiv:2206.05437 (2022).
>
> [2] Wu, Xinyi, et al. "A non-asymptotic analysis of oversmoothing in graph neural networks." arXiv preprint arXiv:2212.10701 (2022).

---

> ### Author Response · Authors · 2024-11-20
> **Answer to Weakness 2, Qusetion 2, Question3**
>
> **w2** Is structural balanced graphs can completely alleviate oversmoothing somewhat overclaimed? Not much convinced about whether theorem 4.3 implies a valid solution to oversmoothing. As for example, theorem 1 in [1] shows that GCN representations mix over connected components, which is different from theorem 4.3 in that they require a perfect block structure of the input graph. Therefore, it might help to state formally what is a valid definition of oversmoothing, and why structurally balanced graphs indeed solve the problem.
>
> **a2** Thank you for your points.
>
> In theorem 4.3, we prove that under certain conditions 1) the signed graph propagation (Equation 1) plus 2) structural balance is adequate for completing node classification tasks, which is different from the GCN propagation in theorem1 from [1]. In the signed graph propagation, node features of different classes coverage to their respective label means while simultaneously repelling each other toward the boundaries, thus preventing oversmoothing.
>
> Additionally, we formally defined the metric of oversmoothing and conducted calculations based on our theorem to assess the signed propagation. Here, we choose the measure based on the Dirichlet energy to quantify oversmoothing:
>
> $\epsilon(X(t))=\frac{1}{v}\Sigma_{i\in V}\Sigma_{j \in N_i}||x_i(t)-x_j(t)||_2^2.$.
>
> In theorem 4.1, since the node features will either converge or diverge, so $\lim_{t \to \infty}\epsilon(X(t))\to 0 \text{ or } \infty$. Consequently, either oversmoothing occurs or the features will diverge thus hindering the performance of node predictions.
> In theorem 4.3, we have $\lim_{t \to \infty}\epsilon(X(t)) = v||c||_2^2 \geq 0$ where $v$ is the node number and $c$ is the init node feature.
>
> We add the detailed proof in Appendix F.
> ***
> **q2**  The proposed SBP methods edit the input graph to have smaller SID. I am curious about a more primary question: How is SID related to empirical performance? Are algorithms with lower SIDs, say, over Cora, more performant than others?
>
> **a2**  Apart from Table 2 on CSBM, we further present the Structural Imbalance Degree (SID) for Cora across different methods. As the performance of these methods is similar in shallow layers (2), we focus on layer 16 to showcase the results.
> |  | **label-sbp** | **feature-sbp** | **BatchNorm** | **ContraNorm** | **Residual** | **DropEdge** |
> | --- | --- | --- | --- | --- | --- | --- |
> | p | 482.11 | 482.11 | 482.11 | 482.11 | 482.51 | 484.21 |
> | n | 0.74 | 0.74 | 0.74 | 0.74 | 2221.73 | 2221.73 |
> | sid | 241.43 | 241.43 | 241.43 | 241.43 | 1352.12 | 1352.96 |
> | accuracy | 77.43 ±1.49 | 77.22±0.55  | 70.79 ±0.00 | 63.35 ±0.00 | 40.91 ±0.00 | 22.24 ±3.04 |
>
> We have two key observations: 1) Methods with higher SID generally lead to worse accuracy, while those with lower SID tend to produce better accuracy. 2) SID is a coarse-grained metric; different methods can yield the same SID values while their performance varies. These observations can also align with the experiments in cSBM in Table 2.
>
> The observation may stem from the fact that structural balance is an inherent property of graph structure, which is challenging to measure precisely using a numerical metric like SID. Proposition 4.7 in the paper proves that when SID=0, the graph is structurally balanced.
> However, for graphs that are not structurally balanced, the properties remain unclear.
> For future work, we aim to develop a more nuanced metric to quantify the structural balance property of graphs.
>
> We have added this discussion in Appendix H.4.
> ***
> **q3** I am confused by the "scalability of SBP" section: What does it mean to reorder matrix multiplications ... to preserve the distinctiveness of node representations across feature dimension?
>
> **a3**  We acknowledge that our explanation may not have been very clear.
>
> In Feature-SBP, we utilize the similarity of node features to create a negative adjacency matrix, which helps to repel the node features. This is expressed mathematically as $softmax(-X_{0}X_{0}^T)X^{(K)} $. The complexity of this operation is $ O(n^2d) $, where $n$ represents the number of nodes and $d$ indicates the dimension of the features.
>
> For large-scale graphs, we change the order of multiplication to $ X^{(K)} softmax(-X_{0}^TX_{0}) $ following [1]. This alteration reduces the complexity to $O(nd^2)$. In scenarios where $n \gg d$, this modified approach significantly decreases the computational burden. Remark that we also propose a scalable version of Label-SBP which even increases the sparsity of the graph. So we recommend using Label-SBP for large-scale graphs instead of Feature-SBP.
>
> We have included a more detailed explanation in Appendix K.
>
> [1] Guo, Xiaojun et al. “ContraNorm: A Contrastive Learning Perspective on Oversmoothing and Beyond.” *ArXiv* abs/2303.06562 (2023).
> ***
> Thank you for the insightful comment, which makes our paper more complete. Please let us know if there is more to clarify.

---

> > ### Comment · Reviewer_oeBC · 2024-11-22
> >
> > Thank the authors for the feedbacks. I appreciate the effort and I find the relation between SID and model performance to be convincing, therefore I have raised my score.
> > I have one further concern: Using Dirichlet energy as a way to quantify oversmoothing is reasonable. However, it appears to me that you are using Dirichlet energy over the original graph, yet the message passing rule is defined over the signed graph, which is by nature topologically different with the original graph and it appears that a non-vanishing convergence of Dirichlet energy is somewhat trivial. I think the discussion shall be conducted in a more careful way (As you have mentioned in line 1270 that your solution is **optimal**, this is somewhat too bold an assertion as you haven't shown any formal optimality results).

---

> ### Author Response · Authors · 2024-11-23
> **Thank you Reviewer oeBC**
>
> Thank you very much for your reply. We're glad to know that our responses have addressed your concerns and really appreciate your positive scores.
>
>
> We admit that Dirichlet energy is based on the neighbor nodes where the structural balance graph has more neighbors (both negative and positive edges) than the original graph, where we provided a more detailed definition in Appendix F highlighted in orange.
> On the other hand, we solve the oversmoothing in the original graph through the artificial edges of the constructed signed graph (SBP).  We proved that under our constructed structural balance signed graph, oversmoothing in the original graph can be alleviated (Theorem 4.3).
> Furthermore, in Theorem 4.1, when $\beta=0$, the message passing is based on the original graph, which falls under the first case where $\lim_{t \to \infty}\epsilon(X(t))\to 0$.
>
> As for the optimal solution mentioned in line 1270, thank you for your attentive reading, we would like to change it to a more rigorous expression: the provable rather than the optimal. The term "optimal solution" should encompass not just the alleviation of oversmoothing, but also a more comprehensive set of factors, such as classification performance and the distribution of features.
>
> Thank you again for your responsible discussion and suggestions on our paper which encouraged us a lot.

---

### Official Review · Reviewer_HjgM · 2024-11-04

**Soundness:** 3
**Presentation:** 3
**Contribution:** 3
**Rating:** 6
**Confidence:** 3

**Summary:**

This work explores the phenomenon of oversmoothing that is prevalent in GNNs, where the node representations become homogenized as the number of network layers increases, leading to a degradation of model performance. Many strategies have been proposed to address this problem, but they are based on different heuristics and lack a unified understanding. By revisiting the concept of signed graphs, this paper shows that a wide range of anti-oversmoothing methods can be considered as propagation over the corresponding signed graphs containing positive and negative edges. Using structural balance theory, the authors describe the asymptotic behavior of existing methods and reveal that they deviate from the ideal structural balance graph that effectively prevents oversmoothing and improves node classification performance. Driven by this unified analysis and theoretical insights, the authors propose structural balance propagation (SBP), which explicitly designs the addition of negative edges to improve the structural balance of the graph and thus prevents undesirable oversmoothing across different classes. Experimental results on synthetic and real-world datasets confirm the effectiveness of the proposed method.

**Strengths:**

Originality: this paper shows how existing anti-oversmoothing methods can be understood from the perspective of signed graphs and introduces a new propagation method SBP. This not only provides a novel perspective on existing methods, but also proposes a new solution to the oversmoothing problem, which has a nice originality.

Quality: the article systematically analyzes a variety of anti-oversmoothing methods and also provides theoretical and empirical validation of the proposed SBP method to demonstrate its effectiveness. The experimental part uses both synthetic and real-world datasets to demonstrate the effectiveness of the methods. The overall quality is good.

Clarity: the paper performs well on clarity. The proposed theorems are all proved clearly, and the background and previous proposed methods are discussed comprehensively.

Significance: this paper not only provides a new approach to understanding and solving the oversmoothing problem, but also provides a unifying framework for existing anti-oversmoothing methods and promotes a deeper understanding of the mechanisms behind the oversmoothing phenomenon in GNNs. The significance of this work lies in the fact that it not only improves our understanding of existing techniques, but also proposes promising new tools to deal with practical problems.

**Weaknesses:**

Although the experiments show the superior performance of the SBP method on a wide range of datasets, an in-depth analysis of some specific cases is missing, e.g., why the Cornell as a heterogeneous dataset decreases rapidly as the K grows does not provide an explanation. Further analysis could help understand the behavioral patterns of SBP under different conditions and how to optimize its performance. The paper proposes the SID as a measure of the degree of structural balance, but it does not specifically show how to utilize the SID in practice to guide the design or adjustment of SBPs. If some examples can be provided to show how to optimize SBPs according to SID, the combination of theory and practical application will be closer.

**Questions:**

In lines 361 and 362, the A_f is defined as -XX^T, which means A_f is a negative matrix. This means that all nodes are attracting each other rather than repelling, with the level of attraction depending on the similarity between node features. This seems quite different from the Label-enhanced SBP. Could you explain it in more detail? Additionally, it seems to me that the feature-enhanced SBP essentially uses an attention mechanism to strengthen connections between nodes with similar features. Wouldn’t using GAT yield good results in this case? In the experiments, only GCN-related methods were compared.

In line 394, the dimension of H^T H is d×d, which is different from A ̂ (dimension is n×n) in Equation (7). How do you solve this dimension gap?

In Figure 3 (a) Cornell, the Acc decreases rapidly as the K grows. Could you explain the reason?

In line 469, different K range should be used for different datasets. I think this is a spelling mistake and please correct it.

---

> ### Author Response · Authors · 2024-11-20
> **Answer to Weaknesses**
>
> Thank you very much for your positive feedback and acknowledge our theoretical insights! We address your remaining concerns below.
>
> ***
> **w1** An in-depth analysis of some specific cases is missing, e.g., why the Cornell as a heterogeneous dataset decreases rapidly as the K grows does not provide an explanation. Further analysis could help understand the behavioral patterns of SBP under different conditions and how to optimize its performance.
>
> **a1**  Thank you for your question about the in-depth analysis of our SBP.
>
> - We admit that SBPs decrease rapidly as the K grows in Cornell compared to Cora and Citeseer datasets in Figure 3 (a), this is because we fix the negative weight $\beta=1$ and that is enough to show SBP’s superiority over other baselines. However, since $\beta$ is the weight of the negative adjacency matrix (Equation 7) representing the repulsion between different nodes, the best performance of SBP appears when $\beta$ is larger in the heterophilic graphs as discussed in Section 5.3 paragraph 2 of the paper, so the result in Figure 3 (a) is not the best performance of our SBP.
> - To further show the effectiveness of our SBP, we experiment on Cornell with different $\beta$, it seems that the best $\beta$ is 20 where the performance increases 25 points in deep layer 50.
>
> | **Cornell** | **2** | **5** | **10** | **20** | **50** |
> | --- | --- | --- | --- | --- | --- |
> | $\beta=0.1$ | 72.97 ± 0.00 | 67.57± 0.00 | 51.53±0.00 | 35.14±0.00 | 29.73±0.00 |
> | $\beta=1$ (default set) | 72.97 ± 0.00 | 67.57± 0.00 | 51.53±0.00 | 45.95±0.00 | 35.14±0.00 |
> | $\beta=10$ | 70.27 ± 0.00 | 67.57± 0.00 | 58.11±1.35 | 51.53±0.00 | 51.53±0.00 |
> |$ \beta=20$ (best) | 70.27 ± 0.00 | 70.27 ± 0.00 | 67.57±0.00 | 59.46±0.00 | 59.46±0.00 |
> |$ \beta=50$ | 64.60 ± 0.00 | 40.54±0.00 | 40.54±0.00 | 40.54±0.00 | 40.54±0.00 |
>
> We discuss the hyperparameter study and reveal an intriguing phenomenon in Section 5.3 of the paper: as the repulsion weight $\beta$ increases, SBP's performance improves in heterophily graphs but declines in homophily graphs. This behavior, also observed in [1], further confirms the impact of SBP's negative adjacency matrix in heterophily graphs.
>
> [1]Wang, Yuelin, et al. "Acmp: Allen-cahn message passing for graph neural networks with particle phase transition." *arXiv preprint arXiv:2206.05437* (2022).
>
> ***
> **w2** The paper does not specifically show how to utilize the SID in practice to guide the design or adjustment of SBPs. If some examples can be provided to show how to optimize SBPs according to SID, the combination of theory and practical application will be closer.
>
> **a2**  Thank you for the comment.
>
> We would like to clarify that the motivation of the proposed metric SID is to numerically compare different oversmoothing alleviation methods that can be unified under our signed graph analysis.
>
> - In Theorem 4.3 of the paper, we prove that the linear signed graph propagation can alleviate over-smoothing even if the layers are infinite as long as it satisfies the structurally balanced property.  However, the complete structural balance condition is hard to strictly satisfy in practice since it requires intra-label nodes only to have positive edges and inter-label nodes to have negative edges. So we propose the numerical metric SID to measure the distance between the equivalent signed graph by different methods (Section 3 in the paper) to the ideal structural balance, by counting the number of edge signs that must be changed to achieve the structural balance.
>
> - Label-SBP and Feature-SBP can thus be seen as utilizing the SID in principle to guide the signed graph propagation.  In particular, Proposition 4.7 shows that Label-SBP has a direct connection to SID such that a larger training ratio leads to lower SID and thus better structural balance. We will study how to design a method other than SBP that is based on SID in the future.
>
> ***
> We give more discussion about the weaknesses of the heterophilic experiments in Appendix L3.3 and more observations of SID in Appendix H.4.

---

> ### Author Response · Authors · 2024-11-20
> **Answer to Questions**
>
> **q1.1**  the A_f is defined as -XX^T, which means A_f is a negative matrix. This means that all nodes are attracting each other rather than repelling, with the level of attraction depending on the similarity between node features. This seems quite different from the Label-enhanced SBP. Could you explain it in more detail?
>
> **a1.1** Thank you for your attentive reading, and we will provide a more detailed explanation of Feature-SBP.
>
> - Feature-SBP prevents nodes from attracting each other by introducing a negative adjacency matrix $A_f$ based on feature similarity. When nodes $i$ and $j$ have different labels, their normalized features $x_i$ and $x_j$ are likely to differ. The similarity between $x_i$ and $x_j$, denoted as $x_i * x_j \in [-1, 1]$, tends toward negativity. Consequently, the value $-x_i * x_j$ moves toward positivity, resembling an assignment of 1 in cases of dissimilar labels in Label-SBP. And so does the same label situation.
> - It is important to note that Label-SBP effectively demonstrates the practical implications of structural balance theory to counter oversmoothing, Feature-SBP is introduced as an alternative when the training label raw is low when Label-SBP's performance may diminish.
>
> ***
> **q1.2** it seems to me that the feature-enhanced SBP essentially uses an attention mechanism to strengthen connections between nodes with similar features. Wouldn’t using GAT yield good results in this case? In the experiments, only GCN-related methods were compared.
>
> **a1.2**  Thank you for the good question. We would like to clarify the difference between Feature-SBP and GAT:
>
> - GAT uses feature similarity to construct the usual positive graph for message-passing.
> - On the other hand, Feature-SBP uses feature similarity to construct the negative graph used in signed message-passing. In particular, given the design in (6) and (7), it would construct negative edges among nodes that have dissimilar features and thus repel them in message-passing.
>
> To verify that such a repulsion effect in Feature-SBP makes it different from vanilla GAT in practice, we further compare SBP with GAT across different layers, and the results show that our proposed Label/Feature-SBP significantly outperforms the vanilla GAT in the deep layers (layer=8,16,32,64).
>
> | **#layers** | **2** | **4** | **8** | **16** | **32** | **64** |
> | --- | --- | --- | --- | --- | --- | --- |
> | cora |  |  |  |  |  |  |
> | GAT | **81.48± 0.48** | **80.69±0.93** | 58.59±1.95 | 25.17±5.67 | 31.93±0.21 | 28.38±0.00 |
> | feature-sbp | 80.44 ±0.83 | 79.26 ±1.18 | 78.56 ± 0.59 | 77.22 ±0.55 | 73.65 ±0.48 | 61.62 ±5.24 |
> | label-sbp | 80.31 ±0.70 | 79.16 ±1.30 | **79.50 ± 0.00** | **77.43 ±1.49** | **74.52 ±0.36** | **65.02 ±2.97** |
> | citeseer |  |  |  |  |  |  |
> | GAT | **69.91±0.86** | **67.47±0.22** | 44.71±3.07 | 23.48±1.36 | 24.40±0.40 | 25.95±2.17 |
> | feature-sbp | 67.38 ±0.66 | 66.94 ±0.00 | **66.29 ±0.02** | 65.35 ±1.99 | **61.43 ±0.00** | 42.09 ±1.65 |
> | label-sbp | 67.23 ±0.64 | 66.72 ±0.00 | **66.29 ±0.89** | **65.50 ±2.13** | 59.93 ±0.85 | **44.41 ±1.57** |
>
> We add more discussion about our SBP adapted to different baselines in Appendix L3.2 and L3.5.
>
> ***
> **q2** the dimension of H^T H is d×d, which is different from A ̂ (dimension is n×n) in Equation (7). How do you solve this dimension gap?
>
> **a2**  Thank you for your question, let me clarify it and explain it more clearly.  we reorder the matrix multiplication from $softmax(-H_{0}H_{0}^T) H^{(K-1)} $ to $ H^{(K-1)} softmax(-H_{0}^T H_{0} )$. This preserves the distinctiveness of node representations across the feature dimension, rather than across the node dimension as in the original node-level repulsion. We have added more details in Appendix K.
>
> ***
> **q3** In Figure 3 (a) Cornell, the Acc decreases rapidly as the K grows. Could you explain the reason?
>
> **a3** This is the same question as weakness 1. Please refer to a1 to w1 in our answer to the weakness.
>
> ***
> **q4** In line 469, different K range should be used for different datasets. I think this is a spelling mistake and please correct it.
>
> **a4** Sorry for the spelling mistake, we have changed it to '$K \in $ {$2, 5, 10, 20, 50 $} for heterophilic datasets'.
>
> ***
> Hope you find these new experiments and explanations satisfactory! Thank you for the insightful comment, which makes our work more complete.

---

> ### Author Response · Authors · 2024-11-25
>
> Dear Reviewer HjgM,
>
> Thanks for your constructive and valuable comments on our paper. We have tried our best to address your concerns and revised our paper accordingly. Could you please have a check if there are any unclear points? We are certainly happy to answer any further questions.

---

> > ### Comment · Reviewer_HjgM · 2024-11-25
> > **Official Response by Reviewer HjgM**
> >
> > Dear Authors,
> >
> > Thank you for your efforts in answering the questions. Most of the concerns have been addressed, so we will keep the positive score.

---

### Comment · Area_Chair_wKs6 · 2024-11-23

Dear Reviewers,

The authors have uploaded their rebuttal. Please take this opportunity to discuss any concerns you may have with the authors.

AC

---

### Meta-Review · Area_Chair_wKs6 · 2024-12-20

**Metareview:**

This paper tackles the oversmoothing problem in Graph Neural Networks (GNNs) by examining existing anti-oversmoothing techniques through the lens of signed graphs. The authors propose a new framework based on structural balance theory and introduce a method called Structural Balance Propagation (SBP), which integrates negative edges into the graph to prevent oversmoothing. The paper provides both theoretical analysis and empirical validation, showing that SBP outperforms existing methods on synthetic and real-world datasets.

While the proposed SBP model shows promising results, the reviewers have identified several weaknesses that need to be addressed:
1. Over-smoothing has been extensively studied in the GNN community, and it is unclear whether SBP offers significant advantages over existing methods.
2. The paper overlooks recent techniques like GCNII, DropMessage, and methods with residual connections, leading to an incomplete comparison with state-of-the-art solutions for addressing oversmoothing. Although the rebuttal provides a comparison to GCNII, the results are inconsistent with the previous papers.

Based on these weaknesses, we recommend rejecting this paper. We hope this feedback helps the authors improve their paper.

**Additional Comments On Reviewer Discussion:**

During the rebuttal, the authors have made significant efforts to improve the paper, including better presentation and more experiments, which help the reviewers to better understand the contribution of the paper. However, the reviewers still have concerns regarding the technical depth and contributions over existing methods that address over-smoothing issue. Based on their feedback, I recommend rejecting the paper.

---

### Decision · Program_Chairs · 2025-01-22

Reject